# PRIMA.CPP: FAST 30-70B LLM INFERENCE ON HETEROGENEOUS AND LOW-RESOURCE HOME CLUSTERS

**Zonghang Li**[1,2]  **Tao Li**[2]  **Wenjiao Feng**[2]  **Rongxing Xiao**[2]  **Jianshu She**[1]  **Hong Huang**[3]
**Mohsen Guizani**[1]  **Hongfang Yu**[2]  **Qirong Ho**[1]  **Wei Xiang**[4]  **Xue Liu**[1]
[1]MBZUAI  [2]UESTC  [3]City University of Hong Kong  [4]La Trobe University

## ABSTRACT

On-device inference offers privacy, offline use, and instant response, but consumer hardware restricts large language models (LLMs) to low throughput and capability. To overcome this challenge, we present prima.cpp, a distributed on-device inference system that runs 30-70B LLMs on consumer home clusters with mixed CPUs/GPUs, insufficient RAM/VRAM, slow disks, Wi-Fi links, and heterogeneous OSs. We introduce pipelined-ring parallelism (PRP) to overlap disk I/O with compute and communication, and address the prefetch-release conflict in mmap-based offloading. We further propose Halda, a heterogeneity-aware scheduler that co-optimizes per-device CPU/GPU workloads and device selection under RAM/VRAM constraints. On four consumer home devices, a 70B model reaches 674 ms/token TPOT with <6% memory pressure, and a 32B model with speculative decoding achieves 26 tokens/s. Compared with llama.cpp, exo, and dllama, our proposed prima.cpp achieves 5-17× lower TPOT, supports fine-grained model sizes from 8B to 70B, ensures broader cross-OS and quantization compatibility, and remains OOM-free, while also being Wi-Fi tolerant, privacy-preserving, and hardware-independent. The code is available at
`https://gitee.com/zonghang-li/prima.cpp`.

## 1 INTRODUCTION

As a representative of next-generation AI, embodied AI (Gupta et al., 2021) is acutely sensitive to privacy protection, stable connectivity, and long-term financial costs. Home surveillance, always-listening assistants, and companion robots cannot send raw interactions to the cloud. Cloud services also suffer from frequent network failures, queuing, timeouts, and their costs scale linearly with token usage and service time. Consequently, embodied AI is moving from the cloud to the device.

However, on-device inference is constrained by modest chips and small RAM, struggling to run models beyond 8B (MLC, 2025; Lugaresi et al., 2019; Ghorbani, 2025), yet reliable long-term planning and tool use require 32B or more. To support larger models, disk offloading helps capacity but at the cost of speed (Gerganov, 2025; Li, 2023), e.g., Qwen 2.5-14B Q4K on an 8 GiB Mac M1 laptop takes 10 s/token with llama.cpp. Recent efforts use dedicated devices (e.g., Jetson, Mac Studio) for larger models and faster speed (Ye et al., 2024; 2025), but they are too expensive for most households. This motivates our goal: *Can we deliver fast 30-70B inference on user-owned consumer devices that meet embodied AI's demands summarized in Table 1?*

Distributed inference may be the only solution to improve model size and speed while preserving the original outputs. Users usually own multiple devices, e.g., laptops, PCs, phones, and tablets. Some laptops and PCs have low-end GPUs like NVIDIA 20-/30-/40-series, and some Macs have Apple Silicon GPUs. By pooling the computing and memory of home devices, we can run larger models.

Despite recent progress, four limitations remain: (a) Existing systems require sufficient aggregated RAM/VRAM to hold the full model (Ye et al., 2024; Exo, 2025; Tadych, 2025; Zhang et al., 2025; Lee et al., 2024; Zhao et al., 2023; Zhang et al., 2024), driving up hardware costs and restricting model size. (b) Disk offloading slows inference, causing time-per-output-token (TPOT) of tens of seconds (Li et al., 2025). (c) Layer partitioning relies on the strong assumption in (a) and overlooks heterogeneity in OS-specific memory reclamation and disk access. (d) All devices are assumed to

Table 1: Comparison of cloud and local LLM deployments. TPS[1] indicates the per-request token rate, TPS[2] denotes TPS values measured across 32-70B models. OA denotes open-access.

|  |  | Cost ($) | Privacy | Speed (TPS) | Queue | Net | Support Model |
|---|---|---|---|---|---|---|---|
| Cloud | App | < 100 | ✗ | 10-70[1] | ✓ | ✓ | Specific models |
|  | API | < 100 | ✗ | 10-70[1] | ✓ | ✓ | Specific models |
|  | Dedicated | < 1,000 | ✗ | 10-35[2] | ✗ | ✓ | Any OA model |
| **Local** | Dedicated | > 1,000 | ✓ | 4-17[2] | ✗ | ✗ | Any OA model ($\leq$70B) |
|  | **Consumer** | **Free** | ✓ | 2-26[2] | ✗ | ✗ | **Any OA model ($\leq$70B)** |

be necessary, even though removing slow ones could improve speed. These limitations raise two questions that motivate our design: *(Q1) How can memory constraints be relaxed to run larger models? If disk offloading is required, how can disk latency be hidden? (Q2) Given Q1, how can we design heterogeneity-aware layer partitioning and identify bottleneck devices?*

We propose prima.cpp, the first distributed inference system for consumer home clusters, with mixed CPUs/GPUs, insufficient RAM/VRAM, slow disks, Wi-Fi links, heterogeneous OSs, and capable of running 30-70B models at practical speed. To address Q1, Section 3.1 proposes pipelined-ring parallelism with prefetching to overlap disk latency and resolve the prefetch-release conflict in mmap. To address Q2, Section 3.2 models heterogeneity in compute, communication, memory, and OS-specific memory reclamation and disk optimizations. Section 3.3 develops Halda to find the optimal layer partitioning and select the best-performing devices. Section 4 presents experiments on real home clusters and shows that, even under limited memory, a 70B model runs locally at 674 ms/token with <6% memory pressure. With speculative decoding (Leviathan et al., 2023), a 32B model achieves 26 tokens/s, which is fast enough and paves the way for household embodied AI.

As summarized in Table 1, prima.cpp runs on existing, free devices; all data stays local; no queueing or timeouts; offline; and supports models up to 70B. To the best of our knowledge, *prima.cpp is the first on-device system to deliver practical performance for 30-70B models in such constrained environments, without relying on specialized hardware (e.g., Jetson, NPUs) or altering model outputs.*

## 2 RELATED WORK

**On-device LLM systems.** Most of them run on a resource-constrained device and can only handle small models. MLC-LLM (MLC, 2025) and MediaPipe (Lugaresi et al., 2019) bring 7B models to mobile phones and browsers. PocketPal AI (Ghorbani, 2025), which runs on Android using llama.cpp (Gerganov, 2025), supports models up to 3.8B. Some efforts push for larger models, like AirLLM (Li, 2023), which loads only the needed layers to save memory but at the cost of speed. Others turn to high-end hardware, such as the Apple M2 Ultra (192 GiB RAM) for a 65B model or kTransformers (kvcache ai, 2025) (382 GiB RAM) for a 671B model ($\sim$75 GiB for 70B). Like kTransformers, HeteGen (Zhao et al., 2024) also collaborates with the CPU and GPU on one device to accelerate inference. These setups (large RAM, advanced CPUs with specific instruction sets, dedicated hardware) go far beyond common home devices and are inaccessible to most households.

**Distributed on-device LLM systems.** Such distributed systems follow two main parallelism paradigms: tensor parallelism and pipeline parallelism.

- *Tensor Parallelism (TP).* TP splits tensors across devices to share the load (Shoeybi et al., 2019). To speed up all-reduce, dllama (Tadych, 2025) uses USB4 and Thunderbolt 5 for fast connections, and AirInfer (Zhang et al., 2025) uses wireless analog superposition to perform all-reduce over the air. Due to device heterogeneity, Hepti (Lee et al., 2024) optimizes workload partitioning with three slicing strategies for different memory budgets, and Galaxy (Ye et al., 2024) prioritizes compute power first, then memory, to maximize speed and avoid OOM. These systems reside the full model in the aggregate memory. With limited aggregate memory, only small models can run. TPI-LLM (Li et al., 2025) loads model layers on demand and hides disk loading with prefetching. This enables low-end devices with only 4 GiB of RAM to run a 70B model, but at 30 s/token, making it impractical.

Table 2: Comparison of distributed on-device LLM systems. (Abbr.: Quantization, Heterogeneity)

| | Type | Backends | Mem | Quant. | Mem Stress | Speed | Hete. |
|---|---|---|---|---|---|---|---|
| dllama | TP | CPU | RAM | Q4 | Critical | Slow | ✗ |
| AirInfer | TP | CPU | RAM | FP32 | Critical | Slow | ✓ |
| Hepti | TP | CPU | RAM | FP32 | Critical | Slow | ✓ |
| Galaxy | TP+SP | CPU/GPU | RAM/VRAM | FP32 | Critical | Slow | ✓ |
| TPI-LLM | TP | CPU | RAM | FP32 | Medium | Slow | ✗ |
| exo | PP | CPU/GPU | RAM/VRAM | Q4+FP32 | Critical | Slow | ✓ |
| LinguaLinked | PP | CPU | RAM | Q8/FP32 | Critical | Slow | ✓ |
| EdgeShard | PP | CPU/GPU | RAM/VRAM | FP32 | Critical | Fast | ✓ |
| **prima.cpp** | **PRP** | **CPU&GPU** | **RAM&VRAM** | **Q4/IQ1** | **Low** | **Fast** | ✓ |

*- Pipeline Parallelism (PP).* Due to Wi-Fi's high latency, pipeline parallelism becomes more suitable for home clusters as it requires less P2P communication. Exo (2025); Zhao et al. (2023); Zhang et al. (2024) split the model into segments and assign them to devices based on memory, compute, and network conditions. Each device computes its segment and passes the result to the next, until the last device outputs the next token. Exo (2025) partitions model segments based on memory ratio; LinguaLinked (Zhao et al., 2023) uses linear optimization to solve the device assignment problem; and EdgeShard (Zhang et al., 2024) uses dynamic programming.

However, these systems either require dedicated hardware (e.g., Jetson AGX/Nano) or sufficient aggregate memory to reside the full model, support only CPU or GPU backends, overlook heterogeneity (especially in disk offloading), and impose high memory pressure. Table 2 summarizes their features. These limitations raise hardware costs, restrict use to small models, cause slow inference and device freezes, and discourage users from deploying LLMs at home.

Instead, prima.cpp runs on consumer devices, supports disk offloading to run larger models, and uses pipelined-ring parallelism (PRP) to overlap disk latency. It runs on both GPUs and CPUs, combines RAM and VRAM, and models system heterogeneity to optimize GPU-CPU layer partitioning per device. Moreover, it stores model weights in the OS page cache, allowing the OS to reclaim memory as needed to preserve user experience. These features distinguish prima.cpp from existing systems.

## 3  PRIMA.CPP: USE PIPELINED-RING PARALLELISM IN LLAMA.CPP

### 3.1  PIPELINED-RING PARALLELISM WITH PREFETCHING

PP is effective for batched inference when aggregated memory is abundant; however, this prerequisite rarely holds in households. In such cases, on-device LLMs typically serve very few users at low frequency, often a single request at a time[1], which prevents mini-batching and leaves significant pipeline bubbles. Besides, home devices are few and have limited available memory[2], making it difficult for users to build a cluster that can hold the full model in memory.

To relax memory constraints, we extend PP with mmap, building on llama.cpp (Gerganov, 2025): when computation requires the layers, mmap loads them from external storage on demand, and lets the OS evict them under memory pressure, so a small-memory cluster can run larger models. To hide disk loading latency, we employ prefetching, allowing devices to preload the next model segment.

This naive design supports larger models with low memory pressure but suffers from slow speed, because *prefetching will fail due to prefetch-release conflict*: if disk reads are fast, later-loaded layers will evict earlier prefetched layers from page cache, so when computation begins, the needed layers are no longer cached (see Appendix A.1 for details). This triggers page faults and reloads, negating the benefit of prefetching, and pipeline bubbles remain significant (see Fig. 6 d,e in Appendix A.2). Appendix A.14 provides empirical evidence for this conflict and performance breakdown: even after prefetching completes, required model weights still need to be reloaded during computation.

---

[1]We target a single request, but this can be extended to batches via dynamic batching (see Appendix A.11).
[2]We cannot occupy all memory, as this may disrupt other apps (e.g., TikTok) and cause users to kill the LLM.

**Pipelined-ring parallelism (PRP) with prefetching.** To address the prefetch-release conflict, we further propose PRP, which connects devices end-to-end in a ring and runs multiple rounds to predict one token. In each round, devices prefetch different layer segments from the disk, which overlap with other devices' ongoing operations, such as computing, communication, and disk loading. Since only a small segment (whose size is referred to as the layer window size) is loaded per round, memory overflow can be avoided, and prefetched layers are less likely to be evicted, thus mitigating the prefetch-release conflict (see Figs. 4 and 5 in Appendix A.2 for examples). In addition, PRP processes both input and output on the head device, providing enhanced interaction privacy.

In Fig. 1, all devices share a layer window size of 2. Each device processes two model layers per round and takes 3 rounds to predict one token. Fig. 6 a,b in Appendix A.2 show the PRP timeline on homogeneous devices with fast and slow disks. On fast disks, prefetching latency is fully overlapped; on slow disks, CPU and disk operate alternately, removing bubbles but leaving residual page-fault loading latency, since required layers may not be fully loaded when computation begins. Fig. 2 evaluates this multi-round design: for large models, PRP reduces PP's TPOT by about 50% (PP is equivalent to PRP at $k = 1$), while for small models, PRP converges to PP with similar TPOT. Appendix A.12 presents the overall design.

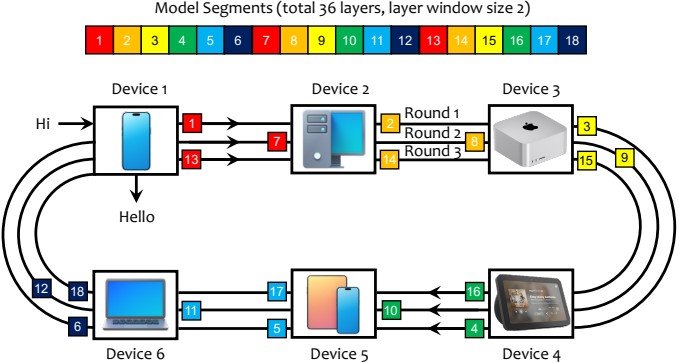

Figure 1: Pipelined-ring parallelism. In this case, 6 devices handle a 36-layer model. With a layer window size of 2, the model is split into 18 segments and assigned to 6 devices in ring order, so each device needs 3 rounds to predict one token.

In practice, however, home devices are heterogeneous. Using a uniform layer window size still produces pipeline bubbles (see Fig. 6c in Appendix A.2), whereas assigning different window sizes per device can reduce them (see Fig. 6f in Appendix A.2). In general, stronger devices should be assigned larger window sizes. However, determining which devices are stronger and how large their window sizes should be is challenging. Beyond common heterogeneity in computing, communication, and memory, disk loading is heavily affected by OS-specific memory reclamation and disk read throughput. Such complex heterogeneity makes disk latency difficult to quantify.

## 3.2 Layer-to-device Assignment Problem

As mentioned in Table 2, prima.cpp is designed to use multiple backends[3]. Within a layer window, some layers reside in VRAM and execute on the GPU, while others are offloaded to RAM and executed on the CPU. If the layers assigned to the CPU exceed available RAM, the overflow is further offloaded to external storage and reloaded into RAM via mmap[4]. This leads to two challenges: *How to set the layer window size for each device? Which layers run on the GPU and which on the CPU?*

Prior work assumes sufficient aggregated VRAM and proposes heuristic partitioning strategies. For example, Exo (2025) partitions model layers based on memory ratio, so devices with more RAM/VRAM handle more layers. Ye et al. (2024) partition by compute power and migrate layers from OOM devices to those with free memory. These heuristics work on some testbeds but are not always optimal (see Fig. 10 in Appendix A.10), since memory size does not guarantee compute power, and under disk offloading, devices with larger memory but weaker CPUs/GPUs can perform worse than those with less memory but faster CPUs and disks.

---

[3]Our prototype supports CUDA, Metal, and CPU backends, but with minor adaptation, it can also work with others such as Vulkan and ROCm.

[4]We pin model layers in VRAM and avoid paging them as in disk-RAM; otherwise, disk-RAM and RAM-VRAM transfers would contend for PCIe (or shared) bandwidth, increase disk reload volume, and amplify disk loading latency, which offsets the gain from GPU acceleration.

To achieve optimal partitioning, TPOT must be quantified through an analytical model. Building such a model is challenging: beyond device heterogeneity, we must account for CPU-GPU coordination, memory contention, OS-specific memory reclamation, disk optimizations, and quantization. For instance, some PCs have dedicated GPUs with separate VRAM; others (e.g., Mac M-series) adopt a UMA architecture where CPU and GPU share memory, and OS reclamation will be more aggressive; some are NUMA systems without GPUs, and they differ in reclamation thresholds across OSs; even on the same device, OS reclamation differs with or without Metal; Linux optimizes sequential disk reads, making reloads faster; quantization influences compute latency, memory access, disk loading, and RAM/VRAM constraints, ultimately shaping the partitioning strategy.

After extensive performance analysis in Appendix A.3 and preliminary experiments, we formalize this as the layer-to-device assignment (LDA) problem as follows.

**Definition 1** (**Layer-to-device assignment**). *Assume there are $M$ devices, $w_m$ is the layer window size on device $d_m$ and $n_m$ is the number of GPU layers within $w_m$. Let the decision variables be $\boldsymbol{w}^{\mathrm{T}} = [w_1, w_2, \cdots, w_M]$ and $\boldsymbol{n}^{\mathrm{T}} = [n_1, n_2, \cdots, n_M]$. Our objective is to find the optimal $\boldsymbol{w}$ and $\boldsymbol{n}$ that minimizes the TPOT:*

$$\min_{\boldsymbol{w},\boldsymbol{n}} \quad L \cdot \frac{\boldsymbol{a}^{\mathrm{T}} \cdot \boldsymbol{w} + \boldsymbol{b}^{\mathrm{T}} \cdot \boldsymbol{n} + \boldsymbol{e}^{\mathrm{T}} \cdot \boldsymbol{c}}{\boldsymbol{e}^{\mathrm{T}} \cdot \boldsymbol{w}} + \kappa, \tag{1}$$

$$\text{s.t.} \quad w_m \in \mathbb{Z}_{>0}, n_m \in \mathbb{Z}_{\geq 0}, n_m \leq w_m \leq L, \tag{2}$$

$$L - k(\boldsymbol{e}^{\mathrm{T}} \cdot \boldsymbol{w}) = 0, k \in \mathbb{Z}_{>0}, \tag{3}$$

$$\boldsymbol{P}_w \cdot \boldsymbol{w}' + \boldsymbol{P}_n \cdot \boldsymbol{n}' + \boldsymbol{e}^{\mathrm{T}} \cdot \boldsymbol{w} \cdot \boldsymbol{z} \leq 0, \tag{4}$$

$$-\boldsymbol{P}_n^{gpu} \cdot \boldsymbol{z}^{gpu} \cdot \boldsymbol{e}^{\mathrm{T}} \cdot \boldsymbol{w} + \boldsymbol{P}_n^{gpu} \cdot \boldsymbol{n} \leq 0. \tag{5}$$

*where $L$ is the number of model layers; $\boldsymbol{a}, \boldsymbol{b}, \boldsymbol{c}$ are latency coefficient vectors determined by compute latency, memory access latency, disk loading latency, and communication latency on each device; $\kappa$ is a constant latency offset; $k$ is the number of rounds to predict one token; $\boldsymbol{w}'$ and $\boldsymbol{n}'$ are the extended vectors of $\boldsymbol{w}$ and $\boldsymbol{n}$; $\boldsymbol{z}$ and $\boldsymbol{z}^{gpu}$ are constraint vectors for RAM and VRAM; $\boldsymbol{P}_w, \boldsymbol{P}_n, \boldsymbol{P}_n^{gpu}$ are diagonal matrices that activate or deactivate the decision variables; and $\boldsymbol{e}$ is an all-ones vector.*

Constraint (2) ensures the layers do not exceed the limit. Constraint (3) enforces that all devices are assigned an equal number of windows and all windows are filled[5]. Constraints (4) and (5) ensure that RAM and VRAM usage stay within limits. Table 6 summarizes the key symbols. To construct $\boldsymbol{a}, \boldsymbol{b}, \boldsymbol{c}, \kappa, \boldsymbol{w}', \boldsymbol{n}', \boldsymbol{z}, \boldsymbol{P}_w, \boldsymbol{P}_n$, we categorize devices into four sets: (a) *Set $\mathcal{M}_1$:* macOS with Metal disabled and insufficient RAM; (b) *Set $\mathcal{M}_2$:* macOS with Metal enabled and insufficient RAM; (c) *Set $\mathcal{M}_3$:* Linux and Android with insufficient RAM; (d) *Set $\mathcal{M}_4$:* devices with sufficient RAM or slow disk. For Sets $\mathcal{M}_1$-$\mathcal{M}_3$ where devices should overload, RAM usage should stay above available RAM; for Set $\mathcal{M}_4$, where overloading is not allowed, RAM usage should stay below available RAM. We can extend other OSs to these sets, or create a new set and adjust variable dimensions.

This problem is an NP-hard integer linear fractional program (ILFP): both the numerator and denominator of the objective are linear in the decision variables, and all constraints are linear inequalities. Whether a device is overloaded depends on $\boldsymbol{w}$ and $\boldsymbol{n}$, e.g., a large $\boldsymbol{w}_m - \boldsymbol{n}_m$ will overload RAM. However, we cannot determine a device's set before solving the LDA problem, and without the set assignment, we cannot solve the LDA problem. This traps us in a circular dependency.

### 3.3 HALDA: AUTOMATIC LAYER PARTITIONING AND DEVICE SELECTION

To solve this non-standard, NP-hard LDA problem, our core ideas include: (i) transform the original problem into a set of standard integer linear programs (ILPs) by enumerating over all valid $k$ that divide $L$, and (ii) search for optimal set assignments $\mathcal{M}_1$-$\mathcal{M}_4$ by iterative optimization.

**Transform into standard ILPs.** Given that the number of layers $L$ in LLMs is typically less than 100, the integer $k$ has a limited range of values: at most 11 valid factors for any $L \leq 100$. By

---

[5]This is not mandatory in our implementation, but it can simplify the problem model.

---

**Algorithm 1:** Heterogeneity-Aware Layer-to-Device Allocation (HALDA)

---

**1** Initialize layer windows $\boldsymbol{w}$ proportionally to devices' memory budgets, and GPU layers $\boldsymbol{n} \leftarrow \boldsymbol{0}$;
**2** Calculate platform-specific coefficients $\alpha_m, \beta_m, \xi_m$ for each device $m$;
**3** Calculate valid factors $\mathcal{K}_L$ of $L$ (excluding $L$);
**4** **while true do**
**5** $\quad$ Calculate $W = \boldsymbol{e}^{\mathrm{T}} \cdot \boldsymbol{w}$ and $k = L/W$;
**6** $\quad$ Reassign devices to sets $\mathcal{M}_1, \mathcal{M}_2, \mathcal{M}_3, \mathcal{M}_4$ based on the latest $\boldsymbol{w}, \boldsymbol{n}, k$ and $\mathcal{M}_4^{\mathrm{force}}$;
**7** $\quad$ **if** *the assignment sets $\mathcal{M}_1, \mathcal{M}_2, \mathcal{M}_3, \mathcal{M}_4$ remain unchanged* **then**
**8** $\quad\quad$ $\lfloor$ **break**;
**9** $\quad$ Calculate the objective coefficients $\boldsymbol{a}, \boldsymbol{b}, \boldsymbol{c}, \kappa$, the RAM upper bound $\boldsymbol{z}$, and the
$\quad\quad$ VRAM/shared memory upper bound $\boldsymbol{z}^{\mathrm{gpu}}$ according to the updated assignment sets;
**10** $\quad$ **foreach** $k \in \mathcal{K}_L$ **do**
**11** $\quad\quad$ Solve the ILP for fixed $k$ using a solver;
**12** $\quad\quad$ Update best solution $(\boldsymbol{w}^\star, \boldsymbol{n}^\star)$ if the current objective is smaller;
**13** $\quad$ **if** *any device has free VRAM but another device is overloaded* **then**
**14** $\quad\quad$ Force the device $m_s$ from $\{\mathcal{M}_1, \mathcal{M}_2, \mathcal{M}_3\}$ with slowest disk reads into $\mathcal{M}_4^{\mathrm{force}}$;
**15** $\quad\quad$ **continue**;
**16** $\quad$ Update $\boldsymbol{w} \leftarrow \boldsymbol{w}^*$ and $\boldsymbol{n} \leftarrow \boldsymbol{n}^*$;
**17** **return** $\boldsymbol{w}^\star, \boldsymbol{n}^\star$;

---

enumerating these factors, we treat $k$ and $W$ as constants, and the problem becomes:

$$\min_{\boldsymbol{w}, \boldsymbol{n}} \quad k(\boldsymbol{a}^{\mathrm{T}} \cdot \boldsymbol{w} + \boldsymbol{b}^{\mathrm{T}} \cdot \boldsymbol{n} + \boldsymbol{e}^{\mathrm{T}} \cdot \boldsymbol{c}) + \kappa, \tag{6}$$

$$\text{s.t.} \quad w_m \in \mathbb{Z}_{>0}, n_m \in \mathbb{Z}_{\geq 0}, n_m \leq w_m \leq L, \tag{7}$$

$$\boldsymbol{e}^{\mathrm{T}} \cdot \boldsymbol{w} = W, \tag{8}$$

$$\boldsymbol{P}_w \cdot \boldsymbol{w}' + \boldsymbol{P}_n \cdot \boldsymbol{n}' + W\boldsymbol{z} \leq 0, \tag{9}$$

$$\boldsymbol{P}_n^{\mathrm{gpu}} \cdot \boldsymbol{n} - W\boldsymbol{P}_n^{\mathrm{gpu}} \cdot \boldsymbol{z}^{\mathrm{gpu}} \leq 0. \tag{10}$$

Hence, for each fixed $k$, the objective and constraints boil down to linear functions/inequalities, and the problem becomes an ILP. Then, we can run a standard ILP solver (e.g., HiGHS (Huangfu & Hall, 2018)) to obtain the optimal $\boldsymbol{w}, \boldsymbol{n}$.

**Iterative optimization for set assignment.** The problem remains unsolvable because the sets $\mathcal{M}_1$-$\mathcal{M}_4$ are unknown. To break this circular dependency, we adopt an iterative optimization procedure. We start by setting $\boldsymbol{w}$ proportional to available memory [6] and initializing $\boldsymbol{n} = \boldsymbol{0}$, which gives an initial division of devices into $\mathcal{M}_1$-$\mathcal{M}_4$. We then solve the ILPs to update $\boldsymbol{w}$ and $\boldsymbol{n}$, reassign devices, and iterate until the sets converge.

However, this approach still has defects. For example, if a device $m \in \{\mathcal{M}_1, \mathcal{M}_2, \mathcal{M}_3\}$ is assigned 30 layers, although the GPU can host 20 layers, memory constraints may end up assigning only 10 to the GPU and 20 to the CPU, underutilizing the GPU. This issue arises from an improper set initialization: if the device was initialized in $\mathcal{M}_4$, its GPU could be fully utilized. Therefore, Algorithm 1 includes a calibration step: if a GPU is underutilized (VRAM not full) while another device is overloaded (VRAM full with layers offloaded to CPU, or RAM exceeded), we move the device with slowest disk reads from the set $\{\mathcal{M}_1, \mathcal{M}_2, \mathcal{M}_3\}$ into $\mathcal{M}_4^{\mathrm{force}}$ ($\mathcal{M}_4^{\mathrm{force}} \subset \mathcal{M}_4$) and solve again. This ensures convergence to the optimal set assignment and the corresponding optimal values of $\boldsymbol{w}, \boldsymbol{n}$, and $k$.

**Complexity analysis.** The main loop alternates between set assignment and LDA solving, repeated for $T$ iterations with $T = O(M)$ in the worst case. The set assignment takes $O(M)$. For LDA, we solve a tiny ILP for each valid factor of $L$, giving $K = O(\log L)$ factors in total. Although ILPs are NP-hard, our instances are small and sparse, allowing modern solvers to finish quickly. Empirically, on 4-32 devices, the global scheduling latency is 10-12 ms. The overhead of rerunning Halda is

---

[6] $d_m^{\mathrm{avail}}$ for macOS without Metal and Linux, $d_{m,\mathrm{metal}}^{\mathrm{avail}}$ for macOS with Metal, and $d_m^{\mathrm{avail}} + d_m^{\mathrm{swapout}}$ for Android.

Table 3: Testbed configuration. Termux is used on D4 and D5.

|  | D1 | D2 | D3 | D4 | D5 | D6 |
|---|---|---|---|---|---|---|
| Device | Mac M1 | Laptop | Desktop | Mate40Pro | Honor Pad | Mac Air |
| OS | macOS | Linux | Linux | HarmonyOS | Android | macOS |
| CPU | Apple M1 | Intel i9 | Intel i9 | Kirin 9000 | Dimensity 8100 | Intel i5 |
| CPU Cores | 8 | 8 | 16 | 8 | 8 | 4 |
| RAM (avail.) | 2.4 GiB | 4.1 GiB | 9.7 GiB | 1.9 GiB | 5.1 GiB | 6.8 GiB |
| Disk Read | 0.7 GB/s | 3.0 GB/s | 3.0 GB/s | 1.4 GB/s | 2.0 GB/s | 0.4 GB/s |
| GPU Type | Apple Silicon | 3070 | 2080TI | - | - | - |
| VRAM (avail.) | - | 8 GiB | 11 GiB | - | - | - |

negligible, and since no cross-device layer migration is required, workloads can be repartitioned whenever the task queue is empty to adapt to environmental changes.

**Device selection.** Weak devices assigned only one layer are removed, since Halda indicates that excluding them would improve speed. Appendix A.7 illustrates this selection process. For small models, Halda prefers to allocate all layers to one most powerful device, reducing prima.cpp to llama.cpp. However, if removing a device would break the communication ring (e.g., it is the only reachable hop due to network policy), Halda keeps it in the ring as a relay node but assigns it no workload. This design frees users from device selection: we only need to discover more available devices, and Halda will select the optimal subset for us.

## 4 EXPERIMENTS

We implement prima.cpp with 20K lines of code based on llama.cpp and build a real testbed of heterogeneous, low-end home devices (see Table 3). These devices connect via a local Wi-Fi router, with inter-device bandwidth ranging from 320-610 Mbps and link latency from 3-7 ms. By default, four devices (D1-D4) with aggregated RAM+VRAM of 37 GiB (insufficient for a Q4K-quantized 70B model) are used. We evaluated Llama models from 8B to 70B (Q4K) in terms of time-per-output-token (TPOT), time-to-first-token (TTFT), and memory pressure. We select llama.cpp, exo, and dllama as baselines, as they are the most popular open-source on-device (distributed) LLM inference systems[7]. Since llama.cpp is a standalone on-device system, we run it on D3, which offers the largest RAM/VRAM and highest decoding efficiency[8]; Exo is a PP system, which we run on D1-D4, but D4 failed because it requires root access on the phone; dllama and prima.cpp successfully run on D1-D4.

### 4.1 FAST INFERENCE ON LARGE MODELS

Table 4 presents TPOT and TTFT across model sizes of 8-70B[9]. As a result, prima.cpp achieves substantially lower TPOT and TTFT than all baselines, and matches llama.cpp on small models (<14B). Compared with llama.cpp, it reduces TPOT by up to $17\times$ and TTFT by up to $8\times$; against exo and dllama, it achieves at least $18\times$ lower TPOT and $42\times$ lower TTFT, without OOMs.

For small models (<14B), Halda finds that D3 can hold the entire model in VRAM, so it removes the other devices and runs only on D3. This reduces prima.cpp to the single-device case (equivalent to llama.cpp), with both achieving the same TPOT and TTFT. At 30B, as shown in Fig. 9a in Appendix A.8, D3-GPU runs out of VRAM and forces layers onto the CPU, causing llama.cpp to deteriorate rapidly. In contrast, Fig. 9d shows that Halda offloads the overflow from D3-GPU to D2-GPU, keeping TPOT and TTFT stable. At 45B, both RAM and VRAM on D3 are exhausted, and mmap in llama.cpp begins frequent reloads, incurring disk latency. At this stage, only a few pages are reloaded, so the efficiency loss is minor. At 60B, high memory stress causes more active pages to be evicted earlier, sharply increasing TPOT and TTFT. This indicates that llama.cpp is ill-suited for large models on consumer-grade devices. In contrast, Halda balances workloads across devices according to their

---

[7]At submission, llama.cpp had 87K stars, exo 31K stars, and dllama 2.7K stars. We also tried to deploy vLLM and SGLang, but as server-oriented systems, they failed to run on our home cluster.

[8]At small batch sizes, decoding is bandwidth-bound, and 2080TI offers higher memory bandwidth than 3070.

[9]Exo supports Llama 8/70B and dllama supports only Llama 3, so we mark unavailable values with "-".

Table 4: TPOT (ms/token) and TTFT (ms) on Llama models for llama.cpp, exo, dllama, prima.cpp.

| Size | llama.cpp | | exo | | dllama | | prima.cpp (w/o halda) | prima.cpp (w/o prefetch) | prima.cpp | |
|---|---|---|---|---|---|---|---|---|---|---|
| | TPOT | TTFT | TPOT | TTFT | TPOT | TTFT | TPOT | TPOT | TPOT | TTFT |
| 8B | 15 | 18 | 263 | 960 | 1150 | 4409 | 78 | 15 | **15** | **18** |
| 14B | 20 | 25 | - | - | 2201 | 10279 | 131 | 20 | **20** | **25** |
| 30B | 202 | 611 | - | - | - | - | 258 | 79 | **72** | **214** |
| 45B | 328 | 712 | - | - | 6235 | 18532 | 409 | 263 | **233** | **440** |
| 60B | 7965 | 8350 | - | - | - | - | 7053 | 532 | **468** | **990** |
| 65B | 8807 | 9662 | - | - | OOM | OOM | 12253 | 688 | **569** | **1770** |
| 70B | 10120 | 10806 | OOM | OOM | OOM | OOM | 20848 | 755 | **674** | **1793** |

capabilities. Even when disk offloading is required, it assigns workloads to devices with strong CPU and fast disk, while PRP with prefetching hides disk latency. *As a result, prima.cpp maintains stable TPOT and TTFT growth and achieves sub-second TPOT even at 70B. To the best of our knowledge, prima.cpp is the first system to achieve this speed on real consumer-grade, low-end devices*[10].

For exo and dllama, even with only a few valid samples, their limitations are already clear. Llama 3-8B (Q4K) requires 5.3 GiB of VRAM, so the entire model could fit into D3-GPU. However, exo allocates layers proportional to each device's memory: D1 (8 GiB RAM), D2 (8 GiB VRAM), and D3 (11 GiB VRAM) are assigned 9, 10, 13 layers, respectively. While D1 has an Apple Silicon GPU, its efficiency is much lower than the 2080TI on D3-GPU, making it a bottleneck. Dllama uses TP. An 8B model needs 64 all-reduce operations, leading to heavy communication delays on high-latency Wi-Fi and a much higher TPOT than prima.cpp. As model depth grows with size, the communication bottleneck worsens. At 70B, dllama runs out of memory, but our independent tests show ~150 ms per all-reduce, so 160 all-reduces add 24 s/token, resulting in a much higher TPOT than prima.cpp's 674 ms/token (442 ms/token with speculative decoding). These results highlight that Halda and PRP are better suited to heterogeneous, high-latency home environments than heuristic partitioning or TP.

More experiments on heterogenous and homogeneous testbeds (Appendices A.4, A.5), on more models (Qwen 2.5, QwQ, and DeepSeek R1) (Appendix A.6), and on a larger heterogenous testbed (Appendix A.19) support the strong generalization of prima.cpp. Appendix A.7 explains the underlying rationale for device selection: more is not always better, and aggregated memory need not match the model's needs. It also shows how Halda selects a subset of devices to build a best-performing cluster. Appendix A.9 shows that with speculative decoding, 32B inference achieves 26 tokens/s, offering both the speed and intelligence needed for LLM agents.

## 4.2 Ablation Study on Prefetching, Halda, and Pipelined-Ring Parallelism

Table 4 also presents ablations on Halda and prefetching. Exo is also a PP system, so we migrate exo's layer partitioning (based on memory ratio) to prima.cpp (w/o halda), making it an exo variant. Unlike exo, this variant uses available instead of total memory, adds a broader range of models, adds cross-platform quantization and prefetching, and supports CPU/disk offloading to prevent OOMs.

**Prefetching.** To evaluate prefetching, we compare prima.cpp with and without it. It has little effect on small models that fit in RAM/VRAM, but for larger models, evicted layers cause frequent page faults and reloads. Without prefetching, all reloads are triggered on demand by page faults. With prefetching, upcoming layers are loaded in advance and their latencies overlapped, reducing page fault reloads and lowering TPOT by 9-17%.

**Halda.** A proper layer partitioning is critical for fast inference. As shown in Table 4, prima.cpp with Halda reduces TPOT by up to $31\times$ over prima.cpp (w/o halda). For small models, Halda selects the most powerful device (D3) to run all layers, while exo partitions by memory ratio, assigning more layers to weak devices. For larger models, where disk offloading is unavoidable, Halda balances workloads to reduce I/O pressure or prioritizes devices with strong CPU and fast disk, whereas exo

---

[10]We exclude sparse inference systems, as they do not meet the conditions in Table 1: they require dedicated hardware such as NPUs, support only sparse models, and may degrade output quality.

Table 5: Memory pressure for each device on Llama models.

| Size | llama.cpp | exo | | | dllama | | | | prima.cpp | | | |
|------|-----------|-----|-----|-----|--------|-----|-----|-----|-----------|-----|-----|-----|
| | D3 | D1 | D2 | D3 | D1 | D2 | D3 | D4 | D1 | D2 | D3 | D4 |
| 8B | 2.0% | 20.0% | 51.3% | 42.5% | 17.3% | 17.3% | 19.0% | 13.7% | 5.3% | 5.4% | 2.7% | ≤1.0% |
| 14B | 2.5% | - | - | - | 29.2% | 27.3% | 24.4% | 24.9% | 5.3% | 4.3% | 2.2% | ≤1.0% |
| 30B | 8.0% | - | - | - | - | - | - | - | 3.0% | 5.7% | 2.9% | ≤1.0% |
| 45B | 3.9% | - | - | - | 61.0% | 73.9% | 62.5% | 73.8% | 4.9% | ≤1.0% | 6.0% | ≤1.0% |
| 60B | 5.5% | - | - | - | - | - | - | - | 6.3% | 4.7% | 4.7% | ≤1.0% |
| 65B | 15.6% | - | - | - | OOM | OOM | OOM | OOM | 3.9% | ≤1.0% | ≤1.0% | ≤1.0% |
| 70B | 6.0% | OOM | OOM | OOM | OOM | OOM | OOM | OOM | 4.7% | 4.8% | 4.8% | ≤1.0% |

often overloads slow-disk devices, causing TPOT to spike. Appendix A.10 compares Halda with two heuristic baselines, further showing its effectiveness and novelty.

**PRP with multiple per-token rounds.** To evaluate PRP in isolation, we built a CPU cluster with 4 devices, each with 8 cores, 8 GiB RAM, and an SSD of 2 GB/s. We tested models from 8-72B, and assigned layers evenly across devices. For example, for models with 80 layers, layers were split 20:20:20:20 at $k = 1$, 10:10:10:10 at $k = 2$, and so on. With large models (>60B) and $k = 1$, PRP degrades to PP and provides no benefit due to the prefetch-release conflict. In contrast, with $k \geq 2$, Fig. 2 shows that PRP halves TPOT by resolving the conflict and allowing finer-grained overlap of disk loading. Appendices A.14.2, A.16 draw the same results on a heterogenous testbed. With small models (<45B) and sufficient memory, the model fits in the cluster without disk offloading. Then, increasing $k$ has minimal impact on

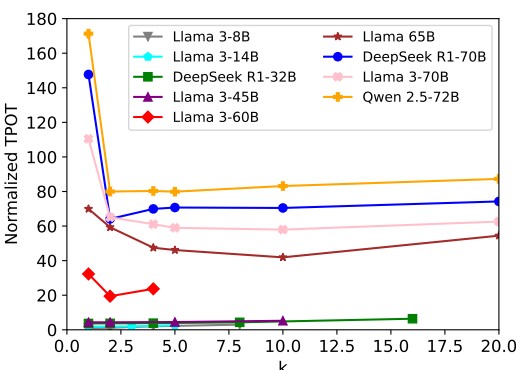

Figure 2: TPOT of PRP under different rounds $k$.

TPOT, aside from minor overhead from additional communication and kernel launches. This shows that PRP is more effective than PP for 70B-scale models.

### 4.3 LOW MEMORY PRESSURE TO PRESERVE USER EXPERIENCE

Memory pressure is critical for user experience, as high pressure can slow apps or crash the device. For example, if a phone runs an LLM service while the user is browsing TikTok, memory competition may cause TikTok to lag or crash, prompting the user to terminate the LLM. An implicit advantage of prima.cpp is that it accounts for this by running at a lower memory priority than other apps: it reduces RAM usage when apps start and uses more when they stop. By giving up RAM to keep other apps responsive, prima.cpp preserves user experience and protects itself from being killed by the user.

We define memory pressure from the LLM as $\Delta$mem_available/mem_total (e.g., 2 GiB used / 8 GiB total = 25%)[11]. Table 5 shows that exo and dllama pin model weights in RAM, raising memory pressure by over 50% on some devices. This can force the OS to reclaim memory from other apps (e.g., compression, swapping), causing lag or crashes, and can still lead to OOM for larger models. The root cause is that they prioritize the LLM at the cost of other apps, which users do not want. In contrast, llama.cpp and prima.cpp exhibit low pressure: both pin only minimal KV caches and compute buffers, while prima.cpp further distributes them across devices. Model weights are loaded via mmap and cached in pages that can be released instantly for other apps. These properties make prima.cpp a practical choice for consumer devices, where LLMs must run alongside user apps.

---

[11]mem_available includes free and reclaimable pages, so its reduction reflects only non-reclaimable pages, i.e., true memory stress. Appendix A.8 shows each device's memory footprint, but not memory pressure, since reclaimable memory (e.g., page cache) is included and can be freed instantly.

## 5 CONCLUSION

This paper proposes prima.cpp, the first on-device distributed system to deliver practical speed for 30-70B LLMs on consumer-grade home devices, achieving 26 tokens/s for 32B models and 2 tokens/s for 70B models with speculative decoding. We introduce PRP to run models exceeding the memory limit and resolve the prefetch-release conflict to reactivate prefetching. To handle device heterogeneity and reduce bubbles in PRP, we develop the LDA model, which captures the heterogeneity in computing, communication, memory, and OS-specific reclamation, disk optimizations. We solve it via Halda, which performs smart layer partitioning for minimal TPOT and automatically selects the best-performing devices. The system is meticulously designed, and this level of detail surpasses prior work that often relies on simplified heuristics. It offers features desired by home users: no extra hardware cost, no specialized hardware, privacy, offline, no queueing or timeouts, models beyond memory limit, practical intelligence and speed, low memory pressure, cross-platform, heterogeneity awareness, automatic workload allocation and device selection, and Wi-Fi ready. These make prima.cpp a compelling choice for home deployment and drive broader adoption of embodied AI at home.

### REPRODUCIBILITY STATEMENT

We release the full prima.cpp implementation with detailed documentation to help readers reproduce the results on their own devices. The repository includes step-by-step instructions and pinned dependencies. A Docker image is also provided for quick validation. We document hardware and software requirements, expected runtimes, and common pitfalls. These artifacts enable faithful, independent reproduction of our results with prima.cpp.

### ACKNOWLEDGEMENT

We thank Robert Wen from Pinclr Co. for his technical and device support for this study.

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

# A APPENDIX

## A.1 PREFETCH-RELEASE CONFLICT

As illustrated in Fig. 3, before computation starts, the OS prefetches 3 model layers to the available memory limit. However, it does not stop and continues to load the 4th layer, causing the 1st layer to be released. This prefetch-release cycle repeats, so by the end, the last 3 layers are in memory, while the first 3 are not. Then, when computation begins, the 1st layer, which is not in memory, triggers a page fault, prompting the OS to reload it and the 4th layer to be released. Finally, all layers are loaded twice, incurring unnecessary disk I/O without any benefit from prefetching.

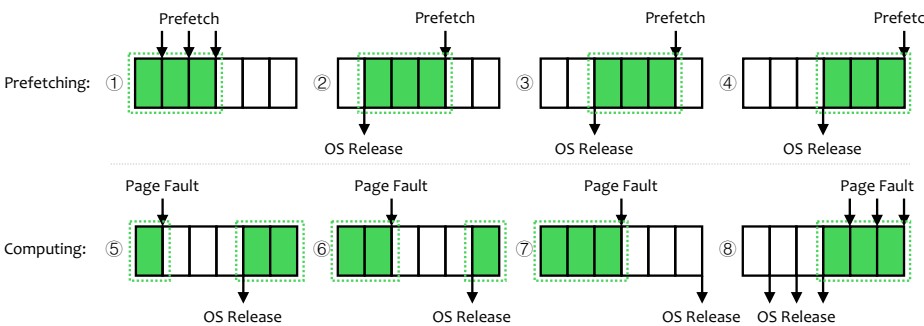

Figure 3: Illustration of model layers loaded into memory in PP with prefetching. In this case, the device handles 6 model layers, but its available memory can only hold 3. The green blocks show the layers loaded into memory, while white blocks indicate those not yet loaded.

## A.2 HOW PIPELINED-RING PARALLELISM SOLVES THE PREFETCH-RELEASE CONFLICT

Fig. 4 illustrates a fast-disk device where prefetching is fast enough to complete before computation begins. In this case, with a fast disk and a layer window size of 2, ① prefetching is fast enough to load 2 layers before computation begins, then ② computation runs without page faults. Then, ③ the next round of 2 layers is prefetched, replacing the used layers. Steps ②-⑦ repeat until inference is complete. Prefetching overlaps with other devices' operations, so its latency does not contribute to TPOT. Here, with no page faults, TPOT comes only from computation. In other words, disk loading latency is fully overlapped.

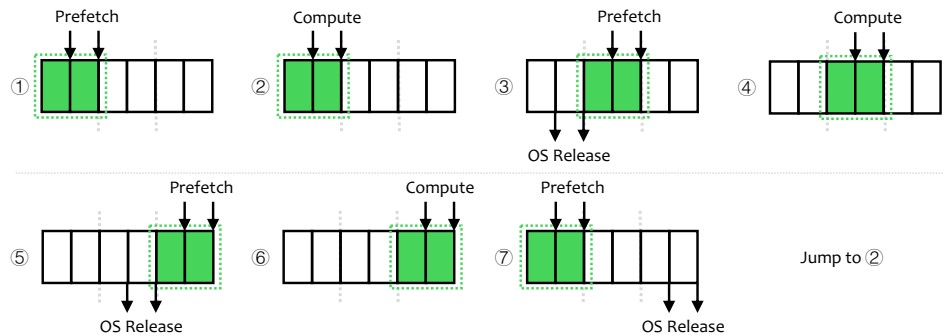

Figure 4: Illustration of model layers loaded into memory in PRP with a fast disk.

Fig. 5 shows a common case with a slow disk. In this case, ① prefetching loads only one layer, then ② computation begins, ③ a page fault is triggered upon reaching the 2nd layer, blocking until it loads. After computation, ④ the device prefetches the next round of 2 layers, but only one layer loads due to the slow disk, and the OS releases the oldest layer. Then, ⑤ the next round of computation begins, and ⑥ at the 6th layer, another page fault occurs. This cycle of "loading (prefetch) - computing -

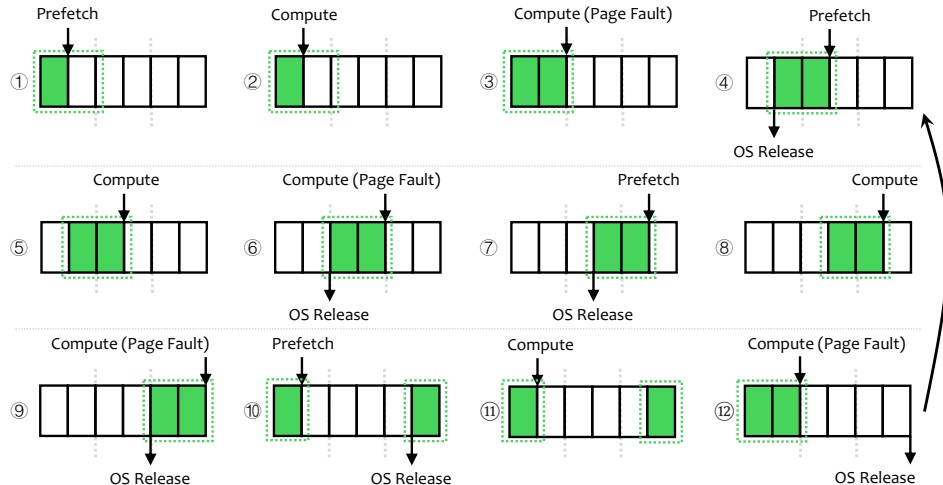

Figure 5: Illustration of model layers loaded into memory in PRP with a slow disk.

loading (page fault) - computing" repeats until inference completes. While page fault-induced loading blocks computation, prefetching helps overlap some latency.

We use a timeline to visualize this overlap. In Fig. 6, green blocks show prefetching that is overlapped, and orange blocks show page fault-induced loading that is not overlapped. In Fig. 6a, with a fast disk, disk loading is fully overlapped. In Figs. 6b and 6c, with a device that has a slow disk, only part of the disk loading is overlapped, while other devices are fully overlapped. In Figs. 6c, 6d, and 6e, although disk loading is not fully hidden, PRP significantly reduces TPOT compared with vanilla PP. In Fig. 6e, while prefetching is used, it exceeds memory limits and triggers prefetch-release, where the OS releases earlier prefetched layers as new ones are loaded, adding disk I/O cost without benefit. This underscores the need to combine PRP with prefetching for higher efficiency.

### A.3 Layer-to-Device Assignment: From Latency Analysis to Vectorized Model

Assume there are $M$ devices, where the layer window size for device $d_m$ is $w_m$. On device $d_m$, the number of GPU layers $n_m$ is defined as follows: within a layer window of size $w_m$, $n_m$ layers run on the GPU, while the remaining $w_m - n_m$ layers run on the CPU ($w_m$ and $n_m$ can vary across devices). Our objective is to find a vector $\boldsymbol{w} = \{w_1, \cdots, w_M\}$ and a vector $\boldsymbol{n} = \{n_1, \cdots, n_M\}$ to minimize the TPOT $T$, which is the sum of latencies from computation $T_m^{\text{comp}}$, memory access $T_m^{\text{mem}}$, disk loading $T_m^{\text{disk}}$, and communication $T_m^{\text{comm}}$ on each device.

$$T = \sum_{m=1}^{M} \left( T_m^{\text{comp}} + T_m^{\text{mem}} + T_m^{\text{disk}} + T_m^{\text{comm}} \right). \tag{11}$$

Here, we minimize $T = \sum_{m=1}^{M} T_m$ instead of $T = \max\{T_m\}$, because Fig. 6f is an idealized illustration. In practice, the OS does not start prefetching immediately after computation ends, and the timing is unknown. As a result, device 4 may experience more bubbles and higher page fault-induced latency than expected. This uncertainty prevents us from solving $T = \max\{T_m\}$ before deployment (which is also hard to measure), and historical data is useless due to fluctuating device conditions. Thus, we take a worst-case approach, assuming the OS has not started prefetching when computation begins, leading to our objective $T = \sum_{m=1}^{M} T_m$. Next, we analyze these latencies in detail.

**Estimation of computation latency** $T_m^{\text{comp}}$. The computation latency on device $d_m$ is defined as the time taken to process $l_m$ model layers and the output layer (if $d_m$ is the head device), where $l_m^{\text{gpu}}$ layers run on the GPU, and the remaining $l_m - l_m^{\text{gpu}}$ layers and output layer run on the CPU. Here, we have $l_m = \lfloor \frac{L}{W} \rfloor w_m + \min(w_m, \max(0, R - \sum_{j=1}^{m-1} \min(w_j, R)))$, $l_m^{\text{gpu}} = \lfloor \frac{L}{W} \rfloor n_m + \min(n_m, \max(0, R - \sum_{j=1}^{m-1} \min(w_j, R)))$, where $n_m \leq w_m$, $W = \sum_{m=1}^{M} w_m$, and

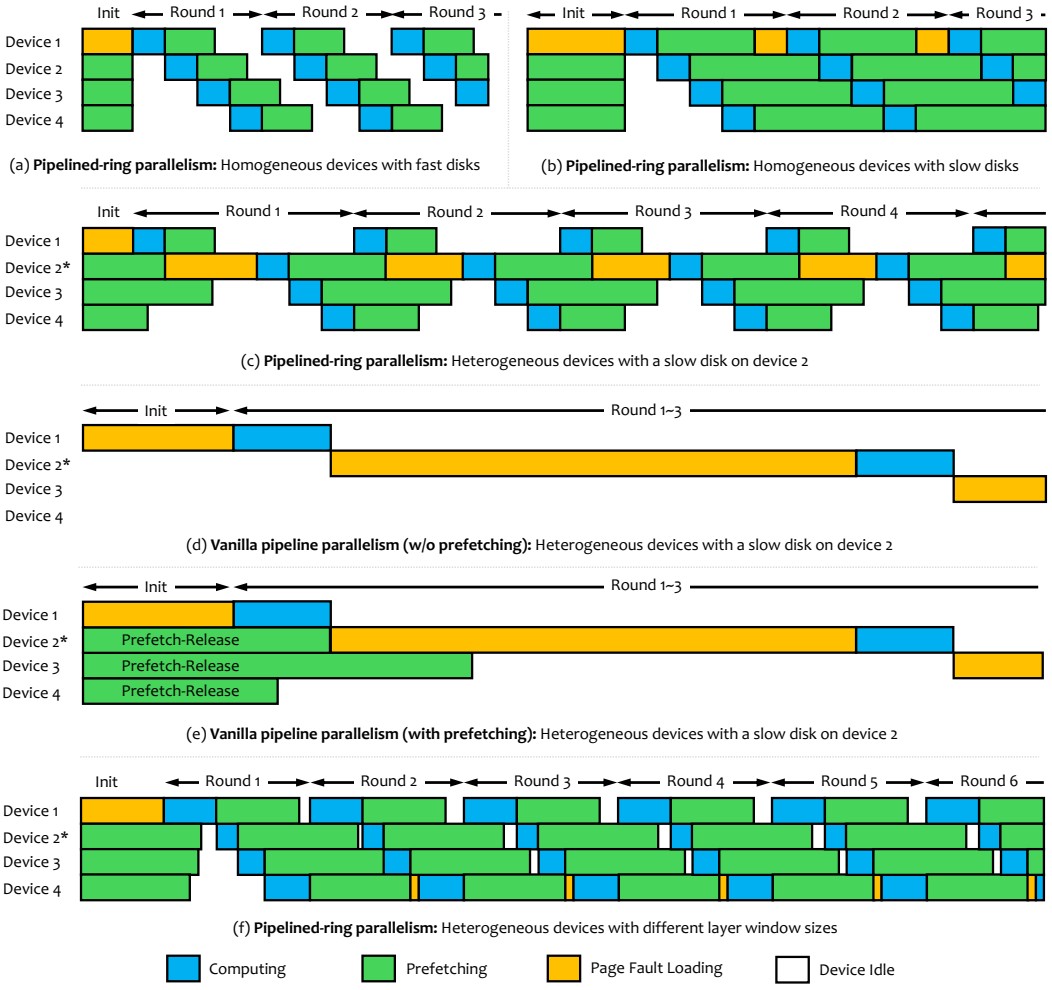

Figure 6: Timeline of (a,b) PRP on homogeneous devices with fast/slow disks; (c,e) PRP on heterogeneous devices with the same/different window sizes; and (d,e) vanilla PP on heterogeneous devices with/without prefetching.

$R = L \mod W$. Since the input layer uses a lookup table, it does not contribute to computation latency.

To estimate the computation time, we develop a model profiler to count the floating-point operations (FLOPs) for each model layer and a device profiler to measure the floating-point throughput (FLOPS) of each device. Taking Q4K as an example, the model weights are primarily quantized in the Q4K format, though some weights use other formats. Specifically, we consider $\mathcal{Q} = \{$Q4_K, Q5_K, Q6_K, Q8_0, FP16, FP32$\}$ and three types of backends: CPU, CUDA, and Metal. The FLOPs for each layer $\mathcal{F}_m$ and output layer $\mathcal{F}_m^{\mathrm{out}}$ consist of 6 values: $\mathcal{F}_m = \{f_m^{q4k}, f_m^{q5k}, f_m^{q6k}, f_m^{q80}, f_m^{fp16}, f_m^{fp32}\}$, $\mathcal{F}_m^{\mathrm{out}} = \{f_{m,\mathrm{out}}^{q4k}, f_{m,\mathrm{out}}^{q5k}, f_{m,\mathrm{out}}^{q6k}, f_{m,\mathrm{out}}^{q80}, f_{m,\mathrm{out}}^{fp16}, f_{m,\mathrm{out}}^{fp32}\}$, each representing the FLOPs under a specific quantization format. The FLOPS $\mathcal{S}_m$ consists of 3 sets: $\{\mathcal{S}_m^{\mathrm{cpu}}, \mathcal{S}_m^{\mathrm{cuda}}, \mathcal{S}_m^{\mathrm{metal}}\}$, with each set consisting of 6 values (e.g., for CPU, $\mathcal{S}_m^{\mathrm{cpu}} = \{s_m^{\mathrm{cpu},q4k}, s_m^{\mathrm{cpu},q5k}, s_m^{\mathrm{cpu},q6k}, s_m^{\mathrm{cpu},q80}, s_m^{\mathrm{cpu},fp16}, s_m^{\mathrm{cpu},fp32}\}$) representing the floating-point throughput for a specific backend-quantization pair. $\mathcal{F}_m$, $\mathcal{F}_m^{\mathrm{out}}$, and $\mathcal{S}_m$ can be easily extended for more backends and quantization formats.

With these profilers, we can estimate the computation time as follows:

$$T_m^{\mathrm{comp}} = (l_m - l_m^{\mathrm{gpu}}) \sum_{q \in \mathcal{Q}} \frac{f_m^q}{s_m^{\mathrm{cpu},q}} + l_m^{\mathrm{gpu}} \sum_{q \in \mathcal{Q}} \frac{f_m^q}{s_m^{\mathrm{gpu},q}} + \mathbb{I}_{m=1} \cdot \sum_{q \in \mathcal{Q}} \frac{f_{m,\mathrm{out}}^q}{s_m^{\mathrm{cpu},q}}. \tag{12}$$

Here, $s_m^{\text{gpu},q}$ refers to GPU FLOPS. If prima.cpp is compiled with CUDA support, $s_m^{\text{gpu},q}$ corresponds to $s_m^{\text{cuda},q}$. If it runs on an Apple device with Metal enabled, $s_m^{\text{gpu},q}$ corresponds to $s_m^{\text{metal},q}$. In our implementation, the output layer is executed on the CPU by the master node ($m = 1$).

**Estimation of memory access latency** $T_m^{\text{mem}}$. This latency consists of three components: (a) *KV cache copy time* $T_m^{kv\text{-}cpy}$: the time taken to copy the new token's cache to the KV cache storage on device $m$; (b) *device copy time* $T_m^{dev\text{-}cpy}$: the time taken to copy hidden states between the CPU and the GPU (e.g., CUDA and Metal); (c) *device loading time* $T_m^{dev\text{-}load}$: the time taken to load data from RAM or VRAM into the processing cores of the CPU or GPU.

For the KV cache copy time, in each token step, new key and value caches are generated with dimensions $(h_k e_k, 1)$ and $(h_v e_v, 1)$, respectively. Here, $h_k$ and $h_v$ are the number of attention heads for the key and value caches, $e_k$ and $e_v$ are the embedding size per head for the key and value vectors, respectively. Thus, for generating one token, each layer needs to copy $h_k e_k + h_v e_v$ values to the KV cache storage. If values are stored in FP16, each takes 2 bytes, so the total number of bytes to be copied is $2(h_k e_k + h_v e_v)$ bytes. In the device profiler module, we measure the time of copying $2(h_k e_k + h_v e_v)$ bytes on CPU, CUDA, and Metal to obtain $t_m^{\text{kv-cpy,cpu}}$ and $t_m^{\text{kv-cpy,gpu}}$. Then, the KV cache copy time $T_m^{kv\text{-}cpy}$ can be estimated by $(l_m - l_m^{\text{gpu}})t_m^{\text{kv-cpy,cpu}} + l_m^{\text{gpu}} t_m^{\text{kv-cpy,gpu}}$.

For the device copy time, this latency arises when the GPU is enabled, as it involves copying the input from RAM to VRAM and then copying the output from VRAM back to RAM. Both input and output have shape $(e, 1)$, where $e$ is the embedding size. These values are typically stored in the FP32 format. In the device profiler module, we measure the latency for two operations: the time taken to copy $4e$ bytes of data from RAM to VRAM, denoted as $t_m^{\text{ram-vram}}$, and the time taken to copy $4e$ bytes of data from VRAM to RAM, denoted as $t_m^{\text{vram-ram}}$. For a sequence of layers within a window, one RAM-to-VRAM copy and one VRAM-to-RAM copy are needed, so the device copy time for one window is $t_m^{\text{ram-vram}} + t_m^{\text{vram-ram}}$. For device $d_m$, it was assigned $\mathcal{W}_m = \lfloor \frac{L}{W} \rfloor + \min(1, \max(0, R - \sum_{j=1}^{m-1} \min(w_j, R)))$ windows. Thus, the device copy time for $d_m$ is $T_m^{\text{dev-cpy}} = \mathcal{W}_m(t_m^{\text{ram-vram}} + t_m^{\text{vram-ram}})(1 - \mathbb{I}_m^{\text{UMA}})$, where $\mathbb{I}_m^{\text{UMA}} = 1$ indicates that device $d_m$ uses a unified memory architecture (UMA, e.g., Apple M-series) and the CPU and GPU share memory, so no explicit RAM-VRAM copy is needed.

For the device loading time, processing cores must load data from RAM/VRAM into registers before executing instructions, which incurs latency. However, the theoretical RAM/VRAM bandwidth does not determine this latency because applications often fail to fully utilize the bandwidth, and multi-level caching also has a significant influence. To capture these effects, our device profiler implements an operator to read data from RAM/VRAM into registers. By measuring its latency at data volumes similar to the tensor sizes, we obtain practical throughputs $\{\mathcal{T}_m^{\text{cpu}}, \mathcal{T}_m^{\text{cuda}}, \mathcal{T}_m^{\text{metal}}\}$. Next, we calculate the data volume that needs to be loaded into registers during each token step, which typically consists of the weight data and the KV cache. In the model profiler, we record the total bytes of weight data for the input and output layers as $b_i, b_o$, and for each layer as $b$. Additionally, the KV cache size for each layer is $2(h_k e_k + h_v e_v)n_{kv}$, where $n_{kv}$ is the number of tokens for which the cache is stored. Then, the device loading time $T_m^{\text{dev-load}}$ can be expressed as $T_m^{\text{dev-load}} = (\frac{l_m - l_m^{\text{gpu}}}{\mathcal{T}_m^{\text{cpu}}} + \frac{l_m^{\text{gpu}}}{\mathcal{T}_m^{\text{gpu}}})(b + 2(h_k e_k + h_v e_v)n_{kv}) + \frac{b_i/V + b_o}{\mathcal{T}_m^{\text{cpu}}} \cdot \mathbb{I}_{m=1}$, where $\mathcal{T}_m^{\text{gpu}}$ depends on the hardware: it equals $\mathcal{T}_m^{\text{metal}}$ for Metal and $\mathcal{T}_m^{\text{cuda}}$ for CUDA, and $V$ is the vocabulary size.

Now we can combine the three latency components and give the formal definition of the memory access latency $T_m^{\text{mem}}$:

$$T_m^{\text{mem}} = (l_m - l_m^{\text{gpu}})t_m^{\text{kv-cpy,cpu}} + l_m^{\text{gpu}} t_m^{\text{kv-cpy,gpu}} + \mathcal{W}_m(t_m^{\text{ram-vram}} + t_m^{\text{vram-ram}})(1 - \mathbb{I}_m^{\text{UMA}}) \quad (13)$$

$$+ (\frac{l_m - l_m^{\text{gpu}}}{\mathcal{T}_m^{\text{cpu}}} + \frac{l_m^{\text{gpu}}}{\mathcal{T}_m^{\text{gpu}}})(b + 2(h_k e_k + h_v e_v)n_{kv}) + \frac{b_i/V + b_o}{\mathcal{T}_m^{\text{cpu}}} \cdot \mathbb{I}_{m=1}. \quad (14)$$

**Estimation of disk loading latency** $T_m^{\text{disk}}$. Prima.cpp is designed to run on memory-constrained devices, so it cannot load the entire model into RAM. To address this, prima.cpp uses mmap to manage model weights. By using mmap, model weights are loaded into memory from disk only when needed for computation, and the OS will release inactive memory-mapped (mmapped) pages when memory pressure is high. This prevents OOM risks but incurs significant disk I/O latency, because evicted pages must be reloaded from disk the next time they are needed. To estimate this disk loading latency, it is necessary to determine the data volume that mmap needs to reload in each token

step. This is a challenging task because different OSs exhibit very different memory management behaviors and high variability.

On macOS (without Metal) and Linux, the OS gradually reclaims memory. When memory pressure is moderate, i.e., when $b_{lio} + 2(h_k e_k + h_v e_v)n_{kv}(l_m - l_m^{\text{gpu}}) + c^{\text{cpu}} > d_m^{\text{avail}}$, mmapped pages are released incrementally until the pressure is alleviated. As a result, some weight data remain in the page cache, and the amount of data that mmap needs to reload is $\max(b_{lio} + 2(h_k e_k + h_v e_v)n_{kv}(l_m - l_m^{\text{gpu}}) + c^{\text{cpu}} - d_m^{\text{avail}}, 0)$, where $b_{lio} = (l_m - l_m^{\text{gpu}})b + (b_i/V + b_o) \cdot \mathbb{I}_{m=1}$, $2(h_k e_k + h_v e_v)n_{kv}(l_m - l_m^{\text{gpu}})$ is the KV cache size, and $c^{\text{cpu}}$ is the compute buffer size. If CUDA is enabled on Linux, the model weights in private VRAM are locked by the CUDA driver, keeping them resident so no disk I/O occurs. Therefore, the disk loading latency for macOS (without Metal) and Linux can be estimated as

$$T^{\text{disk}}_{m,\text{macOS(no Metal)}} = T^{\text{disk}}_{m,\text{Linux}} = \frac{\max\left(b_{lio} + 2(h_k e_k + h_v e_v)n_{kv}(l_m - l_m^{\text{gpu}}) + c^{\text{cpu}} - d_m^{\text{avail}}, b_i/V\right)}{s_m^{\text{disk}}}.$$
(15)

For $T^{\text{disk}}_{m,\text{macOS(no Metal)}}$, $l_m^{\text{gpu}} = 0$, and $s_m^{\text{disk}}$ is the random-read throughput of the disk. On Linux, mmap is configured for sequential access, so $s_m^{\text{disk}}$ is now the sequential-read throughput of the disk.

When Metal is enabled on macOS, the behavior changes. Metal loads mmapped pages into shared memory, and the OS prioritizes retaining these pages. That is, the OS is more inclined to swap out or compress active pages while keeping mmapped model weight pages in shared memory intact. However, when memory is exhausted (with free and inactive pages exhausted, the compression pool nearing saturation, and heavy swap usage), macOS will release these mmapped pages more aggressively. This may cause the entire model weights to be repeatedly reloaded and released. As a result, when the required memory exceeds the total available memory, i.e., when $l_m b + (b_i/V + b_o) \cdot \mathbb{I}_{m=1} + 2(h_k e_k + h_v e_v)n_{kv}l_m + c^{\text{cpu}} + c^{\text{gpu}} > d_{m,\text{metal}}^{\text{avail}}$, device $d_m$ needs to reload $l_m b + (b_i/V + b_o) \cdot \mathbb{I}_{m=1}$ bytes in each token step. Here, $d_{m,\text{metal}}^{\text{avail}}$ denotes the maximum working set size recommended by Metal. By measuring the random-read throughput $s_m^{\text{disk}}$ of the disk, we can calculate the disk loading latency for macOS (with Metal) as:

$$T^{\text{disk}}_{m,\text{macOS (with Metal)}} = \max\Bigg(\frac{l_m b + (b_i/V + b_o) \cdot \mathbb{I}_{m=1}}{s_m^{\text{disk}}} \cdot \mathbb{I}\Big(l_m b + (b_i/V + b_o) \cdot \mathbb{I}_{m=1}$$
$$+ 2(h_k e_k + h_v e_v)n_{kv}l_m + c^{\text{cpu}} + c^{\text{gpu}} - d_{m,\text{metal}}^{\text{avail}}\Big), \frac{b_i}{s_m^{\text{disk}}V}\Bigg).$$
(16)

When running on Android devices, the OS prioritizes swapping out inactive pages to disk, such as memory used by background applications, to ensure that the active application runs smoothly. As a result, the available RAM for prima.cpp can be higher than expected because the OS swaps cold pages to disk, freeing some memory. Thus, on Android, the number of bytes that mmap needs to reload is $\max(b_{bio} + 2(h_k e_k + h_v e_v)n_{kv}(l_m - l_m^{\text{gpu}}) + c^{\text{cpu}} - d_m^{\text{avail}} - d_m^{\text{swapout}}, 0)$, where $d_m^{\text{swapout}} = \min(\max(0, b_{bio} + 2(h_k e_k + h_v e_v)n_{kv}(l_m - l_m^{\text{gpu}}) + c^{\text{cpu}} - d_m^{\text{avail}}), \min(d_m^{\text{bytes\_can\_swap}}, d_m^{\text{swap\_avail}}))$ represents the data bytes that are swapped to disk, $d_m^{\text{bytes\_can\_swap}}$ is the data bytes of currently used memory that can be swapped out, and $d_m^{\text{swap\_avail}}$ is the total available swap space. Then we have:

$$T^{\text{disk}}_{m,\text{Android}} = \frac{\max(b_{bio} + 2(h_k e_k + h_v e_v)n_{kv}(l_m - l_m^{\text{gpu}}) + c^{\text{cpu}} - d_m^{\text{avail}} - d_m^{\text{swapout}}, b_i/V)}{s_m^{\text{disk}}}.$$
(17)

By aggregating them, we obtain a unified expression compatible with cross-platform devices:

$$T^{\text{disk}}_m = T^{\text{disk}}_{m,\text{macOS (no Metal)}} \cdot \mathbb{I}_{\text{macOS (no Metal)}} + T^{\text{disk}}_{m,\text{macOS (with Metal)}} \cdot \mathbb{I}_{\text{macOS (with Metal)}}$$
$$+ T^{\text{disk}}_{m,\text{Linux}} \cdot \mathbb{I}_{\text{Linux}} + T^{\text{disk}}_{m,\text{Android}} \cdot \mathbb{I}_{\text{Android}},$$
(18)

where $\mathbb{I}_{\text{macOS (no Metal)}}, \mathbb{I}_{\text{macOS (with Metal)}}, \mathbb{I}_{\text{Linux}}, \mathbb{I}_{\text{Android}}$ are indicator functions. This expression can be easily extended to include new OSs.

**Estimation of network communication latency $T_m^{\text{comm}}$.** In prima.cpp, devices are connected in a ring, and each device receives inputs from its predecessor, processes them, and forwards the outputs to the next device. After a device completes the computation for one layer window, it transmits the result ($e$ values in the FP32 format, totaling $4e$ bytes) to the next device for further computation in the next

layer window. During each token step, the number of network communications on device $d_m$ equals the number of layer windows, which is $\mathcal{W}_m = \lfloor \frac{L}{W} \rfloor + \min(1, \max(0, R - \sum_{j=1}^{m-1} \min(w_j, R)))$. By measuring the latency $t_m^{\text{comm}}$ of transmitting $4e$ bytes between adjacent devices, we can estimate the network communication latency on device $d_m$ as:

$$T_m^{\text{comm}} = \Big( \Big\lfloor \frac{L}{W} \Big\rfloor + \min(1, \max(0, R - \sum_{j=1}^{m-1} \min(w_j, R))) \Big) t_m^{\text{comm}}. \tag{19}$$

By aggregating these latencies, our objective becomes:

$$
\begin{aligned}
T = \sum_{m=1}^{M} \Bigg[ &(l_m - l_m^{\text{gpu}}) \sum_{q \in \mathcal{Q}} \frac{f_m^q}{s_m^{\text{cpu},q}} + l_m^{\text{gpu}} \sum_{q \in \mathcal{Q}} \frac{f_m^q}{s_m^{\text{gpu},q}} + \mathbb{I}_{m=1} \cdot \sum_{q \in \mathcal{Q}} \frac{f_{m,\text{out}}^q}{s_m^{\text{cpu},q}} \\
&+ (l_m - l_m^{\text{gpu}}) t_m^{\text{kv\_cpy,cpu}} + l_m^{\text{gpu}} t_m^{\text{kv\_cpy,gpu}} + \mathcal{W}_m (t_m^{\text{ram-vram}} + t_m^{\text{vram-ram}})(1 - \mathbb{I}_m^{\text{UMA}}) \\
&+ \Big( \frac{l_m - l_m^{\text{gpu}}}{\mathcal{T}_m^{\text{cpu}}} + \frac{l_m^{\text{gpu}}}{\mathcal{T}_m^{\text{gpu}}} \Big)(b + 2(h_k e_k + h_v e_v)n_{kv}) + \frac{b_i/V + b_o}{\mathcal{T}_m^{\text{cpu}}} \cdot \mathbb{I}_{m=1} \\
&+ \frac{\max(l_m b + (b_i/V + b_o) \cdot \mathbb{I}_{m=1} + 2(h_k e_k + h_v e_v)n_{kv}l_m + c^{\text{cpu}} - d_m^{\text{avail}}, b_i/V)}{s_m^{\text{disk}}} \cdot \mathbb{I}_{\text{macOS (no Metal)}} \\
&+ \max \Big( \frac{l_m b + (b_i/V + b_o) \cdot \mathbb{I}_{m=1}}{s_m^{\text{disk}}} \cdot \mathbb{I}\Big( l_m b + (b_i/V + b_o) \cdot \mathbb{I}_{m=1} + 2(h_k e_k + h_v e_v)n_{kv}l_m + c^{\text{cpu}} + c^{\text{gpu}} - d_{m,\text{metal}}^{\text{avail}} \Big), \\
&\qquad \frac{b_i}{s_m^{\text{disk}}V} \Big) \cdot \mathbb{I}_{\text{macOS (with Metal)}} \\
&+ \frac{\max((l_m - l_m^{\text{gpu}})b + (b_i/V + b_o) \cdot \mathbb{I}_{m=1} + 2(h_k e_k + h_v e_v)n_{kv}(l_m - l_m^{\text{gpu}}) + c^{\text{cpu}} - d_m^{\text{avail}}, b_i/V)}{s_m^{\text{disk}}} \cdot \mathbb{I}_{\text{Linux}} \\
&+ \frac{\max((l_m - l_m^{\text{gpu}})b + (b_i/V + b_o) \cdot \mathbb{I}_{m=1} + 2(h_k e_k + h_v e_v)n_{kv}(l_m - l_m^{\text{gpu}}) + c^{\text{cpu}} - d_m^{\text{avail}} - d_m^{\text{swapout}}, b_i/V)}{s_m^{\text{disk}}} \cdot \mathbb{I}_{\text{Android}} \\
&+ \Big( \Big\lfloor \frac{L}{W} \Big\rfloor + \min(1, \max(0, R - \sum_{j=1}^{m-1} \min(w_j, R))) \Big) t_m^{\text{comm}} \Bigg] \\
= \sum_{m=1}^{M} \Bigg[ &\Big( \sum_{q \in \mathcal{Q}} \frac{f_{m,\text{out}}^q}{s_m^{\text{cpu},q}} + \frac{b_i/V + b_o}{\mathcal{T}_m^{\text{cpu}}} \Big) \cdot \mathbb{I}_{m=1} + (l_m - l_m^{\text{gpu}}) \Big( \sum_{q \in \mathcal{Q}} \frac{f_m^q}{s_m^{\text{cpu},q}} + t_m^{\text{kv\_cpy,cpu}} + \frac{b + 2(h_k e_k + h_v e_v)n_{kv}}{\mathcal{T}_m^{\text{cpu}}} \Big) \\
&+ l_m^{\text{gpu}} \Big( \sum_{q \in \mathcal{Q}} \frac{f_m^q}{s_m^{\text{gpu},q}} + t_m^{\text{kv\_cpy,gpu}} + \frac{b + 2(h_k e_k + h_v e_v)n_{kv}}{\mathcal{T}_m^{\text{gpu}}} \Big) + \mathcal{W}_m \Big( (t_m^{\text{ram-vram}} + t_m^{\text{vram-ram}})(1 - \mathbb{I}_m^{\text{UMA}}) + t_m^{\text{comm}} \Big) \\
&+ \frac{\max(l_m b + (b_i/V + b_o) \cdot \mathbb{I}_{m=1} + 2(h_k e_k + h_v e_v)n_{kv}l_m + c^{\text{cpu}} - d_m^{\text{avail}}, b_i/V)}{s_m^{\text{disk}}} \cdot \mathbb{I}_{\text{macOS (no Metal)}} \\
&+ \max \Big( \frac{l_m b + (b_i/V + b_o) \cdot \mathbb{I}_{m=1}}{s_m^{\text{disk}}} \cdot \mathbb{I}\Big( l_m b + (b_i/V + b_o) \cdot \mathbb{I}_{m=1} + 2(h_k e_k + h_v e_v)n_{kv}l_m + c^{\text{cpu}} + c^{\text{gpu}} - d_{m,\text{metal}}^{\text{avail}} \Big), \\
&\qquad \frac{b_i}{s_m^{\text{disk}}V} \Big) \cdot \mathbb{I}_{\text{macOS (with Metal)}} \\
&+ \max \Big( \frac{(l_m - l_m^{\text{gpu}})\big[ b + 2(h_k e_k + h_v e_v)n_{kv} \big]}{s_m^{\text{disk}}} + \frac{(b_i/V + b_o) \cdot \mathbb{I}_{m=1} + c^{\text{cpu}} - d_m^{\text{avail}} - d_m^{\text{swapout}} \cdot \mathbb{I}_{\text{Android}}}{s_m^{\text{disk}}}, \\
&\qquad \frac{b_i}{s_m^{\text{disk}}V} \Big) \cdot (\mathbb{I}_{\text{Linux}} + \mathbb{I}_{\text{Android}}) \Bigg]
\end{aligned}
$$

To remove the max operator, we decompose the disk loading latency into multiple terms, separately accounting for whether memory is sufficient or not. Let $\mathcal{M}$ be the set of all devices, and $\mathcal{M}_1, \mathcal{M}_2, \mathcal{M}_3, \mathcal{M}_4$ be the subsets of devices that satisfy the respective conditions in Cases 1-4, where $\mathcal{M}_1 \cap \mathcal{M}_2 \cap \mathcal{M}_3 \cap \mathcal{M}_4 = \emptyset$ and $\mathcal{M}_1 \cup \mathcal{M}_2 \cup \mathcal{M}_3 \cup \mathcal{M}_4 = \mathcal{M}$.

*Case 1 (macOS with Metal disabled and insufficient RAM):* If $l_m b + (b_i/V + b_o) \cdot \mathbb{I}_{m=1} + 2(h_k e_k + h_v e_v) n_{kv} l_m + c^{\text{cpu}} > d_m^{\text{avail}}$ and $s_m^{\text{disk}} > s_{\text{threshold}}^{\text{disk}}$, then $T_m^{\text{disk}} = \frac{l_m b + (b_i/V + b_o) \cdot \mathbb{I}_{m=1} + 2(h_k e_k + h_v e_v) n_{kv} l_m + c^{\text{cpu}} - d_m^{\text{avail}}}{s_m^{\text{disk}}}, m \in \mathcal{M}_1$.

*Case 2 (macOS with Metal enabled and insufficient RAM):* If $l_m b + (b_i/V + b_o) \cdot \mathbb{I}_{m=1} + 2(h_k e_k + h_v e_v) n_{kv} l_m + c^{\text{cpu}} + c^{\text{gpu}} > d_{m,\text{metal}}^{\text{avail}}$ and $s_m^{\text{disk}} > s_{\text{threshold}}^{\text{disk}}$, then $T_m^{\text{disk}} = \frac{l_m b + (b_i/V + b_o) \cdot \mathbb{I}_{m=1}}{s_m^{\text{disk}}}, m \in \mathcal{M}_2$.

*Case 3 (Linux and Android with insufficient RAM):* If $(l_m - l_m^{\text{gpu}})\Big[b + 2(h_k e_k + h_v e_v) n_{kv}\Big] + (b_i/V + b_o) \cdot \mathbb{I}_{m=1} + c^{\text{cpu}} > d_m^{\text{avail}} + d_m^{\text{swapout}} \cdot \mathbb{I}_{\text{Android}}$ and $s_m^{\text{disk}} > s_{\text{threshold}}^{\text{disk}}$, then $T_m^{\text{disk}} = \frac{1}{s_m^{\text{disk}}}\Big((l_m - l_m^{\text{gpu}})[b + 2(h_k e_k + h_v e_v) n_{kv}] + (b_i/V + b_o) \cdot \mathbb{I}_{m=1} + c^{\text{cpu}} - d_m^{\text{avail}} - d_m^{\text{swapout}} \cdot \mathbb{I}_{\text{Android}}\Big), m \in \mathcal{M}_3$.

*Case 4 (OS with sufficient RAM or low disk speed):* In these cases, the physical RAM is large enough to hold the model weights or the disk speed is slow (i.e., $s_m^{\text{disk}} < s_{\text{threshold}}^{\text{disk}}$). As a result, no disk loading is expected, except for the latency incurred during lookup table access, thus $T_m^{\text{disk}} = \frac{b_i}{s_m^{\text{disk}} V}, m \in \mathcal{M}_4$.

With these cases, we can rewrite the objective function as follows:

$$
\begin{aligned}
T = &\sum_{q \in \mathcal{Q}} \frac{f_{1,\text{out}}^q}{s_1^{\text{cpu},q}} + \frac{b_i/V + b_o}{\mathcal{T}_1^{\text{cpu}}} + \frac{b_i/V}{s_1^{\text{disk}}} + \frac{b_o}{s_1^{\text{disk}}} \cdot \mathbb{I}_{1 \notin \mathcal{M}_4} \\
&+ \sum_{m \in \mathcal{M}} \Big[(l_m - l_m^{\text{gpu}})\Big(\sum_{q \in \mathcal{Q}} \frac{f_m^q}{s_m^{\text{cpu},q}} + t_m^{\text{kv\_cpy,cpu}} + \frac{b + 2(h_k e_k + h_v e_v) n_{kv}}{\mathcal{T}_m^{\text{cpu}}}\Big) \\
&+ l_m^{\text{gpu}}\Big(\sum_{q \in \mathcal{Q}} \frac{f_m^q}{s_m^{\text{gpu},q}} + t_m^{\text{kv\_cpy,gpu}} + \frac{b + 2(h_k e_k + h_v e_v) n_{kv}}{\mathcal{T}_m^{\text{gpu}}}\Big) + \mathcal{W}_m\Big((t_m^{\text{ram-vram}} + t_m^{\text{vram-ram}})(1 - \mathbb{I}_m^{\text{UMA}}) + t_m^{\text{comm}}\Big)\Big] \\
&+ \sum_{m \in \mathcal{M}_2} \frac{l_m b}{s_m^{\text{disk}}} + \sum_{m \in \mathcal{M}_1 \cup \mathcal{M}_3} \Big[\frac{(l_m - l_m^{\text{gpu}})[b + 2(h_k e_k + h_v e_v) n_{kv}]}{s_m^{\text{disk}}} + \frac{c^{\text{cpu}} - d_m^{\text{avail}} - d_m^{\text{swapout}} \cdot \mathbb{I}_{\text{Android}}}{s_m^{\text{disk}}}\Big]
\end{aligned}
\tag{20}
$$

To further simplify the objective function, we make the following assumption.

**Assumption 1.** *Let $\frac{L}{W}$ be an integer (i.e., $R = 0$), where $W = \sum_{m \in \mathcal{M}} w_m$. Then all devices are assigned an equal number of windows, and all windows are filled.*

Thus, we have $l_m = \frac{w_m L}{W}$, $l_m^{\text{gpu}} = \frac{n_m L}{W}$, $\mathcal{W}_m = \frac{L}{W}$. Let $b' = b + 2(h_k e_k + h_v e_v) n_{kv}$, $\alpha_m = \sum_{q \in \mathcal{Q}} \frac{f_m^q}{s_m^{\text{cpu},q}} + t_m^{\text{kv\_cpy,cpu}} + \frac{b'}{\mathcal{T}_m^{\text{cpu}}}$, $\beta_m = \sum_{q \in \mathcal{Q}} \frac{f_m^q}{s_m^{\text{gpu},q}} - \sum_{q \in \mathcal{Q}} \frac{f_m^q}{s_m^{\text{cpu},q}} + t_m^{\text{kv\_cpy,gpu}} - t_m^{\text{kv\_cpy,cpu}} + \frac{b'}{\mathcal{T}_m^{\text{gpu}}} - \frac{b'}{\mathcal{T}_m^{\text{cpu}}}$, $\xi_m = (t_m^{\text{ram-vram}} + t_m^{\text{vram-ram}})(1 - \mathbb{I}_m^{\text{UMA}}) + t_m^{\text{comm}}$, $\kappa = \sum_{q \in \mathcal{Q}} \frac{f_{1,\text{out}}^q}{s_1^{\text{cpu},q}} + \frac{b_i/V + b_o}{\mathcal{T}_1^{\text{cpu}}} + \frac{b_i/V}{s_1^{\text{disk}}} + \frac{b_o}{s_1^{\text{disk}}} \cdot \mathbb{I}_{1 \notin \mathcal{M}_4} + \sum_{m \in \mathcal{M}_1 \cup \mathcal{M}_3} \frac{c^{\text{cpu}} - d_m^{\text{avail}} - d_m^{\text{swapout}} \cdot \mathbb{I}_{\text{Android}}}{s_m^{\text{disk}}}$, where $\alpha_m, \beta_m, \xi_m$ are platform-specific constants and $\kappa$ is a global constant. Then, we add the first general term to the three platform-specific terms and obtain:

$$
\begin{aligned}
T = &\frac{L}{W} \sum_{m \in \mathcal{M}_1} \Big[(\alpha_m + \frac{b'}{s_m^{\text{disk}}}) w_m + \xi_m\Big] + \frac{L}{W} \sum_{m \in \mathcal{M}_2} \Big[(\alpha_m + \frac{b}{s_m^{\text{disk}}}) w_m + \beta_m n_m + \xi_m\Big] \\
&+ \frac{L}{W} \sum_{m \in \mathcal{M}_3} \Big[(\alpha_m + \frac{b'}{s_m^{\text{disk}}}) w_m + (\beta_m - \frac{b'}{s_m^{\text{disk}}}) n_m + \xi_m\Big] + \frac{L}{W} \sum_{m \in \mathcal{M}_4} \Big[\alpha_m w_m + \beta_m n_m + \xi_m\Big] + \kappa.
\end{aligned}
$$

This objective is a sum over device sets $\mathcal{M}_1, \mathcal{M}_2, \mathcal{M}_3$. Each summand involves expressions linear in $w_m$ and $n_m$, plus platform-specific constant terms. To clarify the form, we define a linear function $f(a, b, c) = a w_m + b n_m + c$, where the platform-specific constants $a, b, c$ are independent of decision variables $w_m, n_m$. Consequently, the objective can be rearranged to a combination of linear functions:

$$
\begin{aligned}
T = \frac{L}{W}\Big[ &\sum_{m \in \mathcal{M}_1} f(\alpha_m + \frac{b'}{s_m^{\text{disk}}}, 0, \xi_m) + \sum_{m \in \mathcal{M}_2} f(\alpha_m + \frac{b}{s_m^{\text{disk}}}, \beta_m, \xi_m) \\
&+ \sum_{m \in \mathcal{M}_3} f(\alpha_m + \frac{b'}{s_m^{\text{disk}}}, \beta_m - \frac{b'}{s_m^{\text{disk}}}, \xi_m) + \sum_{m \in \mathcal{M}_4} f(\alpha_m, \beta_m, \xi_m)\Big] + \kappa
\end{aligned}
$$

Note that the objective $T$ is nonlinear because $W = \sum_{m \in \mathcal{M}} w_m$ depends on the decision variables, and the term $\frac{1}{W}$ introduces nonlinearity. Now, we put everything together:

$$\min_{w_m, n_m} \quad T \tag{21}$$

$$\text{s.t.} \quad w_m \in \mathbb{Z}_{>0}, n_m \in \mathbb{Z}_{\geq 0}, n_m \leq w_m \leq L, \tag{22}$$

$$L = kW, k \in \mathbb{Z}_{>0}, \tag{23}$$

$$W = \sum_{m \in \mathcal{M}} w_m, \tag{24}$$

$$f(a, b, c) = aw_m + bn_m + c, \tag{25}$$

$$\bigcap_{i=1}^{4} \mathcal{M}_i = \emptyset, \bigcup_{i=1}^{4} \mathcal{M}_i = \mathcal{M}, \tag{26}$$

$$w_m > \frac{W}{Lb'}(d_m^{\text{avail}} - b_m^{cio}), m \in \mathcal{M}_1, \tag{27}$$

$$w_m > \frac{W}{Lb'}(d_{m,\text{metal}}^{\text{avail}} - b_m^{cio} - c^{\text{gpu}}), m \in \mathcal{M}_2, \tag{28}$$

$$w_m - n_m > \frac{W}{Lb'}(d_m^{\text{avail}} + d_m^{\text{swapout}} \cdot \mathbb{I}_{\text{Android}} - b_m^{cio}), m \in \mathcal{M}_3, \tag{29}$$

$$w_m \cdot \mathbb{I}_{\text{macOS (no Metal)}} < \frac{W}{Lb'}(d_m^{\text{avail}} - b_m^{cio}), m \in \mathcal{M}_4, \tag{30}$$

$$w_m \cdot \mathbb{I}_{\text{macOS (with Metal)}} < \frac{W}{Lb'}(d_{m,\text{metal}}^{\text{avail}} - b_m^{cio} - c^{\text{gpu}}), m \in \mathcal{M}_4, \tag{31}$$

$$(w_m - n_m)(\mathbb{I}_{\text{Linux}} + \mathbb{I}_{\text{Android}}) < \frac{W}{Lb'}(d_m^{\text{avail}} + d_m^{\text{swapout}} \cdot \mathbb{I}_{\text{Android}} - b_m^{cio}), m \in \mathcal{M}_4, \tag{32}$$

$$b_m^{cio} = (b_i/V + b_o) \cdot \mathbb{I}_{m=1} + c^{\text{cpu}}, \tag{33}$$

$$n_m \cdot \mathbb{I}_{\text{cuda}} \leq \frac{W}{Lb'}(d_{m,\text{cuda}}^{\text{avail}} - c^{\text{gpu}}) \cdot \mathbb{I}_{\text{cuda}}, \tag{34}$$

$$n_m \cdot \mathbb{I}_{\text{metal}} \leq \frac{W}{Lb'}(d_{m,\text{metal}}^{\text{avail}} - c^{\text{gpu}} - b_o \cdot \mathbb{I}_{m=1}) \cdot \mathbb{I}_{\text{metal}}, \tag{35}$$

$$n_m = 0, \text{if } \mathbb{I}_{\text{cuda}} = 0 \text{ and } \mathbb{I}_{\text{metal}} = 0. \tag{36}$$

Constraint (22) requires that the window size $w_m$ must be a positive integer, the number of GPU layers $n_m$ must be a non-negative integer, and $n_m$ cannot exceed $w_m$. Constraint (23) requires that all devices be assigned an equal number of windows and all windows be filled. Constraints (27-29) ensure that devices categorized into sets $\mathcal{M}_1, \mathcal{M}_2, \mathcal{M}_3$ meet the memory condition outlined in Cases 1-3. Similarly, Constraints (30-32) ensure that devices assigned to set $\mathcal{M}_4$ meet the memory condition outlined in Case 4. $b_m^{cio}$ in Eq. (33) is a platform-independent constant. Constraints (34-35) ensure that the VRAM used by CUDA or the shared memory used by Metal does not exceed the available capacity. Here, $d_{m,\text{cuda}}^{\text{avail}}$ denotes the available GPU private memory for CUDA, and $d_{m,\text{metal}}^{\text{avail}}$ denotes the maximum working set size recommended by Metal.

This is an integer linear fractional program (ILFP) because the numerator is a linear function of the decision variables $w_m, n_w$, and the denominator $W$ is also a linear function of $w_m$. Moreover, the constraints are linear inequalities. The platform indicators $\mathbb{I}_{\text{macOS}}, \mathbb{I}_{\text{Linux}}, \mathbb{I}_{\text{Android}}, \mathbb{I}_{\text{cuda}}, \mathbb{I}_{\text{metal}}, \mathbb{I}_{m=1}$ are known a priori, and they activate/deactivate corresponding linear constraints for each device.

Next, we transform the model into a vectorized form. Let the decision variables be $\boldsymbol{w}^{\text{T}} = [w_1, w_2, \cdots, w_M]$, $\boldsymbol{n}^{\text{T}} = [n_1, n_2, \cdots, n_M]$, and the coefficients $\boldsymbol{a}, \boldsymbol{b}, \boldsymbol{c}$ be:

$$\boldsymbol{a} = \begin{bmatrix} \alpha_m + \frac{b'}{s_m^{\text{disk}}} \mid m \in \mathcal{M}_1 \\ \hline \alpha_m + \frac{b}{s_m^{\text{disk}}} \mid m \in \mathcal{M}_2 \\ \hline \alpha_m + \frac{b'}{s_m^{\text{disk}}} \mid m \in \mathcal{M}_3 \\ \hline \alpha_m \mid m \in \mathcal{M}_4 \end{bmatrix}, \quad \boldsymbol{b} = \begin{bmatrix} 0 \mid m \in \mathcal{M}_1 \\ \hline \beta_m \mid m \in \mathcal{M}_2 \\ \hline \beta_m - \frac{b'}{s_m^{\text{disk}}} \mid m \in \mathcal{M}_3 \\ \hline \beta_m \mid m \in \mathcal{M}_4 \end{bmatrix}, \quad \boldsymbol{c} = \begin{bmatrix} \xi_m \mid m \in \mathcal{M}_1 \\ \hline \xi_m \mid m \in \mathcal{M}_2 \\ \hline \xi_m \mid m \in \mathcal{M}_3 \\ \hline \xi_m \mid m \in \mathcal{M}_4 \end{bmatrix}.$$

To apply constraints to the subset of $\boldsymbol{w}$ and $\boldsymbol{n}$ corresponding to $\mathcal{M}_1, \mathcal{M}_2, \mathcal{M}_3, \mathcal{M}_4$, we define diagonal matrices $\boldsymbol{P}_w = \mathrm{diag}(-\boldsymbol{I}_{\mathcal{M}_1}, -\boldsymbol{I}_{\mathcal{M}_2}, -\boldsymbol{I}_{\mathcal{M}_3}, \boldsymbol{P}^1_{\mathcal{M}_4}, \boldsymbol{P}^2_{\mathcal{M}_4}, \boldsymbol{P}^3_{\mathcal{M}_4})$, $\boldsymbol{P}_n = \mathrm{diag}(\boldsymbol{0}_{\mathcal{M}_1}, \boldsymbol{0}_{\mathcal{M}_2}, \boldsymbol{I}_{\mathcal{M}_3}, \boldsymbol{0}^1_{\mathcal{M}_4}, \boldsymbol{0}^2_{\mathcal{M}_4}, -\boldsymbol{P}^3_{\mathcal{M}_4})$, where $\boldsymbol{I}_{\mathcal{M}_1}, \boldsymbol{I}_{\mathcal{M}_2}, \boldsymbol{I}_{\mathcal{M}_3}$ are identity matrices and $\boldsymbol{0}_{\mathcal{M}_1}, \boldsymbol{0}_{\mathcal{M}_2}, \boldsymbol{0}_{\mathcal{M}_3}$ are zero matrices corresponding to the subsets $\mathcal{M}_1, \mathcal{M}_2, \mathcal{M}_3$, and $\boldsymbol{P}^1_{\mathcal{M}_4}, \boldsymbol{P}^2_{\mathcal{M}_4}, \boldsymbol{P}^3_{\mathcal{M}_4}$ are diagonal binary matricies (i.e., selection matricies) corresponding to the three constraints (30-32) within the subset $\mathcal{M}_4$. To construct $\boldsymbol{P}^1_{\mathcal{M}_4}, \boldsymbol{P}^2_{\mathcal{M}_4}, \boldsymbol{P}^3_{\mathcal{M}_4}$, we define a binary vector $\boldsymbol{p}_{\mathrm{macOS}}$, where a value of 1 indicates that the current device is running on macOS and a value of 0 indicates otherwise. The number of elements in $\boldsymbol{p}_{\mathrm{macOS}}$ matches the number of devices in the set $\mathcal{M}_4$. Similarly, we define binary vectors $\boldsymbol{p}_{\mathrm{Linux}}, \boldsymbol{p}_{\mathrm{Android}}, \boldsymbol{p}_{\mathrm{metal}}$. Thus, we have $\boldsymbol{P}^1_{\mathcal{M}_4} = \mathrm{diag}(\boldsymbol{p}_{\mathrm{macOS}} \odot (1 - \boldsymbol{p}_{\mathrm{metal}})), \boldsymbol{P}^2_{\mathcal{M}_4} = \mathrm{diag}(\boldsymbol{p}_{\mathrm{macOS}} \odot \boldsymbol{p}_{\mathrm{metal}}), \boldsymbol{P}^3_{\mathcal{M}_4} = \boldsymbol{p}_{\mathrm{Linux}} + \boldsymbol{p}_{\mathrm{Android}}$.

To handle constraints (34-35), we define $\boldsymbol{P}^{\mathrm{gpu}}_n$ as a similar diagonal binary matrix, with elements set to one for devices with CUDA or Metal support. Specifically, we let $\boldsymbol{P}^{\mathrm{gpu}}_n = \boldsymbol{P}^{\mathrm{cuda}}_n + \boldsymbol{P}^{\mathrm{metal}}_n$, where $\boldsymbol{P}^{\mathrm{cuda}}_n = \mathrm{diag}(\boldsymbol{0}_{\mathcal{M}_1}, \boldsymbol{0}_{\mathcal{M}_2}, \boldsymbol{P}^{\mathrm{cuda}}_{\mathcal{M}_3}, \boldsymbol{P}^{\mathrm{cuda}}_{\mathcal{M}_4})$ and $\boldsymbol{P}^{\mathrm{metal}}_n = \mathrm{diag}(\boldsymbol{0}_{\mathcal{M}_1}, \boldsymbol{I}_{\mathcal{M}_2}, \boldsymbol{0}_{\mathcal{M}_3}, \boldsymbol{P}^{\mathrm{metal}}_{\mathcal{M}_4})$.

Let the decision variables be $\boldsymbol{w}^{\mathrm{T}}_{\mathcal{M}_4} = [w_m \mid m \in \mathcal{M}_4]$, $\boldsymbol{n}^{\mathrm{T}}_{\mathcal{M}_4} = [n_m \mid m \in \mathcal{M}_4]$, $\boldsymbol{w}'^{\mathrm{T}} = [\boldsymbol{w}^{\mathrm{T}}, \boldsymbol{w}^{\mathrm{T}}_{\mathcal{M}_4}, \boldsymbol{w}^{\mathrm{T}}_{\mathcal{M}_4}]$, $\boldsymbol{n}'^{\mathrm{T}} = [\boldsymbol{n}^{\mathrm{T}}, \boldsymbol{n}^{\mathrm{T}}_{\mathcal{M}_4}, \boldsymbol{n}^{\mathrm{T}}_{\mathcal{M}_4}]$, the RAM upper bound be

$$
\boldsymbol{z} = \frac{1}{Lb'} \begin{bmatrix} d^{\mathrm{avail}}_m - b^{cio}_m \mid m \in \mathcal{M}_1 \\ \hline d^{\mathrm{avail}}_{m,\mathrm{metal}} - b^{cio}_m - c^{\mathrm{gpu}} \mid m \in \mathcal{M}_2 \\ \hline d^{\mathrm{avail}}_m + d^{\mathrm{swapout}}_m \cdot \mathbb{I}_{\mathrm{Android}} - b^{cio}_m \mid m \in \mathcal{M}_3 \\ \hline -d^{\mathrm{avail}}_m + b^{cio}_m \mid m \in \mathcal{M}_4 \\ \hline -d^{\mathrm{avail}}_{m,\mathrm{metal}} + b^{cio}_m + c^{\mathrm{gpu}} \mid m \in \mathcal{M}_4 \\ \hline -d^{\mathrm{avail}}_m - d^{\mathrm{swapout}}_m \cdot \mathbb{I}_{\mathrm{Android}} + b^{cio}_m \mid m \in \mathcal{M}_4 \end{bmatrix},
$$

and the VRAM/shared-memory upper bound be $\boldsymbol{z}^{\mathrm{gpu}} = [z^{\mathrm{gpu}}_1, \cdots, z^{\mathrm{gpu}}_M]$, where

$$
z^{\mathrm{gpu}}_m = \frac{1}{Lb'} \cdot \begin{cases} 0, & \text{if } \mathbb{I}_{\mathrm{cuda}} = 0 \text{ and } \mathbb{I}_{\mathrm{metal}} = 0, \\ d^{\mathrm{avail}}_{m,\mathrm{cuda}} - c^{\mathrm{gpu}}, & \text{if } \mathbb{I}_{\mathrm{cuda}} = 1, \\ d^{\mathrm{avail}}_{m,\mathrm{metal}} - c^{\mathrm{gpu}}, & \text{if } \mathbb{I}_{\mathrm{metal}} = 1 \text{ and } m \neq 1, \\ d^{\mathrm{avail}}_{m,\mathrm{metal}} - c^{\mathrm{gpu}} - b_o, & \text{if } \mathbb{I}_{\mathrm{metal}} = 1 \text{ and } m = 1. \end{cases}
$$

The problem model can then be reformatted as:

$$
\min_{\boldsymbol{w}, \boldsymbol{n}} \quad L \cdot \frac{\boldsymbol{a}^{\mathrm{T}} \cdot \boldsymbol{w} + \boldsymbol{b}^{\mathrm{T}} \cdot \boldsymbol{n} + \boldsymbol{e}^{\mathrm{T}} \cdot \boldsymbol{c}}{\boldsymbol{e}^{\mathrm{T}} \cdot \boldsymbol{w}} + \kappa, \tag{37}
$$

$$
\text{s.t.} \quad w_m \in \mathbb{Z}_{>0}, n_m \in \mathbb{Z}_{\geq 0}, n_m \leq w_m \leq L, \tag{38}
$$

$$
L - k(\boldsymbol{e}^{\mathrm{T}} \cdot \boldsymbol{w}) = 0, k \in \mathbb{Z}_{>0}, \tag{39}
$$

$$
\boldsymbol{P}_w \cdot \boldsymbol{w}' + \boldsymbol{P}_n \cdot \boldsymbol{n}' + \boldsymbol{e}^{\mathrm{T}} \cdot \boldsymbol{w} \cdot \boldsymbol{z} < 0, \tag{40}
$$

$$
-\boldsymbol{P}^{\mathrm{gpu}}_n \cdot \boldsymbol{z}^{\mathrm{gpu}} \cdot \boldsymbol{e}^{\mathrm{T}} \cdot \boldsymbol{w} + \boldsymbol{P}^{\mathrm{gpu}}_n \cdot \boldsymbol{n} \leq 0. \tag{41}
$$

Table 6 summarizes the key symbols used in this paper.

## A.4 GENERALIZATION TO A NEW HETEROGENEOUS TESTBED

To show the generalizability of prima.cpp, we repeated the experiments of Table 4 on another testbed. This testbed includes: a host PC (5 GB RAM, 1080 TI GPU with 11 GB VRAM), a Mac Mini (10 GB UMA RAM), a laptop (23 GB RAM, 3060 GPU with 6 GB VRAM), and a Redmi phone (7 GB RAM). Here, all RAM/VRAM values refer to available memory rather than total capacity. We ran llama.cpp on the laptop (with a 3060 GPU), and exo on the host PC, Mac Mini, and laptop (since exo is not supported on the phone). Meanwhile, dllama and prima.cpp ran on all four devices. Their TPOT

Table 6: Summary of key symbols and their explanations.

| Symbol | Explanation |
|---|---|
| $M$ | Number of devices. |
| $w_m$ | Layer window size on device $d_m$. |
| $n_m$ | Number of GPU layers on device $d_m$. |
| $T$ | The optimization objective (i.e., TPOT). |
| $l_m$ | Total model layers processed by device $d_m$. |
| $l_m^{\text{gpu}}$ | Total GPU layers processed by device $d_m$. |
| $L$ | Total number of model layers. |
| $W$ | Total layer window size across all devices ($W = \sum_{m=1}^{M} w_m$). |
| $h_k, h_v$ | Number of attention heads for keys and values. |
| $e_k, e_v$ | Embedding size per attention head. |
| $e$ | Embedding size. |
| $b, b_i, b_o$ | Bytes of weight tensors per layer, and of input/output tensors. |
| $n_{kv}$ | Number of tokens stored in the KV cache. |
| $V$ | Vocabulary size. |
| $d_m^{\text{avail}}$ | Available memory on device $d_m$. |
| $c^{\text{cpu}}, c^{\text{gpu}}$ | Buffer sizes for CPU/GPU computations. |
| $s_m^{\text{disk}}$ | Disk read throughput for device $d_m$. |
| $s_{\text{threshold}}^{\text{disk}}$ | A threshold for disk read throughput. If the throughput is below this threshold, the disk is considered slow. |
| $\mathcal{M}_1, \mathcal{M}_2, \mathcal{M}_3, \mathcal{M}_4$ | Set assignments, corresponding to cases 1-4. |
| $\boldsymbol{a}, \boldsymbol{b}, \boldsymbol{c}$ | Coefficient vectors for the objective function. |
| $\boldsymbol{P}_w, \boldsymbol{P}_n$ | Diagonal binary selection matrices for constraints on $\boldsymbol{w}$ and $\boldsymbol{n}$. |
| $\boldsymbol{P}_n^{\text{gpu}}$ | Diagonal binary matrix that indicates whether a device uses a GPU. |
| $\boldsymbol{w}', \boldsymbol{n}'$ | Extended vectors for $\boldsymbol{w}$ and $\boldsymbol{n}$. |
| $\boldsymbol{z}, \boldsymbol{z}^{\text{gpu}}$ | Vectors of RAM/VRAM upper bounds for constraints. |

Table 7: TPOT (ms/token) on the new testbed.

| Model (4-bit) | llama.cpp | exo | dllama | **prima.cpp** |
|---|---|---|---|---|
| Llama 3-8B | 27 | OOM | 875 | **27** |
| Llama 3-14B | 199 | - | - | **67** |
| Llama 1-30B | 469 | - | - | **308** |
| Llama 3-45B | 623 | - | - | **328** |
| Llama 3-60B | 12762 | - | - | **671** |
| Llama 1-65B | 20073 | - | - | **703** |
| Llama 3-70B | 23834 | OOM | OOM | **718** |

values are listed in Table 7, following a trend similar to Table 4. In this case, llama.cpp encounters a VRAM bottleneck at 14B, and prima.cpp matches (only at 8B) or achieves the fastest speeds across 8-70B. Additionally, due to the limited RAM of the host PC, exo ran out of memory while loading the weight files. This supplementary experiment supports the generalizability of prima.cpp.

A.5 GENERALIZATION TO A HOMOGENEOUS TESTBED

We also compared llama.cpp, exo, dllama, and prima.cpp on a homogeneous, low-resource cluster. Since we do not have identical home devices, we used Docker to create four identical Linux containers on a server with an RTX 4090. Each container had 8 CPU cores, 8 GiB RAM (4 GiB available), 5 GiB VRAM, and 2 GB/s disk read throughput. Table 8 reports their TPOT results.

Prima.cpp achieves substantial speedups over llama.cpp in this low-resource testbed, delivering more than $100\times$ improvement on the 14B model. Exo, however, crashes even on the 8B model due to its full-precision GPU backend, making it unsuitable for low-end devices. Although exo's TPOT at 8B cannot be measured directly, we can infer its lower bound based on the performance of llama.cpp. For example, llama.cpp runs the full 8B (Q4K) model on 5 GiB of VRAM at 15 ms/token, implying that exo's TPOT would be at least this high. Furthermore, prima.cpp matches the speed of llama.cpp at this scale, suggesting that it is at least as fast as exo. Finally, despite the near-zero communication

Table 8: TPOT (ms/token) on a homogeneous testbed.

| Model (4-bit) | llama.cpp | exo | dllama | **prima.cpp** |
|---|---|---|---|---|
| Llama 3-8B | 15 | OOM | 495 | **15** |
| Llama 3-14B | 2243 | - | 888 | **21** |
| Llama 1-30B | 6870 | - | - | **52** |
| Llama 3-45B | 10563 | - | OOM | **195** |
| Llama 3-60B | 14652 | - | OOM | **391** |
| Llama 1-65B | 15798 | - | - | **502** |
| Llama 3-70B | 17590 | OOM | OOM | **1128** |

delay from the Docker bridge, dllama remains slow, indicating that TP can still be inefficient even on homogeneous, high-bandwidth clusters.

A.6    RUN PRIMA.CPP ON LLAMA 1&3, QWEN 2.5, QWQ AND DEEPSEEK R1

Fig. 7 plots TPOT and TTFT for Llama models as curves. The prima.cpp curves (solid and dashed diamonds) consistently lie at the bottom, confirming that prima.cpp achieves the lowest TPOT and TTFT across 8-70B. Disabling prefetching or Halda increases latency, but the performance drop is substantially larger when Halda is disabled. This suggests that Halda contributes more critically to the speedup than prefetching.

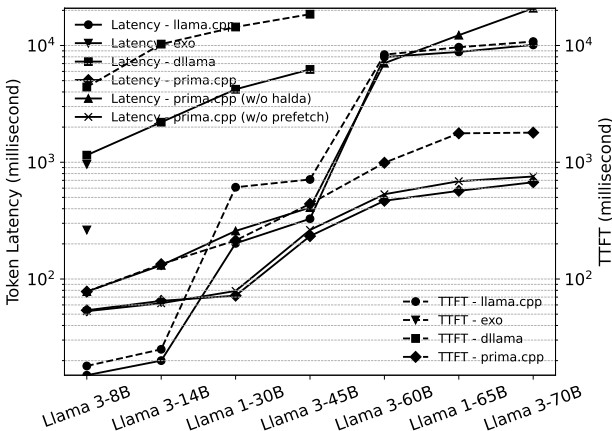

Figure 7: TPOT and TTFT for Llama models across 8-70B.

Table 9: TPOT (ms/token) for Qwen 2.5, QwQ, and DeepSeek R1 models across 7-72B.

| Model (4-bit) | llama.cpp | exo | dllama | prima.cpp |
|---|---|---|---|---|
| Qwen-2.5-7B | 14 | 86 | 1314[2] | **14** |
| DeepSeek-R1-Distill-Qwen-7B | 14 | 68[1] | - | **14** |
| DeepSeek-R1-Distill-Llama-8B | 14 | 77[1] | 1169 | **14** |
| Qwen-2.5-14B | 23 | 31710 | 1565[2] | **23** |
| DeepSeek-R1-Distill-Qwen-14B | 24 | 23475 | - | **24** |
| Qwen-2.5-32B and QwQ-32B | 224 | OOM | 2832[2] | **89** |
| DeepSeek-R1-Distill-Qwen-32B | 232 | OOM | - | **93** |
| DeepSeek-R1-Distill-Llama-70B | 10978 | OOM | OOM | **724** |
| Qwen-2.5-72B | 12227 | OOM | OOM[2] | **867** |

[1] TPOT is lower because exo provides full-precision Qwen models, whereas the DeepSeek-distilled models are quantized to 3-bit.
[2] As dllama doesn't support Qwen-2.5, we use a Qwen-3 model of same size for comparison.

Table 9 extends the evaluation to more models, including Qwen 2.5, QwQ, and DeepSeek R1 (distilled versions) across 7-72B. The results are consistent with Table 4. For small models (<14B), Halda places prima.cpp on D3, so TPOT and TTFT match those of llama.cpp, and 68-94 times faster than dllama. For larger models, Halda distributes the workload across GPUs and CPUs in a smart way, enabling prima.cpp to maintain sub-second TPOT even at 70B, whereas llama.cpp can only offload locally, triggering disk offloading much earlier and causing TPOT to explode.

A special case arises for exo, which only provides an MLX backend for these models. Among the devices in Table 3, D2-D6 failed because Tinygrad only supports Llama-series models and exo can only use D1. For small models, since D1 is less powerful than D3, exo is 6× slower than prima.cpp. At 14B, D1 runs out of memory, and TPOT spikes due to disk swapping. As the model size increases further, exo fails with OOM. This highlights that pinning model weights in memory can induce uncontrolled swapping, whose overhead is often much more severe than proactively managing disk offloading via mmap.

## A.7 SELECT DEVICES TO BUILD THE MOST POWERFUL CLUSTER

Existing systems require clusters with sufficient aggregated RAM/VRAM, often forcing users to add more devices to support larger models. However, assembling enough devices is difficult for households. This raises two questions: *(a) Should we collect enough devices to meet the model's needs? (b) Do more devices always lead to better performance?* The answer is no. Fig. 8 illustrates how the TPOT of prima.cpp on Llama 3-70B (Q4K) is affected by the device set. For each device set, the layer partitioning across device-backend pairs is determined by Halda.

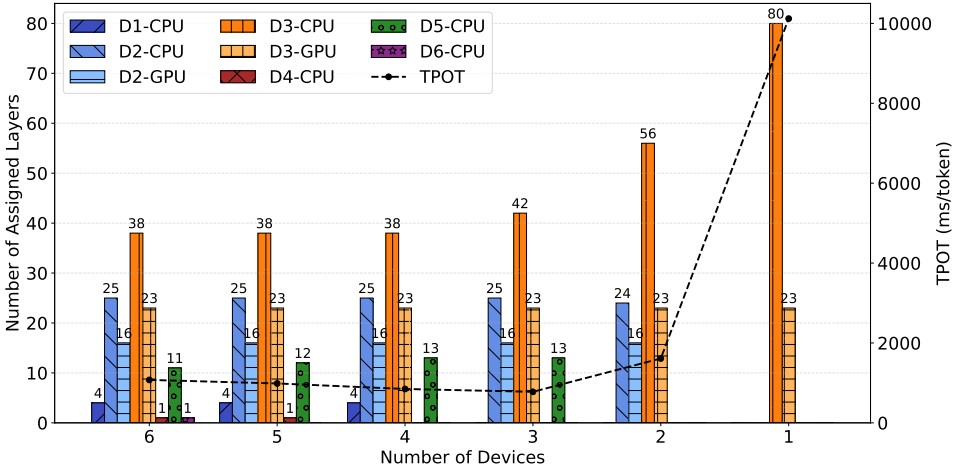

Figure 8: Layer partitioning and TPOT over different device sets.

For question (a), with only 3 devices (D2, D3, and D5), the aggregated RAM+VRAM is 38 GiB, which is insufficient to hold the 40 GiB Llama 3-70B (Q4K) model. However, thanks to the fast SSDs on D2 and D3, mmap can reload model layers quickly, and prima.cpp achieves the lowest latency. Thus, prima.cpp does not require memory sufficient to hold the entire model.

For question (b), when we increase the number of devices to 6, the aggregated RAM+VRAM reaches 50 GiB, which is enough to hold the entire model. However, the TPOT is lower with just 3 devices, because the devices D4 and D6 have weak CPUs and slow disks, creating bottlenecks. This shows that more devices do not always result in faster inference.

This raises a new question: *(c) If the user has a device with a weak CPU or a slow disk, should it be removed from the cluster?* Intuitively, such a weak device would be a bottleneck. However, in cases of severe memory shortage, it does help. For example, D2 and D3 are devices with a GPU, a strong CPU, and a fast disk, while D5 is a weak device. As shown in Fig. 8, adding D5 reduced TPOT by roughly half because D3's disk loading latency (which was heavily overloaded) dominated the impact of D5's weaker CPU.

This raises more questions: if users have some weak devices, which ones should be removed? More generally: *(d) Given a set of heterogeneous devices, how can we select a subset to build the best-performing cluster?* This is challenging due to the uncertain number of devices to be selected, the highly heterogeneous cluster, and the various factors like CPU, GPU, RAM, VRAM, disk, network, and even OS that significantly affect inference speed. Fortunately, Halda offers an easy solution: include all available devices, then remove those with only one assigned layer, since Halda indicates that excluding them would improve speed. This procedure is made automatically in prima.cpp.

### A.8    MEMORY FOOTPRINT ON DEVICE-BACKEND PAIRS

Fig. 9 shows each device's RAM and VRAM usage to illustrate why prima.cpp achieves faster speed and prevents OOM. As exo and dllama don't support Llama 14B-65B and encounter OOM at 70B, the memory usage for 14B-70B in Figs. 9b and 9c is estimated based on system behavior and memory load at 8B. Fig. 9a was discussed in Section 4.1.

Figs. 9b and 9c show that exo and dllama consume high memory. Exo mixes multiple backends, using MLX on macOS for 4-bit computation and Tinygrad on Linux, where model weights are loaded in 16-bit on the CPU and decoded to 32-bit on the GPU. In our case, D1(8 GiB UMA RAM) and D2 (8 GiB VRAM) get the same number of model layers, yet D2-CPU uses $4\times$ more RAM and D2-GPU $8\times$ more VRAM than D1-GPU. This results in high memory usage on Linux devices, increasing the risk of OOM.

For dllama, it uses TP and Q40 quantization to distribute and compress memory usage, but lacks GPU support, so all memory load is on RAM, and inference speed is limited. It has similar memory usage across devices due to its uniform tensor splitting, which causes problems on low-memory devices. In Fig. 9c, for a 30B model, D2 and D3 have more available RAM, while D1 and D4 have less. To allocate enough memory, D1 and D4 must free more active pages or swap out application data, which can slow user apps or even crash the system. In such cases, OOM may be the safer outcome. Additionally, D3 (the head device) loads the entire model before slicing and distributing it, taking significant RAM and making it more prone to OOM.

In contrast, prima.cpp optimizes workload distribution with Halda and prevents OOMs with mmap. Although the solution to the problem Eqs. (1)-(5) is hard to understand, we can observe Halda's preference from Fig. 9d: powerful GPUs > weak GPUs > powerful CPUs > fast disks. For example, at 8B-30B, Halda first fills D2-GPU and D3-GPU. At 45-65B, it fills D1-CPU to D4-CPU. Lastly, the remaining layers are placed on D2-CPU and D3-CPU because they have fast disks. This assignment prevents weak CPUs and slow disks from being used. Finally, only D2-CPU and D3-CPU experience RAM overload, but this does not cause OOM because the OS will free inactive mmapped pages instantly and prefetch model layers in advance. With fast disk reads, disk loading latency stays low, ensuring minimal TPOT, which is exactly the result of our optimization goal (Eq. 1).

Beyond the advanced workload distribution via Halda, prima.cpp also prevents memory waste. With mmap, it loads only the required model layers instead of the full model, eliminating the need for model slicing (which may cause OOMs like in dllama). Additionally, it supports model inference in Q4K format across heterogeneous platforms, eliminating the need to decode back to 16-bit or 32-bit, thereby further reducing RAM/VRAM usage.

### A.9    RUN PRIMA.CPP WITH SPECULATIVE DECODING

Prima.cpp can be further accelerated with speculative decoding. In prima.cpp, we add support for this technique. Since the draft model is small (0.5-3B), we run it as a standalone process on the head device and set the most powerful device as the head. The draft model predicts 5 tokens per step, which are then verified in batch by the larger target model. The testbed consists of 4 Linux devices, each with an 8-core CPU, 8 GiB RAM, and a sequential disk read throughput of 600 MB/s. Two nodes are equipped with a 4090 GPU, but each is limited to 11 GiB VRAM.

Table 10 compares the TPOT of llama.cpp and prima.cpp with and without speculative decoding. With speculative decoding, prima.cpp delivers an additional 25-45% latency reduction across 14-70B models. For example, Qwen 2.5-32B speeds up from 18 to 26 tokens/s, and Llama 3.3-70B from 1.2 to 2.3 tokens/s. At this throughput, a 32B model meets the 20-50 tokens/s throughput commonly required by LLM agents, facilitating broader deployment of frontier LLM agents on home devices.

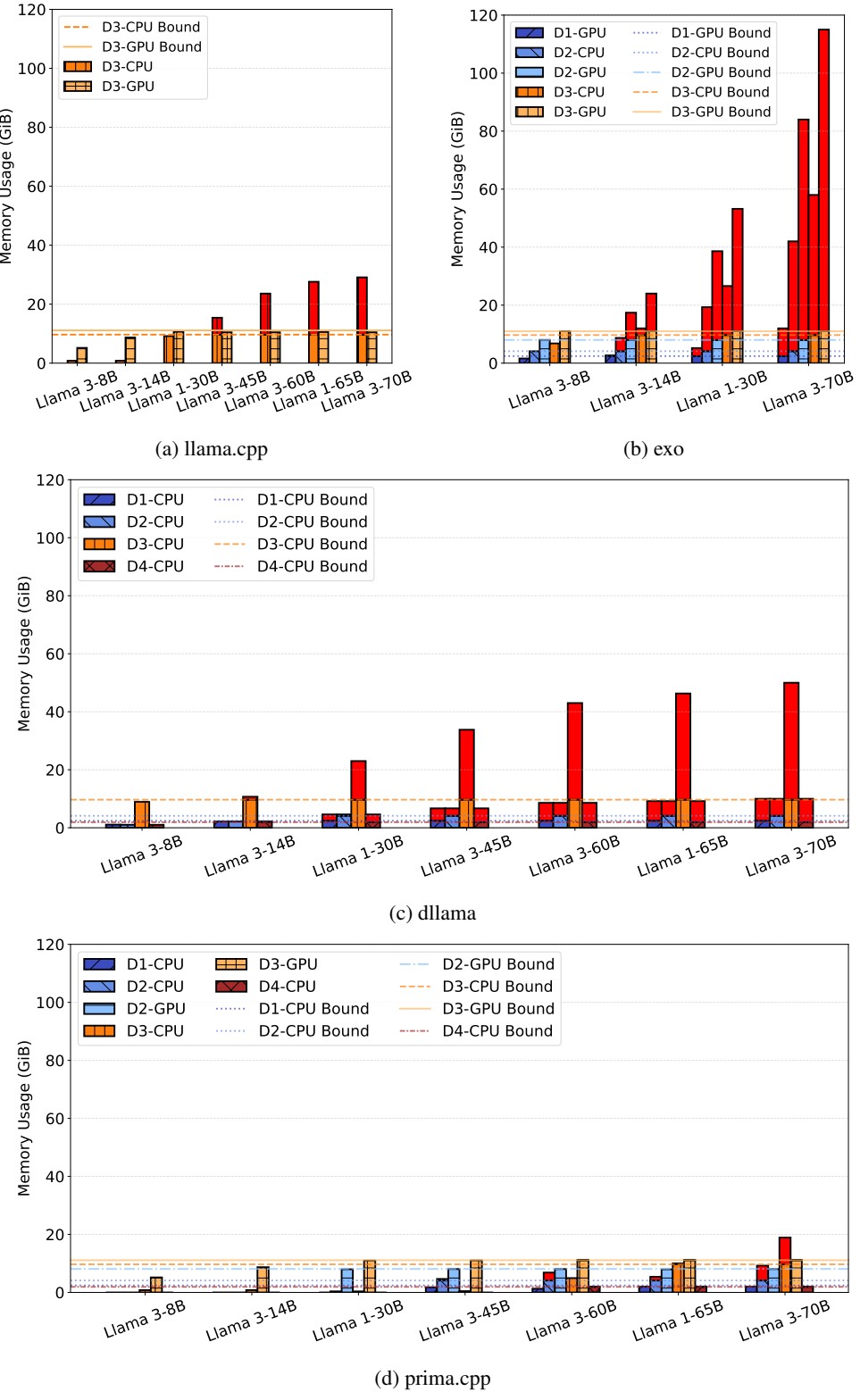

Figure 9: Memory footprint on each device-backend pair.

Table 10: TPOT (ms/token) for llama.cpp and prima.cpp with and without speculative decoding.

| Model (4-bit) | llama.cpp | prima.cpp | **prima.cpp (with speculative)** |
|---|---|---|---|
| Qwen-2.5-7B | $20 \pm 0$ ms | $20 \pm 0$ ms | $18 \pm 1$ ms (draft 0.5B on GPU) |
| Llama 3.2-8B | $20 \pm 0$ ms | $20 \pm 0$ ms | $20 \pm 1$ ms (draft 1B on GPU) |
| Qwen-2.5-14B | $36 \pm 1$ ms | $36 \pm 0$ ms | $27 \pm 1$ ms (draft 1.5B on GPU) |
| DeepSeek-R1-Distill-Qwen-14B | $32 \pm 1$ ms | $32 \pm 0$ ms | $22 \pm 1$ ms (draft 1.5B on GPU) |
| Qwen-2.5-32B | $6551 \pm 38$ ms | $55 \pm 0$ ms | $38 \pm 5$ ms (draft 0.5B on GPU) |
| DeepSeek-R1-Distill-Llama-70B | $20118 \pm 69$ ms | $859 \pm 40$ ms | $593 \pm 65$ ms (draft 8B on CPU) |
| Llama 3.3-70B | $20083 \pm 10$ ms | $803 \pm 9$ ms | $442 \pm 11$ ms (draft 3B on GPU) |
| Qwen-2.5-72B | $21600 \pm 80$ ms | $963 \pm 18$ ms | $544 \pm 19$ ms (draft 3B on CPU) |

### A.10 ABLATION STUDY OF HALDA WITH HEURISTIC SCHEDULER BASELINES

While prior work has studied layer partitioning across heterogeneous devices, these approaches often rely on strong and unrealistic assumptions that limit their use in real home clusters: (a) They only support workload scheduling within GPU clusters; (b) They require the cluster's aggregated VRAM to meet the model's needs; (c) They assume all devices are necessary. For (a) and (b), most households cannot afford an expensive machine with sufficient VRAM, let alone a GPU cluster. For (c), since home devices can vary widely, it could be better to drop weak devices than to keep them.

The novelty in Halda lies in supporting GPU/CPU-mixed clusters while relaxing memory requirements, and in explicitly accounting for disk latency and OS-specific memory behaviors. Moreover, Halda can automatically select the optimal subset of devices from a candidate pool to serve an inference engine that runs at maximum speed. We call this ability "device selection". No previous work has considered these features, but they are necessary in home clusters.

To better demonstrate Halda's effectiveness and novelty, we conduct a supplementary experiment comparing it with two heuristic scheduler baselines:

- *MemSched:* Partitions layers according to RAM/VRAM ratios. This method originates from exo and is also the default strategy in prima.cpp (w/o halda).
- *PerfSched:* This method originates from Galaxy. First, partition layers by devices' compute power, then migrate OOM layers to other devices. This can still hit OOM, so we add a greedy fallback: If OOM persists, offload to CPUs (in proportion to CPU compute power).

The testbed uses a Mac Mini (16GB RAM), a host PC (16GB RAM and 11GB 1080TI GPU), a laptop (32GB RAM and 6GB 3060 GPU), and a Redmi phone (16GB RAM). In this setup, aggregated VRAM is insufficient, but aggregated RAM+VRAM is sufficient, so OOM does not occur.

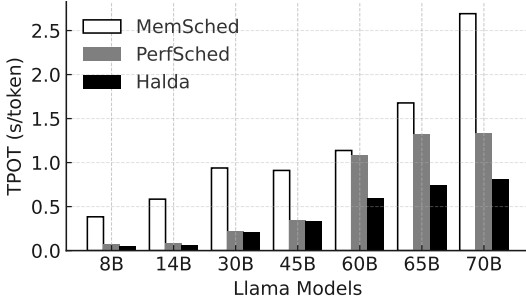

Figure 10: TPOT of MemSched, PerfSched, and Halda on Llama models across 8-70B.

As shown in Fig. 10, on Llama 3-70B (Q4K), Halda runs $3.3\times$ faster than MemSched and $1.6\times$ faster than PerfSched because it learns to drop the 4th device for faster speed. We also tested models of smaller sizes, and Halda always outperforms heuristic schedulers. For 8-45B models, Halda is $1.1$-$1.5\times$ faster than PerfSched, $4.5$-$8.8\times$ faster than MemSched. For 60-70B models, $1.6$-$1.8$x faster than PerfSched, $1.9$-$3.3\times$ faster than MemSched. This demonstrates the effectiveness and novelty of the proposed Halda scheduler.

### A.11 Ablation Study on Multi-Request Workloads

This paper focuses on single-request inference because: (a) home/mobile deployments typically have low concurrency, and (b) continuous batching is not our contribution. However, prima.cpp does implement continuous batching and can handle concurrent requests.

In our implementation, continuous batching merges active requests at the token step. Since PRP also operates on token-step batches, PRP and Halda work unchanged in multi-user/request settings. Each request maintains an independent KV cache. We run Llama-3-14B and 70B models with 1-40 concurrent requests on 2 heterogeneous testbeds:

- *Low-mem testbed:* including a host PC (5 GB RAM, 1080 TI GPU with 11 GB VRAM), a Mac Mini (10 GB UMA RAM), a laptop (23 GB RAM, 3060 GPU with 6 GB VRAM), and a Redmi phone (7 GB RAM);
- *Medium-mem testbed:* including four Linux PCs, one with a 1080 Ti GPU (11 GiB VRAM) and 24 GiB RAM; one with a 2080 Ti (11 GiB) and 32 GiB RAM; one with a 3090 (24 GiB) and 16 GiB RAM; one with a 4060 Ti (8 GiB) and 32 GiB RAM; and a Mac mini with 16 GB of unified memory.

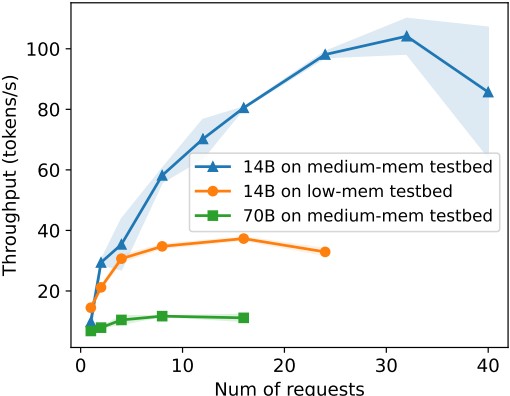

Figure 11: Token throughput over concurrent requests on low-/medium-memory testbeds.

As shown in Fig. 11, when the model is large (70B on the medium-mem testbed) or memory is tight (14B on the low-mem testbed), throughput grows sublinearly, peaks around 8-16 concurrent requests, then declines. This behavior is expected: higher concurrency consumes more memory for KV caches and compute buffers, leading to more frequent model weight eviction and reloads. When resources are more abundant (14B on the medium-mem testbed), throughput scales nearly linearly at first, saturates around 32 concurrent requests, and then drops. This range covers typical household usage, which is often 1-8 concurrent requests.

### A.12 System Design and Implementation

Fig. 12 illustrates the overall framework design of prima.cpp. Our goal is to enable efficient 70B-level LLM inference on heterogeneous, memory-constrained home devices while minimizing per-token latency. In prima.cpp design, we achieve this through:

- **PRP:** A new mmap- and prefetch-based parallelism paradigm that supports low-cost disk offloading to overcome limited memory and resolves prefetch-release conflicts to maximize kernel efficiency.
- **Halda:** An adaptive scheduler that determines the optimal layer partitioning rule according to device heterogeneity.

The system should also support:

- **Multi-request batching:** dynamic batching for concurrent multi-user/multi-request inference.

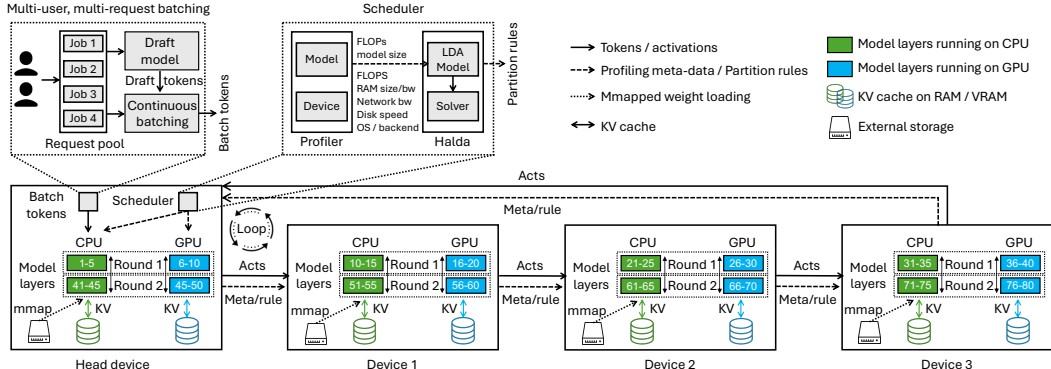

Figure 12: Overall system design of prima.cpp.

- **Speculative decoding:** uses a draft model to transform per-token generation into batched token evaluation, thus improving throughput further.

- **Device join, dropout, and selection:** adds or removes devices at runtime to enable fault recovery and automatic device selection.

- **Cross-LAN connectivity:** enables distributed inference across multiple local networks.

Following the system design illustrated in Fig. 12, we describe how each component is implemented:

**Mmap- and prefetch-based PRP.** PRP executes the model in striped segments, dividing all layers into small chunks that are distributed across devices in a rotating, multi-round schedule. For example, in an 80-layer model, during the first round, devices 0-3 process layers 1-10, 11-20, 21-30, and 31-40, respectively; then in the second round, they process 41-50, 51-60, 61-70, and 71-80, respectively. Within each window, some layers run on the CPU and the rest on the GPU.

During inference, only KV caches, compute buffers, and GPU-resident layer weights remain in RAM or VRAM. CPU-resident weights are lazily loaded via mmap and automatically released when memory becomes scarce, allowing prima.cpp to maintain low memory pressure even for very large models. KV caches are sharded across devices/backends based on layer partitions. Throughout execution, only activations are exchanged between devices or backends, and no migration of KV caches or model weights occurs.

**Profiling and heterogeneity-aware layer partitioning.** To ease description, we divided model layers evenly across devices and backends in Fig. 12, but this is suboptimal in heterogeneous settings. To solve this, we introduce the Halda scheduler, which determines both the size of each device's model segment and the split between CPU and GPU layers, so as to adapt to diverse compute, memory, communication, and I/O speed conditions.

We implement a profiler on each device to profile both model and hardware characteristics, including model FLOPs and size, device FLOPS, RAM capacity and bandwidth, network bandwidth and latency, disk I/O speed, OS, and execution backend. These metadata are propagated along the ring topology and gathered at the head device, where the Halda module will use them to construct the LDA optimization model and solve it using our proposed solver to obtain a layer partitioning rule that satisfies RAM and VRAM constraints while minimizing TPOT. The resulting rules are then broadcast back along the ring topology to all devices. Profiling and Halda scheduling typically occur during system initialization or when the task queue is idle, introducing only 10-12 ms overhead, which is negligible compared to inference time.

**Multi-request batching.** This paper focuses on improving single-request inference, but the proposed PRP and Halda are compatible with batch inference. On the head device, a job queue is maintained, where multiple jobs are batched into micro-batches using continuous batching and then executed by prima.cpp to produce batched outputs. Notably, unlike PP, which may process multiple micro-batches concurrently, PRP handles only one micro-batch at a time to avoid contention for memory and disk across micro-batches.

**Speculative decoding.** We support speculative decoding to further improve throughput. When enabled, a draft model runs to generate draft tokens quickly. Since the draft model is small (0.5-8B), it runs solely on the head device, and the draft tokens are consumed by the target model's input layer on the same device. The larger target model runs on multiple devices using prima.cpp and verifies the draft tokens. Notably, speculative decoding keeps model outputs unchanged and therefore does not impact output quality or perplexity.

**Node join, dropout, and automatic device selection.** Devices communicate through a unidirectional ring throughout the workflow, which provides additional flexibility:

- *Node joining:* a new device can be added by linking it between its predecessor and successor in the communication ring.
- *Node dropout:* when a device leaves, its neighboring devices reconnect directly.

This design enables joining and removal of devices during runtime without disrupting other devices' communication.

Combined with Halda's layer partitioning, prima.cpp can automatically exclude low-load or idle devices (e.g., those assigned 0-1 layers) from the ring, so we can achieve automatic device selection and reduce ring size to further accelerate inference.

**Cross-LAN collaboration.** By adding a proxy device in the ring with zero load, prima.cpp can form a cross-LAN ring topology to involve more devices. For example, if one device can reach another LAN via a VPN connection, it can serve as a proxy, linking the two LANs into a single closed communication ring. This enables prima.cpp to scale inference across multiple local networks.

### A.13 ENERGY CONSUMPTION: FROM CLOUD TO LOCAL & STANDALONE TO DISTRIBUTED

Energy consumption is a key metric for both cloud and edge serving. In this section, we quantify energy per 1K output tokens and break it down into compute kernels, memory/PCIe, disk I/O, communication, and non-zero wait power. On a single device with tight memory, prefetch-release conflicts cause frequent page-fault reloads, stretching I/O waits while hardware still draws non-zero idle power. In prima.cpp, we pool memory across devices and run faster, so reloads and wait energy drop substantially.

To compare the energy consumption of local single-/multi-device inference and cloud inference, we set up a local testbed with 4 mobile devices:

- Mac M1 Pro (8 cores, 8 GiB UMA RAM);
- Mac M3 Pro (12 cores, 18 GiB UMA RAM);
- Linux laptop (8 cores, 4 GiB available RAM, and a 4090 GPU with 16 GiB VRAM);
- Honor Pad (8 cores, 5 GiB available RAM).

For cloud inference, we use a datacenter server with 8 RTX 6000 Ada GPUs. Power consumption was recorded via *PowerMetrics* on Mac devices, *NVIDIA Power Counters* and *powercap* on Linux devices, and *Ludashi* on the Honor Pad. Fig. 13 compares the energy-consumption baseline of running llama.cpp on a single device with the per-device and total energy consumption when using prima.cpp.

**Local single-device vs. prima.cpp distributed.** Across the four local devices, switching from single-device llama.cpp to distributed prima.cpp reduces per-device energy by 91–99% and total energy by 57–90% for 1K tokens. Take the GPU laptop as an example, its average power reduces by 37% (less disk I/O and wait time), time-to-1K-tokens reduces by 86%, hence the total energy reduces by 91%. In the distributed run, the GPU laptop contributes 86% of total energy, while others account for only 14%. This indicates that assigning a few layers to low-power helpers can shorten the critical path at low energy cost.

**Local serving vs. cloud serving.** On the cloud GPU, abundant VRAM eliminates disk/network power and greatly shortens running time, so despite higher average power, the total energy is slightly lower (28%) than a local cluster. However, if we take into account the cooling power consumption of data centers, the energy gap narrows. Regarding energy cost, many countries/regions use tiered

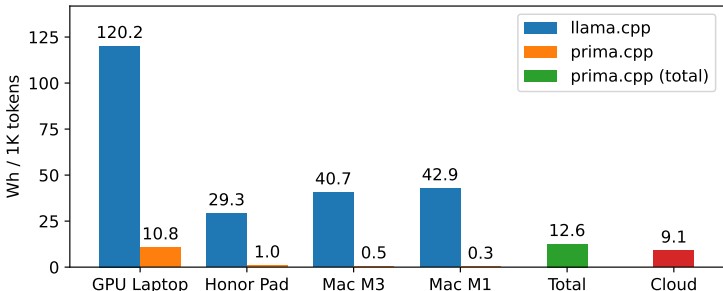

Figure 13: Energy consumption for local single-/multi-device inference vs cloud inference.

electricity pricing: residential prices are about 30% cheaper than commercial prices, which offsets the energy-cost gap between cloud and local serving.

To sum up, prima.cpp significantly reduces energy consumption compared to local single-device serving, while maintaining comparable energy use and cost to cloud serving. If we consider token pricing, bandwidth fees, cloud failure, and cloud costs, local serving with prima.cpp would be more cost-effective.

### A.14 PERFORMANCE BREAKDOWN FOR PREFETCH-RELEASE CONFLICTS, PRP, AND HALDA

In this section, we include a performance breakdown to show the existence and impact of prefetch-release conflicts, as well as the latency composition of disk, compute, communication before and after applying PRP/Halda.

#### A.14.1 AN INTUITIVE LOOK AT THE PREFETCH-RELEASE CONFLICTS

To show that prefetch-release conflicts exist, we should analyze the latency composition of compute time and prefetch time. Compute time refers to the execution time of the computation graph. During kernel execution, the model weights required by a kernel may have already been reclaimed by the OS, triggering page faults and reloading from disk. Thus, measured compute time will include this reloading overhead, which belongs to disk I/O and cannot be separated. In addition, when cache hit rates are low, kernels incur frequent page faults and off-chip memory access penalties, so even if disk I/O time is excluded, kernel efficiency drops.

Once we clarify that compute time includes disk reloads, we can measure both compute time and in-compute reload bytes to observe how reloading affects its time. The following experiment was conducted on an RTX 4090 laptop with 16 GiB VRAM, running Llama-3-70B Q4K, with available memory limited to 32 GiB, 24 GiB, 16 GiB, and 8 GiB. To prove the existence of prefetch-release conflicts, we only need to show that after prefetching, in-compute reload bytes still occur during the compute phase, indicating that some prefetched weights were later reclaimed due to memory shortage and must be reloaded via page faults. The results in Fig. 14 clearly confirm its existence.

Fig. 14 also reveals several key details:

- With sufficient memory (32 GiB RAM): the model fits entirely in the page cache and VRAM. No prefetching is needed, kernels have full cache hits, make compute time minimal.

- At 24 GiB RAM: most model weights remain cached, but about 0.6 GiB of them are reclaimed and reloaded, slowing computation, while disk I/O from prefetching still dominates overall latency.

- At 16 GiB RAM: memory becomes scarce. During prefetching, the last 5 layers loaded evict the first 5 layers. When computation begins, those early layers must be reloaded, which in turn evicts the next few layers (6–10), repeating throughout the process. As a result, although with prefetching, the entire model has to be reloaded during computation. Our results confirm this: even with prefetching completes, the in-compute reload bytes approaches 25 GiB (roughly the total model weights offloaded to CPUs), indicating that

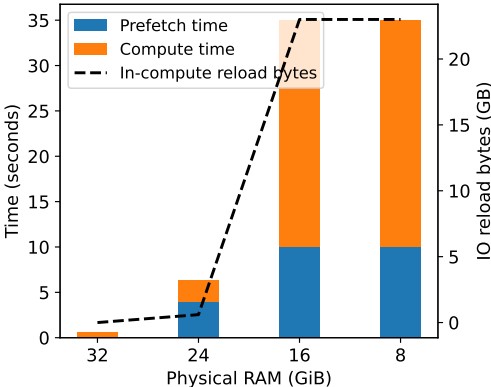

Figure 14: Performance breakdown (per token step) under different RAM capacities and the corresponding in-compute reload bytes.

nearly all prefetched weights were reclaimed before use, which is a clear evidence of prefetch-release conflicts. The sharp drop in cache hit rate causes high-frequent page faults and off-chip memory access, making compute time surge and dominate latency.

- At 8 GiB RAM: since conflicts have reached the worst case at 16 GiB RAM (i.e., almost all model weights must be reclaimed and reloaded), further reducing RAM no longer changes the reload bytes or cache hit rate, so the compute time and prefetch time remains stable.

These results demonstrate that in mmap-based disk offloading, prefetch-release conflicts are real and can severely slow computing, which in turn overwhelms the overlap gains brought by prefetching.

### A.14.2 LATENCY COMPOSITION BEFORE AND AFTER APPLYING PRP

The root cause of prefetch-release conflicts is excessive prefetching under limited RAM: prefetched pages are reclaimed by the OS, and their eviction lowers cache hit rates and triggers frequent reloads, which slow down computation. To address this, prima.cpp introduces the pipelined-ring parallelism (PRP) mechanism, which divides model layers into multiple prefetch-compute rounds to limit the number of model layers prefetched in each round. For example, if a device has enough RAM to hold 5 layers but is assigned 20 layers, PRP sets 4 rounds, prefetching only 5 layers per round. This design alleviates automatic reclamation and improves cache hit rates during kernel execution.

To validate its effectiveness, we build a heterogeneous testbed consisting of four mobile devices:

- D1: Mac M3 Pro (12 cores, 18 GiB UMA RAM);
- D2: Mac M1 Pro (8 cores, 8 GiB UMA RAM);
- D3: Linux laptop (8 cores, 4 GiB available RAM);
- D4: Honor Pad (8 cores, 5 GiB available RAM).

To isolate PRP's effect, Halda is disabled. We measured prefetch, compute, and communication time compositions, as well as in-compute reload sizes and the overlap between prefetching and other devices' runtime. Note that when $k = 1$, PRP degrades to PP. For clarity, Figure 15 reports results on device D3.

From the in-compute reload size curve, PRP clearly reduces the amount of prefetched data reclaimed by the OS. This raises cache hit rates and sharply shortens compute time. The efficiency gain comes from two factors:

- Prefetching loads kernel weights in advance, avoiding frequent disruptions during computing.
- PRP prevents early eviction of required weights under low memory.

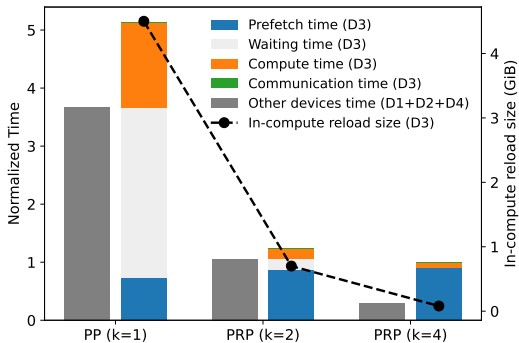

Figure 15: Performance breakdown (per token step) for PP and PRP and the corresponding in-compute reload bytes.

Together, these two factors improve cache hit rate and ensure faster computation, showing that PRP effectively resolves prefetch-release conflicts and improves kernel efficiency over PP.

In addition, since prima.cpp uses disk offloading, PRP also reduces exposed disk I/O more effectively. Comparing PP ($k = 1$) and PRP ($k = 2$), we can see that:

- With PP, other devices can overlap D3's prefetch time, but reload-induced disk I/O during compute remains significant and uncovered.
- With PRP, prefetch time is still fully overlapped, and reload I/O during compute becomes negligible thanks to higher cache hits.

Overall, PRP reduces D3's per-token latency to just 20-23% of that under PP.

### A.14.3   LATENCY COMPOSITION BEFORE AND AFTER APPLYING HALDA

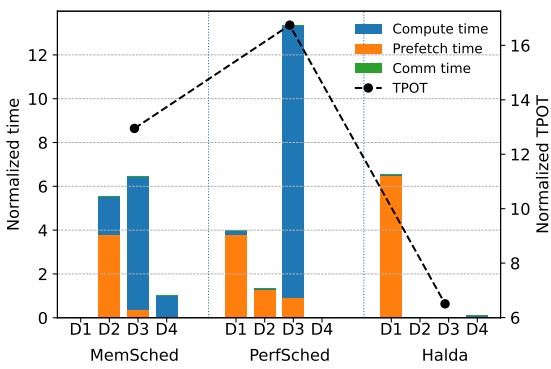

Figure 16: Performance breakdown (per token step) for MemSched, PerfSched, and Halda.

In this experiment, we fix PRP ($k = 2$) and compare Halda with two heuristic layer partitioning rules introduced in Appendix A.10: MemSched and PerfSched. Among them, MemSched follows the same rule as prima.cpp (w/o halda) and thus serves as our Halda-disabled baseline. As shown in Fig. 16, the results of these heuristic rules are highly random:

- *MemSched:* devices with slow CPUs may be assigned many layers; while prefetch time remains small, compute time can become the bottleneck.
- *PerfSched:* devices with small RAM may be assigned many layers, causing frequent kernel disruptions and again making compute time the bottleneck.

In contrast, Halda performs much smarter scheduling. It jointly considers compute capacity, memory size, communication cost, OS/backend heterogeneity, and automatic reclamation behavior, solving an

optimization problem to find the optimal partitioning rule. In this case, Halda assigns most layers to D1, the device with the fastest compute and disk, while slower devices (D2–D4) act as helpers to offload part of D1's memory load. This increases D1's cache hit rate and minimizes overall compute time while keeping global prefetch time low. As a result, on this heterogeneous testbed, Halda reduces TPOT by 50% compared to the baseline (MemSched).

It's important to note that this smart scheduling is not hard-coded but derived automatically from our LDA formulation and Halda's solver. The resulting strategy adapts to device characteristics, so it generalizes across diverse testbeds by design.

### A.15 Ablation Study on Context Length

A longer context length increases the memory demand of the KV cache. Under a fixed memory budget, allocating more space to the KV cache leaves less cache available for model weights, making prefetch-release conflicts more likely. This, in turn, causes more frequent in-compute reloads and lower kernel efficiency.

To study how context length affects TPOT, we set up two heterogeneous testbeds: one with lower memory and another with larger memory:

- *Low-mem testbed:* including Mac M1 Pro (8 cores, 8 GiB UMA RAM), Mac M3 Pro (12 cores, 18 GiB UMA RAM), Linux laptop (8 cores, 4 GiB available RAM, and a 4090 GPU with 16 GiB VRAM), Honor Pad (8 cores, 5 GiB available RAM).

- *Medium-mem testbed:* including four Linux PCs, one with a 1080 Ti GPU (11 GiB VRAM) and 24 GiB RAM; one with a 2080 Ti (11 GiB) and 32 GiB RAM; one with a 3090 (24 GiB) and 16 GiB RAM; one with a 4060 Ti (8 GiB) and 32 GiB RAM; and a Mac mini with 16 GB of unified memory.

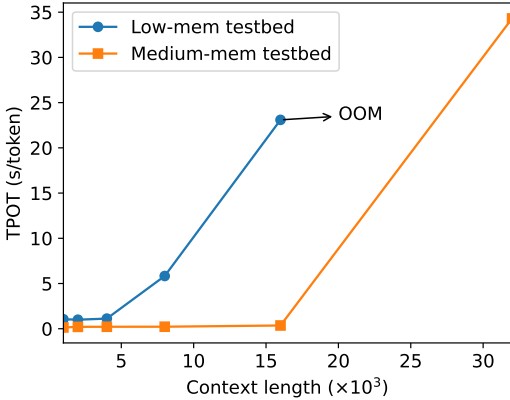

Figure 17: Performance breakdown (per token step) for MemSched, PerfSched, and Halda.

We ran the Llama-3-70B Q4K model on both testbeds and measured its TPOT under context lengths ranging from 1K to 32K tokens. Fig. 17 shows that increasing the context length has a safe region where TPOT remains flat. Larger memory provides a wider flat region, e.g., about 1-4K tokens on the low-memory testbed and 1-16K tokens on the medium-memory testbed. Beyond this range, TPOT rises sharply or causes OOM, as the expanding KV cache exhausts available memory.

### A.16 Ablation Study on Pipelined-Ring Parallelism on Heterogenous Testbed

We reproduced the experiment from Fig. 2 on the heterogeneous testbed described in Appendix A.14.2, comparing PP and PRP in terms of TPOT across models from 8B to 72B. As shown in Fig. 18, the results are consistent with the findings reported in the main text. When $k = 1$ or the model is small (i.e., each device's memory can fully hold its assigned model layers), PRP degenerates to PP. For smaller models, using PRP introduces additional kernel launch overhead, slightly increasing

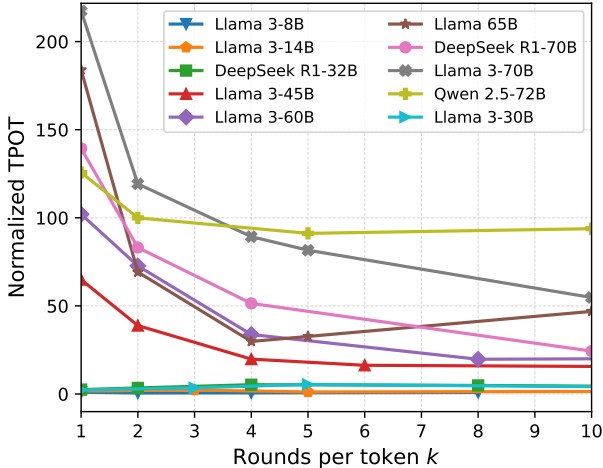

Figure 18: TPOT vs. per-token rounds ($k$) for 8-72B models on the heterogeneous testbed.

TPOT. However, for larger models (> 45B), PRP with $k > 1$ effectively reduces compute time, leading to a significant TPOT drop by 51%-84%.

### A.17 ABLATION STUDY ON NETWORK SENSITIVITY

To analyze prima.cpp's sensitivity to network bandwidth and link latency, we built a four-device heterogeneous testbed:

- Mac M1 Pro (8 cores, 8 GiB UMA RAM);
- Mac M3 Pro (12 cores, 18 GiB UMA RAM);
- Linux laptop (8 cores, 4 GiB available RAM, and a 4090 GPU with 16 GiB VRAM);
- Honor Pad (8 cores, 5 GiB available RAM).

All devices were connected via Wi-Fi. We configured bandwidth and link latency using *Network Link Conditioner* on the Macs and *tc* on the Linux laptop and tablet (simulated via Termux). The baseline RTT was 94 ms, and we incrementally added 100 ms latency to observe TPOT variation. Each experiment was repeated 5 times to plot the shaded area. On this testbed, we ran Llama-3-70B (Q4K) for inference.

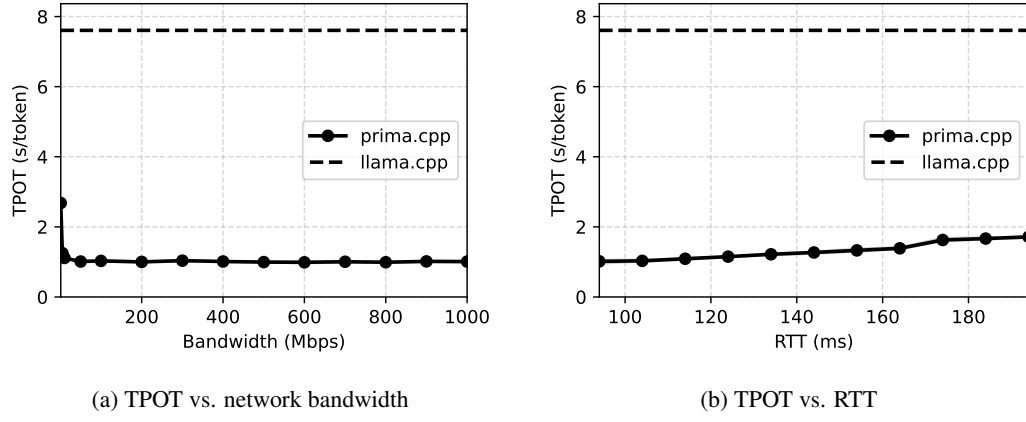

(a) TPOT vs. network bandwidth  (b) TPOT vs. RTT

Figure 19: Sensitivity of TPOT to (a) bandwidth and (b) RTT on the heterogeneous testbed.

From Fig. 19, we can draw two insights:

- Prima.cpp is insensitive to bandwidth changes, except under extremely low bandwidth (e.g., lower than 50 Mbps).
- Prima.cpp will be affected by link latency, but the impact is modest: when RTT increases from 100 ms to 200 ms, TPOT rises by 68%, but still far outperforms the baseline.

Therefore, from the network aspect, bandwidth is no longer the primary bottleneck, but link latency has become the main limiting factor.

### A.18  RUN PRIMA.CPP ON BUSY SYSTEMS

By default, all experiments were run on an idle system with no background apps running. In this section, we evaluate prima.cpp's performance on a busy system by running 6 types of background apps on the head device:

- *TikTok:* TikTok stresses multiple resources: network (audio/video fetch, bandwidth contention, added latency); CPU/GPU (video decoding, UI compositing, ABR bitrate control); and memory (A/V buffers, decoded-frame queues, prefetch caches).
- *Work:* This setup includes several commonly used productivity apps running simultaneously in the background: Chrome, Teams, Slack, ChatGPT, VSCode, Outlook, and WhatsApp.
- *YouTube:* Similar to TikTok, but YouTube consumes more CPU/GPU/memory/network resources due to long-form 1080p/60fps (or 4K) streams.
- *Zoom:* We co-ran Zoom with screen share, camera video, audio, and live captions enabled. It stresses CPU/GPU (real-time screen capture and HW encode, camera capture, UI compositing, noise suppression and echo cancellation), network (sustained uplink + download streams increase RTT/jitter), and memory (large frame buffers and capture/encode queues reduce effective page cache residency for model weights, causing more in-compute reloads).
- *Download:* Download a large game from App Store while prima.cpp is running in the background. It stresses network (sustained downlink saturates Wi-Fi bandwidth and uplink ACKs + CDN retries add jitter; queue buildup raises RTT and variance), CPU (TLS/QUIC encryption, checksum verification, and package integrity), disk (continuous writes contend with prima.cpp's read-only mmap of model weights), and memory (App Store and decompression buffers add moderate RAM pressure, shrinking available cache for model weights).
- *Game:* Play a large realtime game Asphalt (always ranked 1st in each race) while prima.cpp running in the background. It stresses GPU (3D rendering at high FPS), CPU (game logic/physics/input/render thread scheduling preempts background work, real-time threads and frame pacing reduce CPU headroom), RAM/VRAM (large textures and geometry increase shared memory usage, shrinking page-cache residency for model weights), disk (asset/texture streaming can add read bursts that interfere with mmap readahead), and network (multiplayer/telemetry traffic raises RTT/variance).

The experimental results are shown in Fig. 20. Each run was repeated 5 times. Everyday apps such as TikTok, YouTube, downloads, work suites, and video conferencing increase TPOT by only 5-18%, showing good resilience under normal workloads. The sharpest slowdown appears with large realtime 3D games, which heavily compete for GPU time and increase TPOT by 66%.

Although prima.cpp shows a slight slowdown, this behavior is expected. For home users, the smooth operation of everyday apps matters more than perfectly consistent LLM speed. For example, if an LLM service causes a game to lag or crash, users will likely terminate it to preserve gameplay. Our prima.cpp addresses this by adopting a lower RAM priority: it reduces memory use when other apps start and reclaims it when they stop, thanks to mmap-based on-demand loading and OS reclamation. By yielding RAM to keep other apps responsive, prima.cpp both protects the user experience and avoids being killed by the user.

### A.19  RUN PRIMA.CPP ON LARGER DEVICE POOL

In this section, we compare the performance of llama.cpp, exo, dllama, and prima.cpp on a larger 10-node heterogeneous testbed. The hardware configuration of this testbed is shown in Table 11,

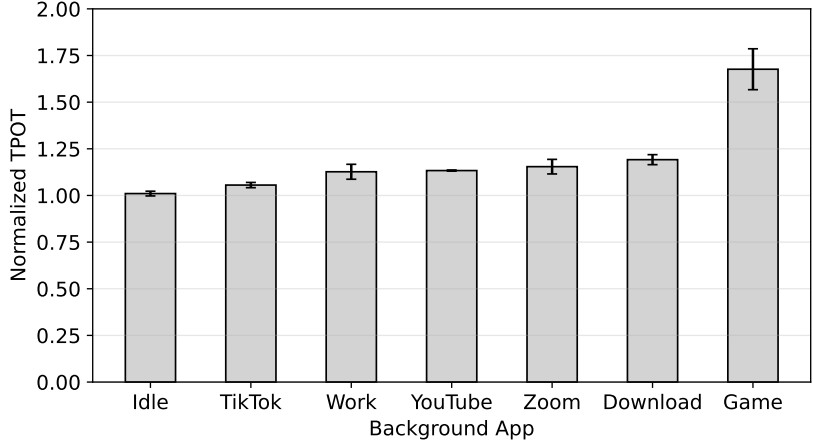

Figure 20: Normalized TPOT of prima.cpp under different types of background apps.

Table 11: Testbed hardware summary

| ID | Type | OS | Cores | RAM | GPU | VRAM | Disk Read |
|----|------|------|-------|--------|--------|--------|-----------|
| D0 | PC | Linux | 12 | 32 GiB | 4060Ti | 8 GiB | 0.56 GB/s |
| D1 | PC | Linux | 16 | 32 GiB | N/A | - | 1.89 GB/s |
| D2 | PC | Linux | 8 | 8 GiB | 3090 | 24 GiB | 1.23 GB/s |
| D3 | PC | Linux | 12 | 16 GiB | N/A | - | 2.64 GB/s |
| D4 | PC | Linux | 24 | 16 GiB | 1080Ti | 11 GiB | 0.32 GB/s |
| D5 | PC | Linux | 12 | 24 GiB | 1080Ti | 11 GiB | 1.57 GB/s |
| D6 | PC | Linux | 12 | 32 GiB | 2080Ti | 11 GiB | 2.67 GB/s |
| D7 | PC | macOS | 10 | 16 GiB | Metal | 11 GiB | 2.36 GB/s |
| D8 | Laptop | Linux | 18 | 32 GiB | N/A | - | 3.33 GB/s |
| D9 | Laptop | macOS | 8 | 16 GiB | Metal | 11 GiB | 3.88 GB/s |

covering both GPU and non-GPU devices, CPU/CUDA/Metal backends, heterogeneous OSs, RAM capacities from 8-32 GiB, VRAM capacities from 8-24GiB, and disk read speeds ranging from 0.32-3.88 GB/s.

### A.19.1 COMPARISON OF TPOT FOR LLAMA.CPP, EXO, DLLAMA, AND PRIMA.CPP

In this experiment, we compared the TPOT of llama.cpp, exo, dllama, and prima.cpp on models ranging from 7B to 70B. For Qwen models, exo supports only Qwen-2.5, but dllama only Qwen-3, so we compared their same-size models; llama.cpp and prima.cpp default to Qwen-2.5.

Llama.cpp ran on the most powerful device (D2), while exo (Llama) and prima.cpp ran on D0-D9. We also tested exo (Qwen) on D0-D9, but its Linux backend supports only Llama models, so it could run only on two macOS devices D7 and D9. Dllama splits tensors by KV heads and supports up to 8 devices, so we removed D1 and D8, two CPU-only devices with lowest FLOPS. To run the 70B model, we upgraded D2 to 128 GiB RAM for exo and dllama (64 GiB still caused OOM to dllama). Even so, exo still ran out of memory, as it requires to upgrade all Linux devices.

The results in Table 12 are consistent with those in Table 4:

- **Prima.cpp reduces to llama.cpp on small models.** For models smaller than 32B, Halda identifies D2-GPU as the most capable device, with 24 GiB VRAM sufficient to hold the full 4-bit model. It therefore assigns all layers to D2-GPU and removes other devices, reducing prima.cpp to llama.cpp.

- **Llama.cpp faces a sharp spike in TPOT due to prefetch-release conflicts on larger models.** When scaling to 70B, llama.cpp must offload excess layers to the CPU, but its 8

Table 12: TPOT (ms/token) for llama.cpp, exo, dllama, and prima.cpp.

| Model (4-bit) | llama.cpp | exo | dllama | prima.cpp |
|---|---|---|---|---|
| Llama-3.1-8B | 35 | 2500 | 2408 | **35** |
| Qwen-2.5-7B | 34 | 116[1] | 2177[2] | **34** |
| Qwen-2.5-14B | 52 | 169[1] | 2608[2] | **52** |
| Qwen-2.5-32B | 66 | 313[1] | 3399[2] | **66** |
| Llama-3.3-70B | 24037 | OOM | 8415 | **146** |

[1] Qwen models have lower TPOTs/TTFTs because devices D0-D6, D8 failed in exo.

[2] As dllama doesn't support Qwen-2.5, we use a Qwen-3 model of same size for comparison.

GiB RAM cannot hold the 16 GiB overflow, leading to prefetch-release conflicts. Frequent slow disk reloads cause TPOT to spike to an unusable 23 s/token.

- **Prima.cpp's TPOT scales smoothly and remains highly efficient.** With Halda's smart layer partitioning and device selection (see Appendix A.19.2), prima.cpp remains efficient at the 70B scale with a TPOT of only 146 ms/token.

- **Exo suffers a major slowdown due to its memory-proportional layer partitioning.** D1/D8 CPUs are the weakest, and D3's CPU, though faster, is still far slower than Metal GPUs. However, with exo's memory-proportional layer partitioning, D1/D3/D8 received half of the model layers, causing severe computation bottlenecks.

- **Exo's backend limits its OS compatibility.** For Qwen models, we run them on D0-D9, but D0-D6 and D8 failed because exo's Linux backend supports only Llama models, and exo provides only 8B/70B Llama variants. This limitation actually favors exo, as it excludes the slow CPU-only devices (D1/D3/D8). The remaining D7/D9 macOS devices have 22 GiB of shared Metal memory in total, enough for 4-bit models under 32B, and benefit from Apple Silicon GPU acceleration, so exo's TPOTs on 8-32B models are significantly lower for Qwen models than for Llama.

- **TP doesn't work well over high-latency networks.** In dllama, TP requires two all-reduce operations for each model layer, so an 80-layer model performs 160 communications per token step. With more devices, the overhead grows, leading to seconds of pure communication delay per token on a typical home network, even for small 7B/8B models.

### A.19.2 HALDA DEVICE SELECTION VS. HEURISTIC LAYER PARTITIONING

**Halda performs better than human intuition and heuristic methods.** Fig. 21 compares Halda with MemSched (used by exo) on Llama-3-70B (Q4K). For MemSched, its memory-proportional partitioning assigns nearly half the layers to CPU-only nodes (D1/D3/D8). On pure CPU backends, 4-bit weights are decoded into FP16/FP32 and stored in limited RAM, causing these nodes to run out of memory. Although we couldn't measure its TPOT, it is evidently higher than the 8B model's 2.5 s/token, which is already unacceptably slow.

Since the aggregate VRAM can hold the full model, we also propose an intuitive "Fill-Fastest-GPUs-First" heuristic that partitions layers by GPU compute rank (D2 > D0 > D6 > D5 > D4 > D7 > D9 > D3 > D1 > D8), yielding a 13/46/19/2 split across D2/D0/D6/D5 (80 layers total).

Halda adopts a slightly different plan: it removes D5 and moves its two layers to D2's CPU. This is counterintuitive because D5's GTX 1080Ti GPU has about 30× the FLOPS of D2's CPU. However, Halda's adjustment lowers TPOT from 156 ± 7 ms/token to 146 ± 5 ms/token. The reason behind this counterintuitive result is that adding D5 introduces an extra RTT per token on the high-latency Wi-Fi ring, which costs more than D5 GPU's gain. In contrast, offloading to local CPUs avoids the additional RTT and keeps the communication ring shorter.

**More devices do not mean better performance.** In this case, Halda achieves the fastest speed with just 3 devices, and adding more weak devices offers no benefit because Halda will remove them automatically. In practice, home users prefer fewer devices for lower power use and simpler setup, so a candidate pool of up to 5 devices is enough (as users may not have more).

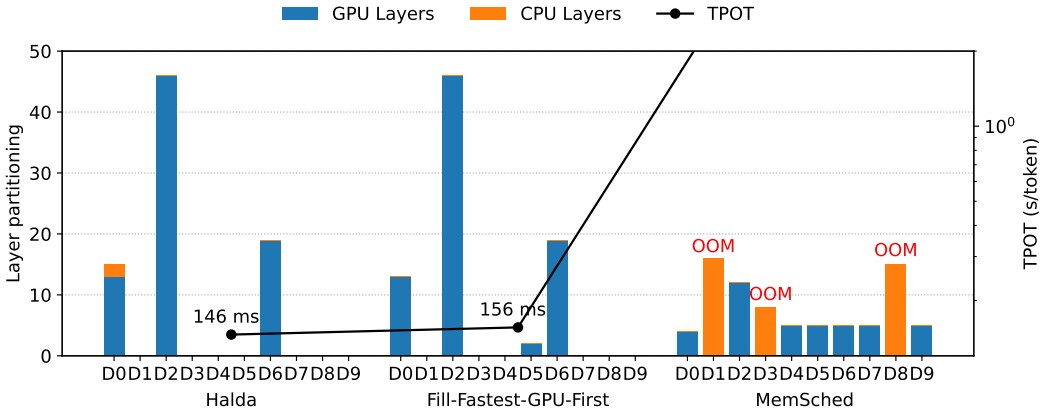

Figure 21: Layer partitioning distribution of prima.cpp and exo on Llama-3-70B (Q4K).

