# OpenReview forum: "Prima.cpp: Fast 30-70B LLM Inference on Heterogeneous and Low-Resource Home Clusters"
_ICLR.cc/2026/Conference — ICLR 2026 Poster_

### Official Review · Reviewer_kAiH · 2025-10-29

**Soundness:** 2
**Presentation:** 2
**Contribution:** 2
**Rating:** 4
**Confidence:** 3

**Summary:**

The paper introduces prima.cpp, a distributed inference framework designed to run 30–70B parameter language models on heterogeneous and memory-constrained “home clusters.” The core ideas are (i) pipelined-ring parallelism (PRP), a variant of pipeline parallelism that repeatedly circulates activations around a ring of devices to overlap disk I/O with computation, and (ii) Halda, a heterogeneity-aware scheduler formulated as an integer linear program (ILP) that assigns per-device layer windows and CPU/GPU splits while enforcing RAM/VRAM constraints. Experiments on a Wi-Fi connected cluster of four to six consumer devices compare prima.cpp against llama.cpp, exo, and dllama, reporting large speedups in time-per-output-token (TPOT) and time-to-first-token (TTFT). The authors also present ablations on Halda and prefetching and discuss memory pressure considerations.

**Strengths:**

Originality: The paper tackles a timely and underexplored setting—distributed on-device inference across heterogeneous, low-cost consumer hardware. Integrating mmap-based offloading with pipeline parallelism and heterogeneous scheduling is an interesting direction. The attempt to capture OS-level memory reclamation in the scheduling formulation is a thoughtful touch.

Quality: The implementation effort is substantial (∼20k LOC) and the evaluation spans a range of model sizes up to 70B parameters, which is rare for “home cluster” work. The ablation that removes Halda highlights the importance of scheduling in this regime, lending credibility to the design.

Clarity: The writing is generally clear, with useful figures (e.g., Fig. 1 showing PRP) and tabular summaries of baselines and experimental results. The high-level algorithmic flow of Halda (Algorithm 1) is easy to follow.

Significance: Demonstrating sub-second TPOT for 70B models on commodity hardware, if validated, would be impactful for privacy-preserving and offline LLM deployment. The emphasis on low memory pressure addresses an important practical concern for end users.

**Weaknesses:**

Soundness of PRP claims: PRP is positioned as a key innovation, yet the mechanistic explanation for resolving the “prefetch-release conflict” is largely qualitative. There is no formal or empirical breakdown of the time saved per component (disk vs. compute vs. communication), and the homogeneous-cluster experiment in Fig. 2 lacks realism (uniform 8-core CPUs and SSDs) relative to the heterogeneous setting the paper targets. It remains unclear how much of the reported TPOT gains stem from PRP versus simply better scheduling or quantization choices.

Baseline fairness and completeness: The experimental section focuses exclusively on latency metrics. There is no evaluation of model quality (perplexity, task accuracy) under the aggressive quantization and speculative decoding used. Comparisons with exo and dllama are limited: exo “runs on D1–D3” but the authors disable D4 due to root access requirements, while prima.cpp does leverage D4; dllama is measured only on the 8B model and declared OOM later without empirical numbers. Additionally, llama.cpp is constrained to a single device (D3), even though its distributed variants or multiple instances could also use the other devices, making the comparison potentially unfair.

Ablations insufficient for causal attribution: The ablation “prima.cpp (w/o halda)” simply reuses exo’s partitioning rule but retains the rest of the system. There is no attempt to isolate Halda’s scheduling quality from the PRP machinery, nor to test alternative schedulers in a controlled manner. Similarly, the prefetching ablation shows 9–17% improvements, yet TPOT reductions of up to 17× over llama.cpp are claimed; the gap between these numbers is not reconciled.

Scalability and generality concerns: The entire evaluation uses a single anonymized cluster with modest Wi-Fi bandwidth. There is no analysis of how prima.cpp scales with more than six devices, higher-latency networks, or weaker hardware (e.g., devices without SSDs). The claim that Halda “selects” devices is not backed by experiments on a large pool where many must be excluded. Furthermore, the scheduler assumes access to per-device disk throughput and available memory, but it is unclear how those are measured or kept up to date in deployment.

Clarity issues and missing details: Several important implementation choices are omitted: handling of activation checkpointing or KV cache distribution, the cost of coordinating speculative decoding across devices, and how OS-specific behaviors (e.g., macOS page cache policies) are detected at runtime. The paper also references Appendix sections for critical arguments (e.g., Appendix A.1 for the prefetch-release conflict) rather than summarizing key empirical evidence in the main text.

**Questions:**

PRP vs. standard PP: Can you quantify the incremental benefit of multi-round PRP over single-round pipeline parallelism on the heterogeneous cluster, with and without Halda? A per-stage latency breakdown (compute, disk, communication) would help attribute the gains to specific mechanisms.

Baseline configuration fairness: Why is llama.cpp restricted to a single device while prima.cpp uses all available devices? Have you evaluated llama.cpp with manual layer partitioning (e.g., via tensor or pipeline parallel modes) to assess whether the comparison is fair?

Quality metrics under quantization: Do the aggressive quantization formats (e.g., Q4K, IQ1) and speculative decoding degrade output quality? Please report perplexity or downstream task accuracy to demonstrate that prima.cpp preserves model performance relative to the baselines.

Device selection robustness: Halda is claimed to automatically drop slow devices. Could you provide experiments on a larger heterogeneous pool (≥10 devices) showing how many devices are selected, how often the assignment changes, and the resulting impact on TPOT?

Convergence and overhead of Halda: What is the typical number of iterations before Halda converges, and how long does the scheduling phase take relative to inference time? Are there scenarios where the ILP fails to find a feasible solution or oscillates between assignments?

Network sensitivity: How does prima.cpp perform under higher-latency or lower-bandwidth Wi-Fi (e.g., 50–100 ms RTT or 100 Mbps throughput)? Since home networks can be noisy, understanding the tolerance to adverse conditions is important for practical deployment.

Memory pressure measurement methodology: Table 5 reports “memory pressure” as percentage reductions in mem_available. Can you detail the measurement interval (sampling frequency), the baseline state (idle system vs. with other apps running), and whether fluctuations during inference were observed?

---

> ### Author Response · Authors · 2025-11-25
> **Authors' response to Weaknesses 1, 3, and Question 1 (sharing the same concern).**
>
> We appreciate your careful and detailed review. As each of your comments includes multiple questions, we summarize your main concerns and address them point by point.
>
> ---
>
> **For Weaknesses 1, 3, and Question 1:**
>
> Thank you for this valuable suggestion. To address your concern, **we have added Appendix A.14 to include a performance breakdown to show the existence and impact of prefetch-release conflicts, as well as the latency composition of disk, compute, and communication before and after applying PRP/Halda. This experiment runs on a heterogeneous testbed.**
>
> Below, we briefly summarize the key points:
> - **Results in Fig. 14 confirm the existence of prefetch-release conflicts:** even with prefetching completes, the in-compute reload bytes approach 25 GiB (roughly the total model weights offloaded to CPUs), indicating that nearly all prefetched weights were reclaimed before use, which is clear evidence of prefetch-release conflicts. The sharp drop in cache hit rate causes high-frequency page faults and off-chip memory access, making compute time surge and dominate latency.
> - **From the in-compute reload size curve in Fig. 15, PRP clearly reduces the amount of prefetched data reclaimed by the OS.** This raises cache hit rates and sharply shortens compute time, **showing that PRP effectively resolves prefetch-release conflicts and improves kernel efficiency over PP.** In addition, PRP reduces exposed disk I/O more effectively. Overall, **PRP reduces per-token latency to just 20-23% of that under PP.**
> - The results of heuristic rules (MemSched, PerfSched) are highly random, but Halda performs much smarter scheduling: it always finds a much better partitioning rule. This increases devices’ cache hit rate and minimizes overall compute time while keeping global prefetch time low. As a result, **Halda reduces TPOT by ~50\% compared to exo’s layer partitioning rule.**
>
> **Regarding your concern about Fig. 2's homogeneous testbed.** In prima.cpp, Halda handles device heterogeneity, while PRP focuses only on the prefetch-release conflict. To isolate PRP's contribution, we chose a homogeneous testbed for ablation analysis to remove heterogeneity. To address your concern, **we reproduced this experiment on a heterogeneous testbed in Appendix A.16. The results are consistent with the findings reported in the main text.** When $k = 1$ or the model is small (i.e., each device's memory can fully hold its assigned model layers), PRP degenerates to PP. For smaller models, using PRP introduces additional kernel launch overhead, slightly increasing TPOT. However, for larger models (> 45B), PRP with $k > 1$ effectively reduces compute time, leading to a significant TPOT drop by 51\%-84\%.
>
> **For quantization,** both llama.cpp and prima.cpp use the Q4K format, while exo employs MLX for 4-bit computation and dllama uses Q40. **All of them are based on 4-bit quantization schemes.**
>
> In summary, the performance gain mainly comes from PRP, which resolves prefetch-release conflicts to improve cache hit rate and kernel efficiency, and from Halda’s smart layer partitioning scheduling, rather than from using a different quantization bitwidth.

---

> ### Author Response · Authors · 2025-11-25
> **Authors' response to Weakness 2 and Questions 2,3 (sharing the same concerns).**
>
> **For Weakness 2 and Questions 2, 3:** Thank you for this insightful comment. We would like to clarify that:
>
> **a. For model quality:** Prima.cpp uses the same pre-quantized models as the baseline (e.g., llama.cpp) and does not change model outputs. For quantization, prima.cpp deploys models that are already quantized and does not perform any additional quantization itself. For speculative decoding, the verification step ensures the output tokens are identical to those of the original model. Thus, **prima.cpp produces outputs identical to the baseline (llama.cpp)**, and metrics such as perplexity and task accuracy can be directly referenced from the open-source model card.
>
> **b. For exo:** We understand your concern about excluding device D4 in the exo experiments, but we would like to clarify that **we attempted to run exo on D1-D4, but D4 failed because it requires root access on the phone, not because we intentionally excluded it**. This limitation is inherent to exo’s implementation, highlighting prima.cpp’s better compatibility and lower permission requirements for home devices.
>
> In fact, removing D4 benefits exo, since D4 is the weakest device. Excluding it allows more model layers to run on stronger devices (D2 and D3). For example, deploying a Q4K-quantized 8B model requires only about 5 GiB VRAM, and the optimal setup is single-device inference on D3 (the device with the most powerful decoding) using llama.cpp. In prima.cpp, the Halda scheduler learns this and assigns all layers to D3, yielding the same minimal TPOT and TTFT as llama.cpp. For exo, however, even though removing D4 is better, its memory-ratio-based layer partitioning leads D1 and D2 to take most of the layers (8:8:11), reducing D3’s workload and causing much higher TPOT than llama.cpp and prima.cpp.
>
> In summary, the exo experiment supports two conclusions:
> - Compared with exo, prima.cpp requires no root access, offers better cross-platform compatibility (including phones and tablets), and provides more model size options (exo only supports 8B and 70B models across devices with diverse OSes).
> - **Removing D4 is NOT unfair; instead, it benefits exo,** but due to its inefficient layer partitioning, exo still performs much worse than prima.cpp (thanks to Halda).
>
> **c. For dllama:** Fortunately, a recent dllama update added support for more models. To address your concerns about testbed size, exo, and dllama, **we have added Appendix A.19.1 to compare them on a larger, heterogeneous testbed with more available models.**
> - We built a 10-node testbed with larger memory to allow exo and dllama to distribute their workloads more effectively.
> - We added newly-supported/customized models for dllama and updated Tables 4, 5, 8, 9 to include more valid dllama results.
> - We upgraded the master node to 128 GiB RAM for dllama.
>
> After these efforts, **we collected more data for exo and dllama, and their results remained consistent with the conclusions in Table 4.**
>
> **d. For llama.cpp:** Although llama.cpp supports distributed inference through its RPC backend, **it relies on dense communication, which performs poorly on high-latency home Wi-Fi.** On a two-node testbed with a 70B model, prima.cpp achieved 0.4 s/token, while llama.cpp (RPC backend) reached 7.5 s/token, much higher than both its standalone version and prima.cpp. **After confirming with the llama.cpp authors, the RPC backend is intended for low-latency HPC environments, not high-latency home networks.** Hence, for fair comparison, we didn't use it for comparison but kept its standalone version as a baseline.
>
> **e. Why not vLLM or SGLang?** We also tried vLLM and SGLang, however, they are designed for servers and none could run on our testbed. Please see the response to Reviewer 7c8W Question 3 for details.
>
> **f. Any other options?** Some inference systems are designed for end devices, but most run on a single device, e.g., SpecExec/UMbreLLa recommended by Reviewer 6ZVN. Experiments show that, compared with SpecExec/UMbreLLa, which rely on sufficient RAM capacity and RAM offloading, prima.cpp achieves a higher speedup simply by adding a readily available helper device. Please see the response to Reviewer 6ZVN Question 1 for details.
>
> **To the best of our knowledge, only exo (32k stars) and dllama (2.8k stars) are viable and most well-known baselines. Their limitations should NOT be attributed to our “unfair” comparison**, as we have chosen the most popular alternatives of the same category to ensure fairness. In contrast, their limitations further highlight prima.cpp’s clear advantages in usability and performance.
>
> Finally, I would like to quote a comment from an external reviewer:
>
> > *To the best of my knowledge, prima.cpp is the first system to deliver practical performance for 70B models in such constrained environments.*
>
> We hope this addresses your concerns.

---

> ### Author Response · Authors · 2025-11-25
> **Authors' response to Weakness 4 and Questions 4, 6 (sharing the same concerns).**
>
> **For Weakness 4 and Questions 4, 6:** We appreciate your valuable suggestions.
>
> **a. For the diversity of testbeds:** We would like to clarify that we evaluated prima.cpp on 6 heterogeneous testbeds (Table 3; Appendices A.4/A.10, A.5/A.9, and the new A.11/A.15, A.13/A.15, A.19), spanning 3 real home networks with different latencies, rather than a single cluster. In total, we used 18 devices: 5 Mac devices (Air/M1/M3/M4/Mini), 4 Linux laptops with 3070/4060 Ti/4090 GPUs and CPU-only, 6 Linux PCs with 1080 Ti/2080 Ti/3090 GPUs and CPU-only, 2 phones and one tablet (HarmonyOS and Android). Some only had HDDs/UFSs, and some featured low compute power, high latency, and small memory. These diverse testbeds and devices cover a wide range of real-world network and hardware conditions, and all derived conclusions remain consistent with those reported in Table 4.
>
> **b. For the scale of testbeds:** To address your concern, **we have added Appendix A.19 to compare llama.cpp, exo, dllama, and prima.cpp on a larger, heterogeneous testbed with 10 devices, and show Halda’s device selection capability versus human experts.** The results show that:
> - Our prima.cpp outperforms llama.cpp, exo, and dllama in both TPOT and OOM risks, consistent with the findings in Table 4.
> - Halda selected 3 devices out of 10 to form the best-performing cluster, achieving a TPOT 6% lower than the human experts' rule and 99% than exo’s rule.
>
> **c. For Halda's device selection on a larger, heterogeneous testbed:** To address your concern, **we have added Appendix A.19.2 to show Halda’s device selection capability on a heterogeneous pool with 10 devices.** In this case, Halda selected 3 devices out of 10 to form the best-performing cluster, achieving a TPOT 6% lower than human experts' rule and 99% than exo’s rule.
>
> Halda's assignments are determined during startup and remain fixed during inference. When the job queue is idle, we can perform light profiling to update profiling metadata and Halda's assignments. This idle-time update does not block inference or impact TPOT.
>
> If the environment changes, e.g., when the user starts TikTok, YouTube, work suites (e.g., Chrome, Teams, Slack, ChatGPT, VSCode, Outlook, WhatsApp), or video conferencing apps (e.g., Zoom, Live Captions), TPOT may drop by 5-18%. Please see the response to Question 7 for details.
>
> **d. For the network sensitivity:** To address your concern, **we have added Appendix A.17 to include an ablation study on network bandwidth and link latency.**
>
> Below, we briefly summarize the key points:
> - Prima.cpp is insensitive to bandwidth changes, except under extremely low bandwidth (e.g., lower than 50 Mbps).
> - Prima.cpp will be affected by link latency, but the impact is modest: when RTT increases from 100 ms to 200 ms, TPOT rises by 68\%, but still far outperforms the baseline.
> - Bandwidth is no longer the primary bottleneck, but link latency has become the main limiting factor.
>
> Nonetheless, under your specified conditions (50-100 ms RTT and 100 Mbps bandwidth), prima.cpp still delivers strong performance.
>
> **e. For disk and memory profiling:** We run a profiler module on each device to collect metadata, including the CPU/GPU FLOPS, RAM/VRAM capacity and bandwidth, disk sequential/random read speed, and OS/backend type.
> - FLOPS: measured by running typical tensor operations on the CPU/GPU.
> - RAM capacity: read from system files such as */proc/meminfo* and *memory.stat* on Linux, *host_statistics64* on macOS, and *smaps* in Termux, then calculated using the formula in Appendix A.3.
> - VRAM capacity: read from Metal/CUDA APIs.
> - RAM/VRAM bandwidth: estimated by loading typical tensor sizes into device memory without computation.
> - Disk throughput: measured with *fio* for sequential and random reads.
>
> As shown in Appendix A.12, Fig. 12, each device forwards its collected metadata along the communication ring to the head device’s global profiler, which uses this data for LDA modeling and Halda scheduling. Details can be found in Appendix A.12 “Profiling and Heterogeneity-Aware Layer Partitioning.”
>
> **f. How to keep profiling up to date?** Profiling runs at startup, and when the job queue is idle, so it never blocks inference. A full profiling (during startup) takes about 6s (5.3s from disk testing on slow devices), while light profiling (during idle time) skips disk tests and finishes within 1s. Thus, the profiling metadata keeps up to date during runtime without pausing inference.
>
> We hope this addresses your concerns.

---

> ### Author Response · Authors · 2025-11-25
> **Authors' response to Weakness 5.**
>
> **For Weakness 5**: Due to page limits, the main text focuses on presenting the core ideas behind our key contributions, while implementation details were distributed across several sections. To address your concern, **we have added Appendix A.12 to include a consolidated description of prima.cpp framework.**
>
> **a. For activation checkpointing:** There is NO activation checkpointing during inference.
>
> **b. For KV cache distribution:** Model layers and their corresponding KV caches stay on the same device-backend and never migrate. Please see Appendix A.12 “Mmap- and prefetch-based PRP” and Fig. 12 for details.
>
> **c. For speculative decoding:** Since the draft model is small, it runs solely on the head device, and the draft tokens are consumed by the target model’s input layer on the same device, so NO additional cross-device coordination is introduced. Please see Appendix A.12 “Speculative decoding” and Fig. 12 for details.
>
> **d. For OS-specific behaviors:** **OS-specific behaviors are public and well-documented.** For example, macOS without Metal and Linux use gradual reclamation; macOS with Metal releases memory aggressively only under high pressure; Android evicts cold pages first; Linux optimizes sequential reads. Thus, **we detect the OS/backend type at runtime rather than OS-specific behaviors**, so we can estimate reload data size and latency using metadata like disk-read speeds and cache size.
>
> **e. Empirical evidence for the prefetch-release conflict:** We appreciate this valuable suggestion. **We have added Appendix A.14 to include empirical evidence for the prefetch-release conflict.** Also, we update the main text on page 3, line 155 as below:
>
> > Appendix A.14 provides empirical evidence for the existence and impact of prefetch-release conflicts: even after prefetching completes, required model weights still need to be reloaded during computation.

---

> ### Author Response · Authors · 2025-11-25
> **Authors' response to Questions 5, 7.**
>
> **For Question 5:** Thank you for this insightful comment.
>
> **a. For the typical iterations and time of Halda:** According to the complexity analysis in Section 3.3, the outer *while* loop runs at most *O(M)* iterations in the worst case, where *M* is the number of devices. In our practice, it typically converges within 1-3 iterations, and the Halda scheduling delay is only 10-12ms, negligible compared to the total inference time. Similar to full/light profiling, Halda’s scheduling/re-scheduling occurs only during the startup phase or when the job queue is idle, so the scheduling phase will never block inference.
>
> **b. For ILP failures and oscillations:** Most ILP failures we observed were caused by *fio* errors, which produced abnormal zeros in profiling metadata and made the ILP unsolvable. When profiling runs correctly, ILP failures do not occur in our practice.
>
> For assignment oscillations, we have identified and fixed them in earlier implementations. We added a rollback mechanism and a forced set $M^{\text{force}}_4\subset M_4$ that monotonically shrinks the feasible region. Each iteration either moves forward by adding more devices to $M_4$ or converges, preventing any oscillation. In our long-term practice, no further oscillations have been observed.
>
> ---
>
> **For Question 7:** Thank you for this comment.
>
> **a. For sampling frequency and baseline state:** We need only two memory samples, one before and one after inference. Let the total memory be $M$, we read *mem_available* from *free* as $M_1$ at process startup and $M_2$ after inference, then memory pressure can be calculated as $\frac{M_1-M_2}{M}$. To isolate the memory usage of prima.cpp itself, the baseline state is an idle system running only system processes.
>
> **b. For fluctuations during inference:** To address your concern, we re-ran the experiment and recorded the *mem_available* data every 0.5 seconds. The results show a flat line with only small, infrequent spikes, indicating that:
> - Resident memory usage of prima.cpp is deterministic: only the KV caches and compute buffers are pinned in memory, while model weights reside in clean file-backed pages that are not counted in *mem_available*. Hence, available memory remains stable during inference, consistent with the observed curve.
> - Fluctuations exist but are minimal (< 1% of total memory) and infrequent (> 1 min apart). These slight variations come from normal OS cache reclamation and have no impact on performance.
>
> **c. Idle system vs. busy system:** We appreciate your reminder to include a comparison between the idle and busy system states. Since the resident memory of prima.cpp is fixed across idle and busy states, we turn to compare the overall TPOT. As shown in **Appendix A.18**, everyday apps such as TikTok, YouTube, downloads, work suites, and video conferencing increase TPOT by only 5-18\%, showing good resilience under normal workloads. The sharpest slowdown appears with large real-time 3D games, which heavily compete for GPU time and increase TPOT by 66\%.
>
> Although prima.cpp shows a slight slowdown, this behavior is expected. For home users, the smooth operation of everyday apps matters more than perfectly consistent LLM speed. For example, if an LLM service causes a game to lag or crash, users will likely terminate it to preserve gameplay. Our prima.cpp addresses this by adopting a lower RAM priority: *it reduces memory use when other apps start and reclaims it when they stop*, thanks to mmap-based on-demand loading and OS reclamation. **By yielding RAM to keep other apps responsive, prima.cpp both protects the user experience and avoids being killed by the user.**
>
> ---
>
> We hope this rebuttal addresses your concerns, and we appreciate your kindness in reevaluating our score.

---

### Official Review · Reviewer_6ZVN · 2025-10-31

**Soundness:** 3
**Presentation:** 3
**Contribution:** 2
**Rating:** 6
**Confidence:** 5

**Summary:**

This paper presents prima.cpp, a distributed on-device inference framework enabling large language models (30–70B) to run efficiently on heterogeneous consumer clusters with limited resources. It introduces pipelined-ring parallelism (PRP) to overlap disk I/O, computation, and communication, and Halda, a heterogeneity-aware scheduler optimizing CPU/GPU utilization under RAM/VRAM constraints. Experiments on four home devices demonstrate strong performance—674 ms/token for 70B and 26 tokens/s for 32B models.

**Strengths:**

1. Efficient large-model deployment under limited VRAM/RAM.
2. Clear, well-structured, and easy-to-follow paper.
3. Solid, industrial-grade implementation with strong reproducibility.

**Weaknesses:**

1. Not show comparison with capable baselines, like SpecExec [1], which reaches 100-250ms/token TPOT.

2. Leveraging the Disk to offload (because of frequent writing and erasing) will cause damage to the hardware, which is not discussed in the limitation.

3. Lack of an ablation study of the impact of workload, such as context length, frequency of requests, etc.




[1] https://github.com/Infini-AI-Lab/UMbreLLa

**Questions:**

As those in weakness.

1. Could you provide insights on energy consumption when running prima.cpp?
2. How is the framework implemented for different hardware backends?

---

> ### Author Response · Authors · 2025-11-25
> **Authors' response to Weaknesses 1-3.**
>
> Dear reviewer, thank you for recognizing our work and for giving us the opportunity to have it re-evaluated.
>
> ---
>
> **For Weakness 1:** We appreciate your suggestion. We would like to clarify that although prima.cpp and SpecExec/UMbreLLa both target on-device LLM inference, **they belong to different categories.**
> - SpecExec/UMbreLLa is designed for *single-device* inference, but prima.cpp is designed for *distributed on-device* inference.
> - The main contributions of SpecExec/UMbreLLa are RAM offloading and speculative decoding, while prima.cpp focuses on a new parallelism paradigm and heterogeneity-aware layer partitioning. Although prima.cpp supports speculative decoding, it is just an optional feature for faster speed, not a core contribution.
>
> Therefore, SpecExec/UMbreLLa are not our baselines. Nevertheless, **we include a comparison with them to address your concern.**
>
> We built a 3-node testbed for comparison:
>
> - Device D1: 16 GiB RAM, RTX 3090 GPU (24GiB VRAM);
> - Device D2: 32 GiB RAM, RTX 3090 GPU (24GiB VRAM);
> - Device D3: 16 GiB RAM, CPU only (no GPU).
>
> The network bandwidth is 300 Mbps with an RTT latency of 100ms. We used Llama-3.1-70B as the target model and Llama-3.1-8B as the draft model. UMbreLLa was configured with AWQ-INT4 quantization, while prima.cpp used Q4K quantization. Since UMbreLLa is a single-device system, we ran it on the more powerful devices D1/D2. In contrast, our prima.cpp is a distributed system, so we ran it on D1+D3.
>
> UMbreLLa relies on RAM offloading for 70B-level inference, so it requires ample RAM on a single device. Instead, prima.cpp has no strict RAM requirements and works on arbitrary low-resource devices. The results show that, **with only 16 GiB RAM on device D1, UMbreLLa runs out of memory, while prima.cpp runs normally**. To run UMbreLLa, we upgraded D1 to 32 GiB RAM (D2), achieving an 88% speedup.
>
> In contrast, **prima.cpp takes an easier route: instead of spending about $100 on an extra memory card, we simply add an existing device** (D3) with 16 GiB RAM to form a two-node cluster (D1+D3). This setup offers the same total RAM/VRAM as D2 but **delivers a 100% speedup, 12% higher than UMbreLLa.** Hence, prima.cpp is more convenient, cost-effective, and efficient.
>
> (PS: Upgrading memory cards can be inconvenient and risky. When we upgraded D1 to D2, the motherboard was damaged, costing us nearly $1000 in repairs. In contrast, prima.cpp provides an easier and safer alternative.)
>
> ---
>
> **For Weakness 2:** Thank you for this insightful comment. Our prima.cpp uses mmap read-only mapping with automatic OS page-cache reclamation, triggering page faults only when data is needed. The cached pages are clean and therefore not written back when evicted. **We never write model weights or KV data to disk,** nor raise memory pressure to cause swapping. In practice, the measured write bytes are zero, so **there is no write-erase cycle; the disk handles read I/O only, and heavy reads have minimal impact on SSD or UFS lifespan.**
>
> ---
>
> **For Weakness 3:** Thank you for this valuable suggestion. To address your concern, **we have added Appendix A.15 to include an ablation study on context length, and Appendix A.11 to include an ablation study on concurrent requests.**
>
> Below, we briefly summarize the key points:
> - **Context length:** Increasing the context length has a safe region where TPOT remains flat. Larger memory provides a wider flat region, e.g., about 1-4K tokens on the low-memory testbed and 1-16K tokens on the medium-memory testbed. Beyond this range, TPOT rises sharply or causes OOM, as the expanding KV cache exhausts available memory.
> - **Concurrent requests:** When the model is large, or memory is tight, throughput grows sublinearly, peaks around 8-16 concurrent requests, then declines. This is expected: higher concurrency consumes more memory for KV caches and compute buffers, leading to more frequent model weight eviction and reloads. When memory is more abundant, throughput scales nearly linearly at first, saturates around 32 concurrent requests, and then drops. The “safe” ranges cover typical household usage, which is often 1-8 concurrent requests.

---

> ### Author Response · Authors · 2025-11-25
> **Authors' response to Questions 1 and 2.**
>
> **For Question 1:** Thank you for this valuable suggestion. We agree that energy is a key metric for both edge and cloud serving. To address your concern, **we have added Appendix A.13 to compare the energy consumption** per 1K output tokens across 3 settings: local single-device, local prima.cpp-distributed, and cloud inference.
>
> Here we briefly summarize the key points:
> - **Local single-device vs. prima.cpp distributed:** Switching from single-device llama.cpp to distributed prima.cpp reduces per-device energy by 91–99% and total energy by 57–90% for 1K tokens, and assigning a few layers to low-power helpers can shorten the critical path at low energy cost.
> - **Local serving vs. cloud serving:** The total energy consumption of the cloud GPU is 28% lower than a local cluster. If we take into account the cooling power consumption of data centers, the energy gap narrows. Regarding energy cost, many countries/regions use tiered electricity pricing: residential prices are about 30% cheaper than commercial prices, which offsets the energy-cost gap between cloud and local serving.
>
> To sum up, **prima.cpp significantly reduces energy consumption compared to local single-device serving, while maintaining comparable energy use and cost to cloud serving.** If we consider token pricing, bandwidth fees, cloud failure, and cloud costs, local serving with prima.cpp would be more cost-effective.
>
> ---
>
> **For Question 2:** Thank you for this insightful comment. We will answer this question from four aspects.
>
> **(a) How are different hardware backends implemented in prima.cpp?**
> Prima.cpp is built on GGML, which provides a plugin-style backend API for CPU (e.g., x86 AVX/AVX2/AVX-512/AMX, ARM NEON; Apple Accelerate), CUDA, HIP, ROCm, MUSA, Metal, Vulkan, OpenCL, and SYCL, so the higher-level code is backend-agnostic.
>
> **(b) Why can devices use heterogeneous rather than homogeneous backends?**
> Each device typically supports up to two types of backends: CPU and GPU. In prima.cpp, communication occurs only between CPU-GPU within a device and CPU-CPU across devices, using Ethernet instead of backend-specific networks like NCCL or RCCL. Thus, every device treats others as black boxes, exchanging only activations in a unified tensor format. Model weights and KV caches stay on their owning devices and never migrate, avoiding cross-backend consistency issues. As a result, prima.cpp runs efficiently without requiring homogeneous backends across devices.
>
> **(c) Is there any special design for heterogeneous backends in prima.cpp?**
> Yes. Although prima.cpp is backend-agnostic in implementation, **Halda is backend-aware**: it models backend-specific compute, memory, I/O, and OS behaviors to schedule layer partitioning accordingly.
>
> Beyond the FLOPS differences, hardware backends vary significantly in many aspects. Take Metal and CUDA as examples:
> - Metal uses UMA where CPU and GPU share memory, eliminating RAM-VRAM copies. In contrast, discrete CUDA GPUs require additional device-copy operations.
> - On CUDA, model weights pinned in private VRAM remain resident, so no disk reloads are needed. With Metal, mmapped weights reside in shared memory, but once memory is exhausted, macOS aggressively reclaims them, potentially forcing full-model reloads.
> - Reclamation on Linux or non-Metal setups is more gradual.
>
> These differences have significant effects on the optimal layer partitioning rule. To this end, Halda incorporates backend heterogeneity into its LDA model (see Appendix A.3), ensuring the derived scheduling strategy adapts to diverse backends.
>
> **(d) How to extend to more backends?**
> In this prototype, we support CPU, CUDA, and Metal backends (limited by our available hardware). However, as described in (a), GGML also supports other backends. To extend prima.cpp to others (e.g., Vulkan or ROCm), we only need to incorporate their performance characteristics, such as compute throughput, memory behavior, and I/O patterns, into Halda’s cost model. Once these parameters are modeled, prima.cpp can easily support new backends.
>
> ---
>
> We hope this rebuttal addresses your concerns, and we appreciate your kindness in reevaluating our score.

---

### Official Review · Reviewer_7c8W · 2025-11-01

**Soundness:** 3
**Presentation:** 3
**Contribution:** 2
**Rating:** 4
**Confidence:** 3

**Summary:**

The paper introduces prima.cpp which is a distributed on-device inference system to run LLMs on home clusters connected via Wi-Fi. It enables fast, private, offline-inference using heterogeneous CPUs/GPUs and limited memory and slow disks.

The paper extends pipeline parallelism to pipelined-ring parallelism where the devices form a ring, each processing a layer window of the model across multiple rounds per token. The work also includes Halda which partitions the model layers across devices considering CPU/GPU spec. It formulates the problem as a Integer Linear Fractional Program to minimize TPOT.

**Strengths:**

* On-device AI is an important topic considering the privacy concerns with regard to LLM inference that may handle personal sensitive data.
* Formulation of resource mapping seems to be neat and seems to provide a nice performance.
* Pipelined Ring Parallelism seems to be a little extension that seems to be effective.
* Code is open-sourced https://anonymous.4open.science/r/prima-cpp

**Weaknesses:**

* Experiments seem to be limited to a few devices and it does not really show the generalizability of the work.
* Only single inference request is presented whereas real use-cases may really want multi-user (multi-request) workloads. In other words, it seems that the overall serving related discussions are missing in the paper.
* Extending from previous point, it seems to lack comparisons to vLLM, SGLang, ... which are very widely used frameworks in both academia and industry.
* Power consumption and energy efficiency OR cost vs cloud inferences seem to be missing.
* More analysis of the performance breakdown would really help.
* Details about how it is designed is not really presented in the paper, really limiting the amount of insight that the readers can get from reading the paper.
* Discussion about how the work relates to various optimization approaches such as quantization, pruning are missing.

**Questions:**

* How well do the experiments generalize beyond the few tested devices?
* How does the system handle multi-user or multi-request workloads common in real applications?
* Why are comparisons to frameworks like vLLM and SGLang missing?
* What is the power consumption or energy efficiency compared to cloud inference costs?
* Can the authors provide a detailed performance breakdown showing compute, communication, and disk I/O contributions?
* Can more system design and implementation details be provided to give readers better insight?
* How does the proposed approach relate to optimization techniques such as quantization and pruning?

I like the paper overall. However, it has a lot of missing parts hence the score (4). I would be open to reevaluating.

---

> ### Author Response · Authors · 2025-11-25
> **Authors' response to Weaknesses 1-3 and Questions 1-3.**
>
> Dear reviewer, thank you for recognizing our work and for giving us the opportunity to have it re-evaluated.
>
> ---
>
> **1. Weakness 1 and Question 1:** We appreciate your valuable suggestions. As we are not clear whether you referred to the diversity or the scale of the testbed, we address both aspects below.
>
> **For the diversity of testbeds:** We would like to clarify that we evaluated prima.cpp on 6 heterogeneous testbeds (Table 3; Appendices A.4/A.10, A.5/A.9, and the new A.11/A.15, A.13/A.15, A.19), spanning 3 real home networks with different latencies, rather than a single cluster. In total, we used 18 devices: 5 Mac devices (Air/M1/M3/M4/Mini), 4 Linux laptops with 3070/4060 Ti/4090 GPUs and CPU-only, 6 Linux PCs with 1080 Ti/2080 Ti/3090 GPUs and CPU-only, 2 phones and one tablet (HarmonyOS and Android). Some only had HDDs/UFSs, and some featured low compute power, high latency, and small memory. These diverse testbeds and devices cover a wide range of real-world network and hardware conditions, and all derived conclusions remain consistent with those reported in Table 4.
>
> **For the scale of testbeds:** To address your concern, we have added Appendix A.19 to compare llama.cpp, exo, dllama, and prima.cpp on a larger-scale heterogeneous testbed with 10 devices (as also requested by Reviewer kAiH), and show Halda’s device selection capability versus human experts. The results show that:
> - Our prima.cpp outperforms llama.cpp, exo, and dllama in both TPOT and OOM risks, consistent with the findings in Table 4.
> - Halda selected 3 devices out of 10 to form the best-performing cluster, achieving a TPOT 6% lower than the human experts' rule and 99% than exo’s rule.
>
> We hope this addresses your concerns about generalizability.
>
> ---
>
> **2. Weakness 2 and Question 2:** We appreciate your valuable suggestion. This paper focuses on improving single-request inference, but **prima.cpp does implement continuous batching and can handle concurrent requests (as mentioned in footnote 1). We have added Appendix A.11 to include this ablation study.**
>
> Below, we briefly summarize the key points:
> - We run 14B and 70B models with 1-40 concurrent requests on 2 heterogeneous testbeds: a low-memory testbed and a medium-memory testbed.
> - When the model is large, or memory is tight, throughput grows sublinearly, peaks around 8-16 concurrent requests, then declines. This is expected: higher concurrency consumes more memory for KV caches and compute buffers, leading to more frequent model weight eviction and reloads.
> - When memory is more abundant, throughput scales nearly linearly at first, saturates around 32 concurrent requests, and then drops.
> - These “safe” ranges cover typical household usage, which is often 1-8 concurrent requests.
>
> We also describe the implementation of continuous batching in prima.cpp in Appendix A.12. On the head device, a job queue is maintained, where multiple jobs are batched into micro-batches using continuous batching and then executed by prima.cpp to produce batched outputs. Unlike PP, which may process multiple micro-batches concurrently, PRP handles only one micro-batch at a time to avoid contention for memory and disk across micro-batches.
>
> ---
>
> **3. Weakness 3 and Question 3:** Thank you for this comment. Our work targets home/edge clusters with heterogeneous CPU/GPU devices, mixed OS/backends, tight RAM/VRAM, and Wi-Fi links. As noted in footnote 7 of the paper, **vLLM and SGLang are designed for datacenter/server settings, and none of them could run on our testbeds.**
> - *vLLM* : (a) No official mobile/tablet support. (b) CPU‑only on Apple Silicon, and optimizations like quantization and PagedAttention are not available (slow and easy to OOM). (c) Requires identical nodes, mixing GPUs/CPUs, OSes, or CUDA/Metal is not allowed (from official doc), and no one has ever succeeded.
> - *SGLang* : (a) No official Apple Silicon/mobile/tablet support; (b) Official SGLang reuses vLLM for optimizations, like PagedAttention and quantization, so vLLM's limitations also affect SGLang. (c) Backend libraries and operators are not supported for Apple/mobile platforms. Also, no successful reports were found on such a mixed testbed.
> Given this mismatch, we select home/mobile device-oriented, broadly used baselines that can run in our setting: llama.cpp (90K stars), and 2 distributed systems exo (32K stars) and dllama (2.7K stars).

---

> ### Author Response · Authors · 2025-11-25
> **Authors' response to Weaknesses 4-7 and Questions 4-7.**
>
> **4. Weakness 4 and Question 4:** Thank you for this valuable suggestion. We agree that energy is a key metric for both edge and cloud serving. To address your concern, **we have added Appendix A.13 to compare the energy consumption** per 1K output tokens across 3 settings: local single-device, local prima.cpp-distributed, and cloud inference.
>
> Here we briefly summarize the key points:
> - **Local single-device vs. prima.cpp distributed:** Switching from single-device llama.cpp to distributed prima.cpp reduces per-device energy by 91–99% and total energy by 57–90% for 1K tokens, and assigning a few layers to low-power helpers can shorten the critical path at low energy cost.
> - **Local serving vs. cloud serving:** The total energy consumption of the cloud GPU is 28% lower than a local cluster. If we take into account the cooling power consumption of data centers, the energy gap narrows. Regarding energy cost, many countries/regions use tiered electricity pricing: residential prices are about 30% cheaper than commercial prices, which offsets the energy-cost gap between cloud and local serving.
>
> To sum up, **prima.cpp significantly reduces energy consumption compared to local single-device serving, while maintaining comparable energy use and cost to cloud serving.** If we consider token pricing, bandwidth fees, cloud failure, and cloud costs, local serving with prima.cpp would be more cost-effective.
>
> ---
>
> **5. Weakness 5 and Question 5:** Thank you for this valuable suggestion. To address your concern, **we have added Appendix A.14 to include a performance breakdown to show the existence and impact of prefetch-release conflicts, as well as the latency composition of disk, compute, and communication before and after applying PRP/Halda.**
>
> Below, we briefly summarize the key points:
> - Results in Fig. 14 confirm the existence of prefetch-release conflicts: even with prefetching completes, the in-compute reload bytes approach 25 GiB (roughly the total model weights offloaded to CPUs), indicating that nearly all prefetched weights were reclaimed before use, which is clear evidence of prefetch-release conflicts. The sharp drop in cache hit rate causes high-frequency page faults and off-chip memory access, making compute time surge and dominate latency.
> - From the in-compute reload size curve in Fig. 15, PRP clearly reduces the amount of prefetched data reclaimed by the OS. This raises cache hit rates and sharply shortens compute time, showing that PRP effectively resolves prefetch-release conflicts and improves kernel efficiency over PP. In addition, PRP reduces exposed disk I/O more effectively. Overall, PRP reduces per-token latency to just 20-23% of that under PP.
> - The results of heuristic rules (MemSched, PerfSched) are highly random, but Halda performs much smarter scheduling: it always finds a much better partitioning rule. This increases devices’ cache hit rate and minimizes overall compute time while keeping global prefetch time low. As a result, Halda reduces TPOT by ~50\% compared to exo’s layer partitioning rule.
>
> ---
>
> **6. Weakness 6 and Question 6:** Thank you for pointing out this. Due to page limits, the main text focuses on presenting the core ideas behind our key contributions, while system implementation details were distributed across several sections. To address your concern, **we have added Appendix A.12, which provides a consolidated description of prima.cpp framework.**
>
> In Appendix A.12, we illustrate the overall framework design of prima.cpp, and describe how each component is implemented, including:
> - Mmap- and prefetch-based PRP;
> - Profiling and heterogeneity-aware layer partitioning;
> - Multi-request batching;
> - Speculative decoding;
> - Node join, dropout, and automatic device selection;
> - Cross-LAN collaboration.
>
> ---
>
> **7. Weakness 7 and Question 7:** Thank you for this comment. We would like to clarify that the main contributions of this paper lie in introducing a new parallelism paradigm (PRP) and a heterogeneity-aware workload scheduler (Halda), rather than proposing new quantization or pruning methods. Our prima.cpp supports deploying quantized models (e.g., Q4K, Q6K, Q80, IQ1), but it does not introduce new quantization schemes. Instead, it explicitly models the effects of quantization on computation, memory, communication, and disk I/O within the LDA optimization model, allowing Halda to account for these factors when making layer partitioning decisions. This has been discussed in Appendix A.3. Therefore, prima.cpp is compatible with existing quantization and pruning methods.
>
> ---
>
> We hope this rebuttal addresses your concerns, and we appreciate your kindness in reevaluating our score.

---

### Author Response · Authors · 2025-11-25
**Added 12 experiments and 9 appendices to address reviewers' concerns.**

Dear AC and reviewers:

Sorry for the late reply.

To address your concerns, we have added 12 experiments and 9 appendices. Please check them in the latest revised paper PDF. You can find it in "Supplementary Material".

We hope this rebuttal addresses your concerns, and we appreciate your kindness in reevaluating our score.

Best regards!

---

### Author Response · Authors · 2025-11-29
**Author Final Remarks by Authors**

We sincerely thank the AC and the reviewers for their recognition of our work and valuable feedback. Here, we provide a summary of how we have addressed each reviewer's concerns to ease your evaluation.

---

**Reviewer 7c8W:**
- **Weakness 1 and Question 1:** We have added Appendix A.19 to compare llama.cpp, exo, dllama, and prima.cpp on a larger-scale heterogeneous testbed with 10 devices. The results are consistent with the findings in the main text.
- **Weakness 2 and Question 2:** We have added Appendix A.11 to include an ablation study on multi-request serving.
- **Weakness 3 and Question 3:** vLLM and SGLang are designed for datacenter/server settings, and none of them could run on our testbeds.
- **Weakness 4 and Question 4:** We have added Appendix A.13 to compare the energy consumption across 3 settings: local single-device, local prima.cpp-distributed, and cloud inference.
- **Weakness 5 and Question 5:** We have added Appendix A.14 to include a performance breakdown to show the existence and impact of prefetch-release conflicts, as well as the latency composition of disk, compute, and communication before and after applying PRP/Halda.
- **Weakness 6 and Question 6:** We have added Appendix A.12, which provides a consolidated description of prima.cpp framework.
- **Weakness 7 and Question 7:** Prima.cpp is compatible with existing quantization and pruning methods, and prima.cpp has considered quantization in its LDA optimization model.

---

**Reviewer 6ZVN:**
- **Weakness 1:** SpecExec/UMbreLLa is designed for single-device inference, but prima.cpp is designed for distributed on-device inference. Therefore, SpecExec/UMbreLLa are not our baselines. Nevertheless, we conducted a comparison with them: UMbreLLa runs into OOM on our testbed, even after device upgradation, it is still 12% slower than our prima.cpp.
- **Weakness 2:** We never write model weights or KV data to disk, nor raise memory pressure to cause swapping. There is NO write-erase cycle, the disk handles read I/O only, and heavy reads have minimal impact on SSD or UFS lifespan.
- **Weakness 3:** We have added Appendix A.15 to include an ablation study on context length, and Appendix A.11 to include an ablation study on concurrent requests.
- **Question 1:** We have added Appendix A.13 to compare the energy consumption across 3 settings: local single-device, local prima.cpp-distributed, and cloud inference.
- **Question 2:** Please see the rebuttal.

---

**Reviewer kAiH:**
- **Weaknesses 1, 3, and Question 1 (sharing the same concerns):**
  - We have added Appendix A.14 to include a performance breakdown to show the existence and impact of prefetch-release conflicts, as well as the latency composition of disk, compute, and communication before and after applying PRP/Halda. This experiment runs on a heterogeneous testbed.
  - We reproduced the experiment of Fig. 2 on a heterogeneous testbed in Appendix A.16. The results are consistent with the findings reported in the main text.
- **Weakness 2 and Questions 2,3 (sharing the same concerns):**
  - Our prima.cpp produces outputs identical to the baseline (llama.cpp).
  - Removing D4 is NOT unfair. Instead, this is a limitation inherent to exo's implementation, and it benefits exo.
  - We have added Appendix A.19.1 and updated Tables 4, 5, 8, 9 to include more valid dllama data. Their results are consistent with the conclusions in Table 4.
  - The RPC backend of llama.cpp is intended for low-latency HPC environments, not high-latency Wi-Fi. It performs poorly on our testbeds.
- **Weakness 4 and Questions 4, 6 (sharing the same concerns):**
  - Our 6 testbeds used 18 heterogeneous devices and covered a wide range of real-world network and hardware conditions, and all derived conclusions remain consistent with those reported in Table 4.
  - We have added Appendix A.19 to compare llama.cpp, exo, dllama, and prima.cpp on a larger, heterogeneous testbed with 10 devices.
  - We have added Appendix A.19.2 to show Halda’s device selection capability on a larger heterogeneous pool with 10 devices.
  - We have added Appendix A.17 to include an ablation study on network bandwidth and link latency.
  - For disk and memory profiling, please see the rebuttal.
- **Weakness 5:** We have added Appendix A.12 to include a consolidated description of prima.cpp framework.
  - For the queries, please see the rebuttal.
  - We have added Appendix A.14 to include empirical evidence for the prefetch-release conflict.
- **Questions 5:**
  - In our practice, Halda typically converges within 1-3 iterations, and scheduling delay is only 10-12ms, negligible compared to the total inference time.
  - No oscillations were observed.
- **Questions 7:** We have added Appendix A.18 to include a comparison between the idle and busy system states. For other queries, please see the rebuttal.

---

**We added 12 experiments and 9 appendices in total** and we believe that **all reviewers' concerns have been perfectly addressed**.

---

### Meta-Review · Area_Chair_1xRq · 2026-01-07

**Summary:**

The paper proposes prima.cpp, a distributed on-device LLM inference system that enables 30–70B models to run on heterogeneous, low-resource home clusters. Its key technical contributions are pipelined-ring parallelism (PRP) to overlap disk I/O/compute/communication and mitigate mmap “prefetch-release” issues, plus Halda, a heterogeneity-aware scheduler that optimizes layer partitioning and CPU/GPU splits under memory constraints.

Most concerns of reviewers were addressed. Therefore, there are good reasons to accept the paper.

**Reviewer Concerns:**

* Small-scale evaluation [7c8W][kAiH].
 Rebuttal handling: Largely addressed by expanding the experimental scope: the authors claim evaluation across 6 heterogeneous testbeds (18 devices) and add a 10-device heterogeneous testbed comparison (Appendix A.19) to support scalability and consistency of conclusions. This directly targets the “few devices / single cluster” concern, though the strength depends on how clearly the new appendix reports setups, variability, and failure cases.

* Missing multi-request / serving realism (concurrency, batching) [7c8W][6ZVN] .
Rebuttal handling: Addressed via a new concurrent-requests ablation (Appendix A.11) and explanation of continuous batching (Appendix A.12). The response is plausible and reasonably concrete (throughput trends, “safe” concurrency ranges), but it remains an ablation-style treatment rather than a full serving evaluation (e.g., tail latency, request mixes, admission control).

* Missing or questionable baselines (datacenter frameworks / alternative systems) [7c8W][6ZVN][kAiH], vLLM / SGLang absent [7c8W][kAiH].
 Rebuttal handling: Addressed by arguing they cannot run in the target environment (lack of Apple/mobile support, heterogeneous-node constraints, backend/operator limitations). This is a reasonable scope justification; it would be most convincing if the revised paper includes minimal “attempted to run / failure mode” evidence rather than only assertions.

 * SpecExec / UMbreLLa absent [6ZVN] .
Rebuttal handling: Addressed by clarifying category mismatch (single-device vs distributed) and adding a direct comparison: UMbreLLa OOM on a lower-RAM device and is reported ~12% slower even after RAM upgrade, while prima.cpp achieves speedup by adding a helper device. This is a substantive response, though hardware configurations differ and the result hinges on a specific cost/upgrade narrative.

  * Fairness/completeness of exo/dllama/llama.cpp comparisons [kAiH].
 Rebuttal handling: Partially addressed: they justify exo limitations (phone root requirement), add more dllama measurements (Appendix A.19.1, updated tables), and explain why llama.cpp RPC is unsuitable over Wi-Fi (very high TPOT on their test). This improves the story, but a reviewer could still want a stronger “best effort” multi-device llama.cpp baseline (even if it performs poorly) and clearer standardization of quantization/quality settings across systems.

* Lack of model quality evaluation under quantization / speculative decoding [kAiH].
 Rebuttal handling: Partially addressed. The authors claim prima.cpp outputs are identical to llama.cpp because it uses the same pre-quantized models and speculative decoding verifies tokens, so perplexity/accuracy can be “referenced from model cards.” This answers the token-identical concern for comparisons where the same quantized target model is used, but it may not fully satisfy the broader experimental fairness issue when baselines use different quantization stacks or when the paper’s tables mix formats (the rebuttal argues they are all “4-bit-like,” but reviewers may still expect at least a minimal quality sanity check on the exact deployed artifacts).

* Missing energy/power/cost discussion [7c8W][6ZVN] .
Rebuttal handling: Addressed via an added energy-per-1K-tokens comparison across single-device local vs distributed local vs cloud (Appendix A.13) and a qualitative cost discussion (cooling, tiered electricity pricing). This is directionally good, though the cost claims could still be seen as somewhat assumption-heavy unless the appendix clearly specifies measurement methodology, hardware, and cloud configuration equivalence.

* Insufficient performance breakdown / unclear PRP mechanism and attribution [7c8W][kAiH] .
Rebuttal handling: Largely addressed by adding a latency composition and conflict evidence breakdown (Appendix A.14) and reproducing key PRP claims on a heterogeneous testbed (Appendix A.16). However, the reviewer’s deeper “causal attribution” concern (PRP vs Halda vs other factors) is only partially settled unless the new experiments explicitly include clean isolations like PP+Halda vs PRP+Halda and PRP without Halda on the same heterogeneous setup.

* Missing implementation/system detail (framework clarity, KV cache placement, speculative coordination, profiling) [7c8W][kAiH] Rebuttal handling: Addressed by adding a consolidated system description (Appendix A.12) and directly answering specifics (no activation checkpointing; KV caches stay with layers; speculative draft stays on head device; profiling via fio/OS APIs; OS/backend type detection). This seems substantially responsive.

* Disk wear concerns (write/erase cycles) [6ZVN].
 Rebuttal handling: Addressed. Authors state they use read-only mmap, no writes of weights/KV, no swapping, and measured write bytes are zero—so SSD/UFS wear from write cycles is not a concern in their design.

* Sensitivity to network conditions / scalability beyond 6 devices / device selection robustness [kAiH].
 Rebuttal handling: Mostly addressed. Authors add network ablation (Appendix A.17), larger-pool selection experiments (Appendix A.19, A.19.2), and Halda overhead/convergence data (typically 1–3 iterations, ~10–12 ms). What may remain is evidence about how often assignments would change in real homes and robustness to more chaotic interference (beyond the “busy system” tests in Appendix A.18).

**Reviewer Scores:**

Reviewer 7c8W: 4 → 6.
(They explicitly said they were open to reevaluating, and the rebuttal hits all their listed gaps: scale, multi-request, energy, breakdown, more details.)

Reviewer 6ZVN: 6 → 6.
(They were already above threshold and very confident; the rebuttal addresses baseline/disk/ablations/energy/backends, so a bump to 8 is possible but less likely.)

Reviewer kAiH: 4 → 6
(Most concerns are answered with new breakdowns, hetero reruns, 10-device scaling/device selection, network sensitivity, Halda overhead, memory methodology.)

---

### Decision · Program_Chairs · 2026-01-26

Accept (Poster)